# Tailoring to the Tails:
# Risk Measures for Fine-Grained Tail Sensitivity

**Christian Fröhlich**                                                   *christian.froehlich@uni-tuebingen.de*
*Department of Computer Science*
*University of Tübingen*

**Robert C. Williamson**                                                 *bob.williamson@uni-tuebingen.de*
*Department of Computer Science*
*and Tübingen AI center*
*University of Tübingen*

**Reviewed on OpenReview:** *https://openreview.net/forum?id=UntUoeLwwu*

## Abstract

Expected risk minimization (ERM) is at the core of many machine learning systems. This means that the risk inherent in a loss distribution is summarized using a single number – its average. In this paper, we propose a general approach to construct *risk measures* which exhibit a desired tail sensitivity and may replace the expectation operator in ERM. Our method relies on the specification of a reference distribution with a desired tail behaviour, which is in a one-to-one correspondence to a *coherent upper probability*. Any risk measure, which is compatible with this upper probability, displays a tail sensitivity which is finely tuned to the reference distribution. As a concrete example, we focus on *divergence risk measures* based on $f$-divergence ambiguity sets, which are a widespread tool used to foster distributional robustness of machine learning systems. For instance, we show how ambiguity sets based on the Kullback-Leibler divergence are intimately tied to the class of subexponential random variables. We elaborate the connection between divergence risk measures and *rearrangement invariant Banach norms*.

## 1   Introduction

A plain language summary of our work and a diagram showing key conceptual elements are in Appendix A.5.

Expected risk minimization is at the core of machine learning systems. This means that the risk inherent in a loss distribution is summarized using a single number – its average. Yet this does not capture the *tail behaviour* of the distribution. Since the expectation is insensitive to such behaviour, two distributions may have the same expectation, although they exhibit very different tails. In the context of machine learning, this can be problematic: extreme losses correspond to extremely bad predictions, or, for instance in the context of reinforcement learning, even lead to disastrous consequences. In many settings, there appear to be legitimate motivations for sacrificing some average performance in order to avoid rare but extreme losses. A decision maker with this attitude is said to be *risk averse.*

Another issue to consider is that the empirical distribution perfectly corresponds to the 'true' distribution only in the limit of infinite data. With finite data, the observed tails could be much lighter than the 'true' tail of the loss distribution, since extreme events are by definition infrequently observed. This suggests that a cautious approach is to pretend that the tail is heavier than actually observed. This approach has been studied under the research theme of *distributional robustness*, where expected risk minimization has been replaced by the distributionally robust $f$-divergence risk measure objective:

$$R(X) := \sup\{\mathbb{E}_Q[X] : d_f(Q, P) \leq \varepsilon\}, \tag{1}$$

where $X$ is the random loss variable, the $Q$ are probability measures and the $f$-divergence with divergence function $f$ is defined as

$$d_f(Q, P) \coloneqq \int_\Omega f\left(\frac{\mathrm{d}Q}{\mathrm{d}P}\right) \mathrm{d}P,$$

if $Q$ is absolutely continuous with respect to $P$. Here, the decision maker takes a worst-case attitude with respect to a set of probability measures, which are contained in an $f$-divergence ball of radius $\epsilon$, centered at the base distribution $P$. These probability measures are said to constitute an *ambiguity set*. A decision maker, who takes a worst-case stance towards an ambiguity set, is *ambiguity averse*. In practice, we take $P$ to be the empirical distribution, that is, the training data. Intuitively, the function $f$ controls how much reweighting $\frac{\mathrm{d}Q}{\mathrm{d}P}$ is allowed. As a consequence, we guard against possibly different tail behaviour of the 'true' distribution.

The goal of this paper is to construct tail-sensitive *coherent risk measures* (Section 3.4), where the tail sensitivity can be controlled by the machine learning engineer. Such risk measures are then potential replacements for the expectation in ERM and can model both risk and ambiguity aversion, depending on the decision maker's intention. For instance, the $f$-divergence risk measure $R$ in (1) is a coherent risk measure, and we demonstrate how the function $f$ corresponds exactly to a choice of tail sensitivity. Consequently, the choice of $f$ determines the structure of the divergence ball, that is, the tail behaviour of the reweightings it contains. Due to their popularity as a means to achieve distributional robustness in machine learning, we focus on $f$-divergence risk measures. However, we also explicate its relation to other coherent risk measures with the same *fundamental function*, which corresponds (to some extent) to the same tail sensitivity. To investigate these relations, we view coherent risk measures from the perspective of *rearrangement invariant Banach norms*.

For each class of specified tail behaviour (for instance, exponential tails, Gaussian tails etc.), subject to a suitable equivalence relation, we show how the following objects can be derived:

- An $f$-divergence risk measure as in (1), which can be employed to replace the expectation in the risk minimization loop.

- A *Lorentz* and *Marcinkiewicz* norm and related coherent risk measures, which are further such replacement candidates.

- A *utility-based shortfall risk measure*, which is a *convex* risk measure, with potential applications for instance in reinforcement learning.

The main guiding thread is the family of *Orlicz norms*, which offer fine-grained control over tail risk via the specification of a convex function. The class of such functions is approximately as rich as the class of $f$-divergences. The $f$-divergence risk measure (1) then arises as the *natural extension* of what we call an *Orlicz regret measure*. The theory of Orlicz spaces is well-developed, and not only underlies divergence risk measures but also offers a general approach to obtaining concentration inequalities and maximal inequalities (Edgar & Sucheston, 1992), but it appears that the machine learning community is not aware of this link and thus cannot exploit the rich mathematical literature on the subject.

## 1.1 Related Work

We weave together ideas from different literatures, from coherent risk measures (Artzner et al., 1999; Pflug & Römisch, 2007) and rearrangement invariant Banach spaces (Kreĭn et al., 1982; Bennett & Sharpley, 1988; Rubshtein et al., 2016) to distributional robustness (Rahimian & Mehrotra, 2022).

Our goal is to construct risk measures which are sensitive to the heaviness of the tails. The importance of heavy-tailed phenomena has been popularized by Taleb (2007) and they have been studied generally by Nair et al. (2022) and Resnick (2007). In terms of applications, heavy-tailed phenomena have been investigated in economics (Nordhaus, 2012; Ibragimov et al., 2015), earth sciences (Cavanaugh et al., 2015; Merz et al., 2022), neuroscience (Roberts et al., 2015) and many other areas. Within machine learning, Gurbuzbalaban et al. (2021) have identified heavy tails in the iterates of stochastic gradient descent. Empirical risk minimization

in the context of a heavy-tailed loss distribution has been studied by Brownlees et al. (2015) and Hsu & Sabato (2016). Vladimirova et al. (2019) have demonstrated the heavy-tailed nature of prior distributions in Bayesian deep learning.

We are inspired by the work of Dommel & Pichler (2021), themselves building on Ahmadi-Javid (2012) and Ahmadi-Javid & Pichler (2017), who connected $f$-divergence risk measures to Orlicz norms. Ambiguity sets based on $f$-divergences have received much attention in the machine learning literature so far, see for instance (Ben-Tal & Nemirovski, 1999), (Bayraksan & Love, 2015), (Namkoong & Duchi, 2016), (Shapiro, 2017), (Namkoong & Duchi, 2017), (Hu et al., 2018) and (Duchi et al., 2021). In mathematical finance, the special case of the *coherent entropic risk measure*, which corresponds to the choice of Kullback-Leibler divergence for $d_f$ in (1), was studied by Föllmer & Knispel (2011). Closely related is the *convex entropic risk measure*, an empirical variant of which has been proposed by Li et al. (2021) in machine learning. Standard material on Orlicz spaces can be found in (Krasnoselskii & Rutickii, 1961), (Bennett & Sharpley, 1988), (Edgar & Sucheston, 1992), (Kosmol & Müller-Wichards, 2011), (Pick et al., 2013) and (Rubshtein et al., 2016).

## 1.2 Contributions

We define the *Orlicz regret measure*, which is an asymmetric variant of an Orlicz norm, and show that its *natural extension* yields the $f$-divergence risk measure. We embed the Orlicz regret measure and the divergence risk measure in the mathematical framework of rearrangement invariant Banach function spaces and study their fundamental functions. Subsequently we explicate how the fundamental function is, via its one-to-one relation to the *envelope function*, intricately linked to the tail sensitivity of a risk measure. We show how one can therefore construct a corresponding Marcinkiewicz norm, Orlicz norm (hence, a divergence risk measure) and a Lorentz norm, which share this tail sensitivity. We illuminate the connections and (non)-equivalences between these norms. Overall, we do not attempt to establish that one among these three families of norms is 'superior', but rather to analyze their relations and implications. Finally, we link our approach to tail sensitivity to *Orlicz deviation inequalities*, of which many well-known concentration inequalities are special cases.

## 1.3 Use in Machine Learning

In a supervised learning problem, we may wish to replace the expectation operator in ERM by a *risk measure*:

$$\arg\min_{f \in \mathcal{F}} \mathbb{E}[\ell(f(X), Y)] \longrightarrow \arg\min_{f \in \mathcal{F}} R(\ell(f(X), Y))$$

for some risk measure $R$, a function $f$ from some hypothesis space $\mathcal{F}$, a loss function $\ell$, input $X$ and ground truth labels $Y$. Similarly, in reinforcement learning problems, a risk measure may replace the expectation operator in the maximization of *expected* discounted sum of returns. Correspondingly, in *empirical* risk minimization, the risk measure can be applied to the empirical loss distribution.

Possible motivations for replacing the expectation with a risk measure are manifold: in machine learning, risk measures have been employed to express a risk-averse attitude (Tamar et al., 2015; Dabney et al., 2018; Singh et al., 2020; Urpí et al., 2021; Vijayan & Prashanth, 2022) or to achieve distributional robustness (Namkoong & Duchi, 2016; Duchi et al., 2021; Zhang et al., 2021). Moreover, consider the case where the individual losses correspond to individual humans: in this context, it has been argued that risk measures can promote subgroup fairness (Hashimoto et al., 2018; Williamson & Menon, 2019).

We restrict ourselves to theoretical analysis as there is ample experimental evidence for the suitability of risk measures in different settings in the aforementioned works. In addition, Chouzenoux et al. (2019) and Fröhlich & Williamson (2022) provide general treatments on risk measures in machine learning, including experimental evidence. Our focus is therefore on drawing connections and deepening our theoretical understanding. For aspects regarding practical computation see Section 8.

A question remains: which risk measure to choose in a given problem setting? The focus of this paper is to give the reader guidance and intuition to answer this question for themselves, relative to the problem at hand.

Specifically, our focus is the *tail sensitivity* of risk measures. In the previously mentioned problem settings – aiming for risk aversion, distributional robustness or fairness – the tail behaviour of the loss random variable deserves special attention. For instance, a tail event in a fairness context is an individual who receives an extreme loss. Our goal is therefore *tailoring to the tails* instead of *pandering to the masses*[1].

### 1.4 Notation and Assumptions

Throughout, we assume an atomless probability space $(\Omega, \mathcal{F}, P)$, where $P$ is the Lebesgue measure and we take $\Omega = [0, 1]$. We denote the set of probability measures $Q$ which are absolutely continuous with respect to $P$ as $\mathcal{Q}$. Let $\mathcal{M}$ denote the class of Lebesgue measurable functions from $\Omega$ to the reals $\mathbb{R}$, i.e. random variables; however, we have no regard for non-measurable functions throughout and thus often drop writing $X \in \mathcal{M}$. By writing $P$-*a.e.* we mean "almost everywhere with respect to the measure $P$". We denote by $\mathcal{L}^p$ the Lebesgue spaces, i.e. the set of functions with finite $p$-th moment:

$$||X||_p := \begin{cases} \left( \int_0^1 |X|^p \, \mathrm{d}P \right)^{\frac{1}{p}} & 1 \le p < \infty \\ \text{ess sup}(X) & p = \infty, \end{cases} \qquad \mathcal{L}^p := \{ X \in \mathcal{M} : ||X||_p < \infty \},$$

where $\text{ess sup}(X) := \inf \{ \lambda \ge 0 : P(\{\omega \in \Omega : |X(\omega)| > \lambda\}) = 0 \}$. We write $\mathbb{R}^+ = [0, \infty)$ for the nonnegative real line with 0 included and $\mathbb{R}^- = (-\infty, 0)$. By $X^+ := \max(0, X)$ we denote the positive part of a random variable. We write Radon-Nikodym derivatives as $\frac{\mathrm{d}Q}{\mathrm{d}P}$. For a random variable $X$ with cumulative distribution function $F_X(x) := P(\{\omega \in \Omega : X(\omega) \le x\})$, we define the quantile function $F_X^{-1}$ as the generalized inverse of the distribution function: $F_X^{-1}(q) := \sup\{\lambda \ge 0 : F_X(\lambda) < q\}$. For expectations $\mathbb{E}_P[X]$ with respect to the base measure $P$, we often drop the subscript and simply write $\mathbb{E}[X]$. We write indicator functions as $\chi_A$ and convex indicator functions as $i_A$:

$$\chi_A(\omega) := \begin{cases} 1 & \omega \in A \\ 0 & \omega \notin A, \end{cases} \qquad i_A(\omega) := \begin{cases} 0 & \omega \in A \\ \infty & \omega \notin A \end{cases}.$$

By an increasing function we mean a non-decreasing function; similarly, a decreasing function is non-increasing. We write $\phi(0+)$ for the right-hand limit given by $\lim_{x \downarrow 0} \phi(x) = \phi(0+)$. When defining a function $f : \mathbb{R}^+ \to \mathbb{R} \cup \{\infty\}$, we abuse notation and implicitly take this to imply that $f(x) = \infty$ for $x < 0$. The convex conjugate of a function $f : \mathbb{R} \to \mathbb{R} \cup \{\infty\}$ is the function $g : \mathbb{R} \to \mathbb{R} \cup \{\infty\}$:

$$g(y) := \sup_{x \in \mathbb{R}} xy - f(x).$$

Throughout, $f$ and $g$ will denote a conjugate pair of divergence functions (Definition 3.1) and $\Phi$ and $\Psi$ will denote a conjugate pair of Young functions (Definition 2.1).

## 2 Orlicz Norms and Spaces

The motivation behind Orlicz norms is to measure the size of a function in a way which is sensitive to tail behaviour. Often they are introduced as generalizations of $\mathcal{L}^p$ spaces. As a first example, the variance indeed is sensitive to tails and therefore does not exist (is infinite) for many random variables.

**Definition 2.1.** *A function $\Phi : \mathbb{R}^+ \to \mathbb{R}^+ \cup \{\infty\}$ is called Young function if it is left-continuous, increasing, convex, satisfies $\Phi(0) = 0$ and is nontrivial (not identical to the constant zero or infinity function).*

We emphasize that here *increasing* means *non-decreasing*. Since $\Phi$ is convex, continuity is guaranteed on the interior of its effective domain and therefore the left-continuity concerns only the point

$$b_\Phi := \sup\{\Phi < \infty\}.$$

---

[1]That "pandering to the masses" is the opposite of "tailoring to the tails" was suggested to us by Peter Kootsookos.

Note that it is possible that $\Phi(x) = 0$ for $x > 0$. We denote the convex conjugate of $\Phi$ as $\Psi : \mathbb{R}^+ \to \mathbb{R}^+ \cup \{\infty\}$, that is:

$$\Psi(y) := \sup_{x \geq 0} xy - \Phi(x),$$

where the restriction to the nonnegative half line is a consequence of our assumption that $\Phi$ takes on $+\infty$ on the negative half line. If $\Phi$ is a Young function, then so is $\Psi$. In the following, $\Phi$ and $\Psi$ will always refer to a pair of conjugate Young functions.

**Definition 2.2.** *The Luxemburg norm of the random variable $X$ is defined as:*

$$||X||_\Phi^L := \inf \left\{ \lambda > 0 : \mathbb{E}\Phi\left[\frac{|X|}{\lambda}\right] \leq 1 \right\} \quad \forall X \in \mathcal{M},$$

which is the gauge of the set $\{Z \in \mathcal{M} : \mathbb{E}\Phi(|Z|) \leq 1\}$.

**Definition 2.3.** *The Orlicz norm of the random variable $X$ is defined as:*

$$||X||_\Phi^O := \sup\{\mathbb{E}[XZ] : Z \in \mathcal{L}^1, \mathbb{E}[\Psi(|Z|)] \leq 1\} \quad \forall X \in \mathcal{M}.$$

Observe that the conjugate $\Psi$ appears here. Confusingly, the Luxemburg norm (Luxemburg, 1955) is sometimes simply called Orlicz norm in the literature. In fact, these norms are equivalent by a factor of 2:

$$||X||_\Phi^L \leq ||X||_\Phi^O \leq 2||X||_\Phi^L$$

and thus they are finite for the same set of functions. Therefore it is inconsequential whether one takes the Luxemburg or Orlicz norm in the following definition.

**Definition 2.4.** *The Orlicz space $\mathcal{L}^\Phi$ is given by:*

$$\mathcal{L}^\Phi := \{X \in \mathcal{M} : ||X||_\Phi^O < \infty\}.$$

For any Young function $\Phi$, the Orlicz space $\mathcal{L}^\Phi$ is a Banach space, i.e. a complete normed vector space. It is furthermore a Banach lattice, that is, it is monotone in the sense that:

$$|X| \leq |Y| \text{ P-a.e.} \implies ||X||_\Phi^O \leq ||Y||_\Phi^O.$$

Important in the following development will be an identical expression for the Orlicz norm, which is also called the *Amemiya norm* (Hudzik & Maligranda, 2000):

$$||X||_\Phi^O = \inf_{t>0} t \left( 1 + \mathbb{E}\Phi\left(\frac{|X|}{t}\right) \right). \tag{2}$$

This representation is the key to compute Orlicz norms in practice.

## 3 Orlicz Regret Measures

The problem with Orlicz norms in our setting is that, in virtue of satisfying the properties of a norm, they treat losses and gains in the same way. We follow the machine learning convention that losses correspond to positive values, hence gains are negative values. Since we aim for an asymmetric setup, we introduce Orlicz regret measures as the asymmetric variants of Orlicz norms and link them to $f$-divergence risk measures.

### 3.1 $f$-Divergences

First, we define the function class which we consider in the following and their associated $f$-divergences.

**Definition 3.1.** *A divergence function $f$ is a proper, lower semi-continuous, convex function $f : \mathbb{R}^+ \to \mathbb{R}^+ \cup \{\infty\}$, which satisfies $f(1) = 0$, $f(0) < \infty$ and the supercoercivity condition $\lim_{x \to \infty} \frac{f(x)}{x} = \infty$.*

The conditions that $f(0) < \infty$ and $\lim_{x \to \infty} \frac{f(x)}{x} = \infty$ play an important role. They are non-standard restrictions that we impose here and will be further discussed.

We denote the convex conjugate of $f$ as $g$ throughout the paper. It has the following properties.

**Proposition 3.2.** *The conjugate $g$ is finite on $\mathbb{R}$, convex and increasing.*

*Proof.* The statement, which relies on the supercoercivity of $f$, is well known and a proof is included in Appendix A.1.1. $\qquad\square$

**Definition 3.3.** *Given a divergence function $f$, the $f$-divergence between probability measures $Q$ and $P$ is defined as:*

$$d_f(Q,P) := \int_\Omega f\left(\frac{\mathrm{d}Q}{\mathrm{d}P}\right) \mathrm{d}P = \mathbb{E}_P f\left(\frac{\mathrm{d}Q}{\mathrm{d}P}\right),$$

*if $Q$ is absolutely continuous with respect to $P$, and $+\infty$ otherwise.*

A simple shift of perspective, which will be important in the following, is to view $\frac{\mathrm{d}Q}{\mathrm{d}P}(\omega)$ as a single random variable rather than as a fraction of two densities.

We write $\mathfrak{F}$ for the set of divergence functions $f$ and $\mathfrak{F}_Y$ for the subset of divergence functions $f$, which are also valid Young functions. We call elements of $\mathfrak{F}_Y$ *Young divergence functions*. Since we assumed nonnegativity, the only difference to a Young function is that a general divergence function $f$ might be positive and decreasing for $0 \le x < 1$. To a given divergence function $f$, we can associate a canonical Young function $\bar{f}$ by setting:

$$\bar{f}(x) := \begin{cases} 0 & 0 \le x \le 1 \\ f(x) & x > 1, \end{cases}$$

as in (Dommel & Pichler, 2021). We write $\bar{g}$ for the convex conjugate of $\bar{f}$. We will see (Proposition 3.11) that it stands in a close relationship to the original $f$. For various levels of risk aversion[2] $\varepsilon > 0$, we define the shorthand notation $f_\varepsilon(x) = \frac{1}{\varepsilon}f(x)$ and $g_\varepsilon(y) = \frac{1}{\varepsilon}g(\varepsilon y)$, which is the convex conjugate of $f_\varepsilon$. The value of $\varepsilon$ is not of theoretical importance: if $f$ is a divergence function, so is $f_\varepsilon$.

**Example 3.4.** A prominent example is the Kullback-Leibler divergence, where $f_{\mathrm{KL}}(x) := x\log(x) - (x-1)$. It can also be obtained as $f(x) = x\log(x)$, but the affine shift guarantees nonnegativity without altering the divergence. We complete its definition implicitly by setting $f_{\mathrm{KL}}(0) := \lim_{x \to 0} f_{\mathrm{KL}}(x) = 1 < \infty$. The induced $f$-divergence is

$$d_{\mathrm{KL}}(Q,P) = \int_\Omega \log\left(\frac{\mathrm{d}Q}{\mathrm{d}P}\right) \mathrm{d}Q.$$

The conjugate of $f_{\mathrm{KL}}$ is $g_{\mathrm{KL}}(y) = \exp(y) - 1$.

**Example 3.5.** The $\chi^2$ divergence is given by $f_{\chi^2}(x) = (x-1)^2$. Then the divergence is

$$d_{\chi^2}(Q,P) = \int_\Omega \left(\frac{\mathrm{d}Q}{\mathrm{d}P}\right)^2 \mathrm{d}P - 1.$$

Note that the conjugate of $f_{\chi^2}$ is $g_{\chi^2}(y) = \max(-2,y) + \max(-2,y)^2/4$, since we take $f$ to be infinite on the negative half line. In some tables, the reader will find that $g(y) = y + y^2/4$, which is the conjugate if the effective domain of $f(x) = (x-1)^2$ is $\mathbb{R}$. The difference is crucial in our asymmetric setting.

**Example 3.6.** Consider the pathological divergence function

$$f_{\mathbb{E}}(x) := i_{[0,1]}(x) = \begin{cases} 0 & 0 \le x \le 1 \\ \infty & x > 1. \end{cases}$$

We later observe that the corresponding divergence risk measure in (1) is simply the expectation $\mathbb{E}$.

---

[2]For simplicity, we shall use the term "risk aversion level" throughout. However, a divergence risk measure may also capture the arguably distinct phenomenon of ambiguity aversion.

**Example 3.7** (Shapiro (2017))**.** Given some $\alpha \in [0, 1)$, we slightly modify the previous example as:

$$f_{\mathrm{CVar}_\alpha} := i_{[0, 1/(1-\alpha)]}.$$

The corresponding divergence risk measure in (1) is the *conditional value at risk* $\mathrm{CVar}_\alpha$, which is the expectation in the upper $(1 - \alpha)$-tail:

$$\mathrm{CVar}_\alpha(X) := \frac{1}{1 - \alpha} \int_\alpha^1 F_X^{-1}(q) \, \mathrm{d}q.$$

For continuous $F_X$, this can be written in the more suggestive form ("tail-conditional expectation"):

$$\mathrm{CVar}_\alpha(X) = \mathbb{E}\left[X | X \geq F_X^{-1}(\alpha)\right].$$

In the machine learning community, $\mathrm{CVar}_\alpha$ has recently received significant attention, see for instance (Takeda & Sugiyama, 2008), (Fan et al., 2017) and (Curi et al., 2020).

### 3.2  Definition and Functional Properties

We now define the *Orlicz regret measure*, which has similar properties to a norm, yet is fundamentally asymmetric in that it treats losses and gains differently. We will find that it has a very close relation to the divergence risk measure in (1).

**Definition 3.8.** *Given a divergence function $f \in \mathfrak{F}$ with conjugate $g$, the Orlicz regret measure of $X \in \mathcal{M}$ is defined as:*

$$V_g(X) := \inf_{t > 0} t \left(1 + \mathbb{E}g\left(\frac{X}{t}\right)\right).$$

Intuitively, it is $g$ here that plays the role of the Young function of an Orlicz norm. In contrast to the Orlicz norm in (2), there is no use of an absolute value here; hence the behaviour of $g$ on $\mathbb{R}^-$ ($\mathbb{R}^+$) determines the treatment of gains (losses).

**Proposition 3.9.** *For any divergence function $f \in \mathfrak{F}$, the regret $V_g$ has the following properties $\forall X, Y \in \mathcal{M}$:*

    **V1.** $V_g(\lambda X) = \lambda V_g(X) \quad \forall \lambda \in \mathbb{R}^+$     *(positive homogeneity)*

    **V2.** $V_g(X + Y) \leq V_g(X) + V_g(Y)$     *(subadditivity)*

    **V3.** $X \leq Y \ P\text{-}a.e. \implies V_g(X) \leq V_g(Y)$     *(monotonicity)*

    **V4.** $V_g(X) \geq \mathbb{E}[X]$     *(aversity)*

    **V5.** *If $X \geq 0$, then $V_g(X) = 0$ if and only if $X = 0$.*     *(nonnegative point-separating)*

*Proof.* The proof is included in Appendix A.1.2. $\qquad\qquad\qquad\qquad\qquad\qquad\qquad\qquad\qquad\qquad\qquad$ $\square$

It will prove convenient to define Orlicz regret measures for varying levels of risk aversion $\varepsilon$. By a simple scaling, this parameter can be hidden in the function itself. Thus we can express the regret either by hiding the dependence on $\varepsilon$ or by making it explicit.

**Proposition 3.10.** *With $f_\varepsilon(x) = \frac{1}{\varepsilon} f(x)$ and its conjugate $g_\varepsilon(y) = \frac{1}{\varepsilon} g(\varepsilon y)$, we have*

$$V_{g_\varepsilon}(X) = \inf_{t > 0} t \left(1 + \mathbb{E}g_\varepsilon\left(\frac{X}{t}\right)\right) = \inf_{\tilde{t} > 0} \tilde{t}\left(\varepsilon + \mathbb{E}g\left(\frac{X}{\tilde{t}}\right)\right).$$

*Proof.* See Appendix A.1.3. $\qquad\qquad\qquad\qquad\qquad\qquad\qquad\qquad\qquad\qquad\qquad\qquad\qquad\qquad$ $\square$

Recall that $\bar{g}$ is the convex conjugate of $\bar{f}$, where $\bar{f}$ is the Young function canonically associated to $f$.

**Proposition 3.11.** *Let $f \in \mathfrak{F}$ a divergence function with conjugate $g$. Then it holds that $\bar{g} : \mathbb{R} \to \mathbb{R}^+ \cup \{\infty\}$, defined as:*

$$\bar{g}(y) := \begin{cases} 0 & y \leq 0 \\ g(y) & y > 0, \end{cases}$$

*is the conjugate of $\bar{f}$ and it is a valid Young function when restricted to the domain $\mathbb{R}^+$.*

*Proof.* The proof is in Appendix A.1.4. $\qquad\square$

We use $||\cdot||_{\bar{g}}^O$ to denote the Orlicz norm, where the Young function is the restriction of $g$ to the nonnegative domain $\mathbb{R}^+$. A coherent regret measure (we define the term generally in Section 3.4) canonically induces a norm (Pichler, 2013).

**Proposition 3.12.** *Let $f \in \mathfrak{F}$ a divergence function. Then setting*

$$||X||_{V_g} := V_g(|X|)$$

*defines a norm on the space $\mathcal{V}_g := \{X \in \mathcal{M} : ||X||_{V_g} < \infty\}$. In fact it holds that $||X||_{V_g} = ||X||_{\bar{g}}^O$ and $\mathcal{V}_g = \mathcal{L}^{\bar{g}}$. Furthermore, $V_g$ is finite on the space $\mathcal{V}_g$.*

*Proof.*

$$||X||_{V_g} = V_g(|X|) = \inf_{t>0} t\left(1 + \mathbb{E}g\left(\frac{|X|}{t}\right)\right).$$

This is the Amemiya expression of the Orlicz norm with Young function $\bar{g}$, as $g$ is only ever evaluated on the nonnegative domain. Therefore it holds that $||X||_{V_g} = ||X||_{\bar{g}}^O$. The finiteness of $V_g$ on $\mathcal{V}_g$, that is, $X \in \mathcal{V}_g \implies V_g(X) < \infty$, follows from (Pichler, 2013, Prop. 5), essentially from monotonicity of $V_g$. $\qquad\square$

**Corollary 3.13.** *Let $f \in \mathfrak{F}$ a divergence function. Then the norms $||\cdot||_{V_g}$ and $||\cdot||_{V_{\bar{g}}}$ are identical.*

This is due to the inherent asymmetry in our story: we care about *loss tails*, that is, right tails. Consider the meaning of $f(\frac{dQ}{dP})$: the behaviour of $f(x)$ for $0 \leq x < 1$ concerns the situation where $Q$ underestimates with respect to the base measure. In contrast, $f(x)$ for $x \geq 1$ specifies the penalty that overestimation incurs. Due to the absolute value, the Orlicz norm is based only on the treatment of losses. Since $V_g$ and $V_{\bar{g}}$ treat losses the same way, the potentially different behaviour for $0 \leq x < 1$ disappears in the norm-perspective. To capture this desired view, we have imposed the constraint that $f(0) < \infty$: underestimation may only be punished "finitely", hence we can (up to equivalence) neglect underestimation in our approach.

Furthermore, the chosen risk aversion level of the regret measure is not essential, as the resulting norms are equivalent for any choice of risk aversion level.

**Proposition 3.14.** *For any risk aversion level, the norms induced by the regret measures are equivalent. Let $0 < \varepsilon_1 < \varepsilon_2$. We have*

$$||X||_{V_{g_{\varepsilon_1}}} \leq ||X||_{V_{g_{\varepsilon_2}}} \leq \frac{\varepsilon_2}{\varepsilon_1}||X||_{V_{g_{\varepsilon_1}}} \quad \forall X \in \mathcal{M}.$$

*Proof.* The proof is in Appendix A.1.5. $\qquad\square$

**Remark 3.15.** $\mathcal{V}_g$ *is indeed the largest vector space, on which $V_g$ is finite. However, this does* not *mean that $X \notin \mathcal{V}_g \implies V_g(X) = \infty$. This is due to the gain-loss asymmetry: an $X$ with a heavy left tail might have $V_g(X) < \infty$, but $V_g(|X|) = \infty$ and therefore $X \notin \mathcal{V}$. Since we are concerned with right tails, this subtle distinction is inconsequential and we can focus on the natural domain $\mathcal{V}_g$.*

### 3.3 Envelope Representation

Due to subadditivity and positive homogeneity, the Orlicz regret measure $V_g$ possesses an *envelope representation* (or *dual representation*). This insightful representation will, in our case, illuminate the nature of an underlying tail reweighting mechanism in Section 4. As a prelude to the full dual representation, we first consider nonnegative random variables. We denote by $\mathcal{L}_+^{\bar{g}}$ the cone of nonnegative random variables in an Orlicz space $\mathcal{L}^{\bar{g}}$.

**Proposition 3.16.** *Let $f \in \mathfrak{F}$ a divergence function and $\bar{g}$ be the conjugate of the associated Young divergence function $\bar{f}$. For nonnegative $X$, we have:*

$$V_g(X) = \sup\left\{\mathbb{E}[XZ] : Z \in \mathcal{L}^{\bar{f}}, Z \geq 0, \mathbb{E}[\bar{f}(Z)] \leq 1\right\} \quad \forall X \in \mathcal{L}_+^{\bar{g}}.$$

*Proof.* From Proposition 3.12, we know that for nonnegative $X$, we have $V_g(X) = ||X||_{\bar{g}}^O$. Then the statement is simply the definition of the Orlicz norm $||X||_{\bar{g}}^O$. □

The general envelope representation is similar.

**Proposition 3.17.** *Let $f \in \mathfrak{F}$ a divergence function. Then the following envelope representation holds*

$$V_g(X) = \sup\{\mathbb{E}[XZ] : Z \in \mathcal{L}^{\bar{f}}, Z \geq 0, \mathbb{E}[f(Z)] \leq 1\} \quad \forall X \in \mathcal{L}^{\bar{g}}.$$

*Proof.* The proof is in Appendix A.1.6. □

We refer to the $Z$ over which the supremum ranges as *dual variables*. While the dual variables are from $\mathcal{L}^{\bar{f}}$, the *primal* random variables, for which $V_g$ is finite, are from $\mathcal{L}^{\bar{g}}$. To understand the meaning of envelope representations more generally, we turn to the general theory of coherent regret and risk measures.

### 3.4 Coherent Regret and Risk Measures

The framework of coherent risk measures (Artzner et al., 1999; Delbaen, 2002; Föllmer & Weber, 2015) aims to provide tail-sensitive risk control. The goal is to summarize the risk inherent in a financial position, conceptualized as a probability distribution over losses and gains. Here, there is a fundamental asymmetry: exceeding the expected loss is worse than the converse, hence the focus is on the right tail of the distribution.

In the search for possible replacements of the expectation as the tool to measure the risk of a distribution, the question of which properties a suitable replacement must satisfy arises. Artzner et al. (1999) proposed the following influential axioms for coherent risk measures[3]: $\forall X, Y$

**C1.** $R(\lambda X) = \lambda R(X) \quad \forall \lambda \in \mathbb{R}^+ \quad$ (positive homogeneity)

**C2.** $R(X + Y) \leq R(X) + R(Y) \quad$ (subadditivity)

**C3.** $X \leq Y \text{ P-a.e.} \implies R(X) \leq R(Y) \quad$ (monotonicity)

**C4.** $R(X + c) = R(X) + c \quad \forall c \in \mathbb{R} \quad$ (translation equivariance)

These axioms can be motivated from a financial viewpoint, but also from broader considerations regarding decision making under uncertainty; closely related are the works of Gilboa & Schmeidler (1989) in economics and Walley (1991) in imprecise probability.

The divergence risk measure in (1) satisfies all of these and is therefore a coherent risk measure. In contrast, the Orlicz regret measure satisfies C1, C2 and C3, but not translation equivariance (C4). The Orlicz regret measure can be thought of as an "asymmetric norm" (hence *not* a norm), where translation equivariance

---

[3]We translate the axioms to a loss-based orientation. Also, Artzner et al. (1999) worked with finite $\Omega$, hence they stated C3 with the quantification $\forall \omega \in \Omega$ as opposed to P-a.e.

is undesirable. The class of such functionals has been captured by Rockafellar & Uryasev (2013) as the *coherent regret measures*[4].

Coherent regret measures and coherent risk measures are directly characterized by their envelope representations. C1 and C2, together with an appropriate closedness assumption, imply that such an $R$ is a support function. Let $f \in \mathfrak{F}$ and $g$ its conjugate.

**Proposition 3.18** (Biagini & Frittelli (2009, Corollary 28), Arai (2010, Theorem 1))**.** *A proper functional $R : \mathcal{L}^{\bar{g}} \to \mathbb{R}^+ \cup \{\infty\}$ which satisfies C1,C2, C3 and is order lower-semicontinuous[5] allows the envelope representation*

$$R(X) = \sup_{Z \in \mathcal{Z}} \mathbb{E}[XZ], \quad \mathcal{Z} \subseteq \mathcal{L}_+^{\bar{f}}, \text{ where } \mathcal{L}_+^{\bar{f}} := \left\{ Z \in \mathcal{L}^{\bar{f}} : Z \geq 0 \ P\text{-a.e.} \right\},$$

*with some set $\mathcal{Z}$, called the envelope of $R$, of nonnegative random variables.*
*If $R$ is furthermore translation equivariant (C4), then $\mathcal{Z} \subseteq \{Z \in \mathcal{L}_+^{\bar{f}} : \mathbb{E}(Z) = 1\}$.*

Thus the monotonicity of the Orlicz regret measure can be directly observed from its envelope representation. It is, however, lacking translation equivariance. For the use of coherent regret measures in risk-sensitive regression, see (Rockafellar & Uryasev, 2013).

### 3.5 From Regret to Risk

An Orlicz regret measure is in some sense like a norm, but one which distinguishes positive and negative orientation. However, it lacks the property of translation equivariance, which is desirable for a replacement of the expectation as an aggregation operator. Yet there is a simple way to canonically construct a coherent risk measure from a coherent regret measure. In this way, the $f$-divergence risk measure arises from the Orlicz regret measure. The $f$-divergence risk measure is defined as:

$$R_{g,\varepsilon}(X) := \sup \left\{ \mathbb{E}_Q[X] : Q \in \mathcal{Q}, d_f(Q, P) \leq \varepsilon \right\}.$$

For reasons of consistency, we use the conjugate of $f$, which is $g$, in the subscript. Indeed, the conjugate is the appropriate object here, as the following proposition shows. A dual variable $Z$ which satisfies $Z \geq 0$ and $\mathbb{E}[Z] = 1$ is a valid probability density, hence we introduce

$$Q_Z(A) := \int_A Z(\omega) \, \mathrm{d}P(\omega), \tag{3}$$

which is guaranteed to be a probability measure.

**Proposition 3.19.** *The divergence risk measure $R_{g,\varepsilon}(X)$ is given by an infimal convolution of the Orlicz regret measure $V_{g_\epsilon}$. Let $X \in \mathcal{L}^{\bar{g}}$. Then:*

$$R_{g,\varepsilon}(X) = \inf_{\mu \in \mathbb{R}} \mu + V_{g_\varepsilon}(X - \mu) \tag{4}$$

$$= \inf_{\mu \in \mathbb{R}, t > 0} t \left( 1 + \mu + \mathbb{E} g_\varepsilon \left( \frac{X}{t} - \mu \right) \right)$$

$$= \inf_{\tilde{\mu} \in \mathbb{R}, \tilde{t} > 0} \tilde{t} \left( \varepsilon + \tilde{\mu} + \mathbb{E} g \left( \frac{X}{\tilde{t}} - \tilde{\mu} \right) \right). \tag{5}$$

---

[4]Rockafellar & Uryasev (2013) further demanded the aversity condition $V(X) > \mathbb{E}[X]$ for $X \neq 0$. For consistency, we weaken aversity to a weak inequality. Then it is in fact a consequence of "rearrangement invariance", see Proposition 4.3. We only consider rearrangement invariant coherent regret and risk measures throughout the paper. We remark that Rockafellar & Uryasev (2013) do not use the term "coherent regret measure" itself, but instead define regular measures of regret; when adding positive homogeneity, we call it *coherent* by analogy to coherent risk measures

[5]See (Biagini & Frittelli, 2009) for this technical property. We also refer to (Cheridito & Li, 2009) for a discussion of risk measures on *Orlicz hearts*, which are particularly interesting subspaces of Orlicz spaces.

Its envelope representation is:

$$
\begin{aligned}
R_{g,\varepsilon}(X) &= \sup\left\{E[XZ] : Z \in \mathcal{L}^{\bar{f}}, Z \geq 0, \mathbb{E}[Z] = 1, \mathbb{E}f(Z) \leq \varepsilon\right\} \\
&= \sup\left\{\mathbb{E}_{Q_Z}[X] : Z \in \mathcal{L}^{\bar{f}}, Q_Z \in \mathcal{Q}, d_f(Q_Z, P) \leq \varepsilon\right\} \\
&= \sup\left\{E_{Q_Z}[X] : Z \in \mathcal{L}^{\bar{f}}, Q_Z \in \mathcal{Q}, \mathbb{E}_{Q_Z}[Y] \leq V_{g_\varepsilon}(Y) \quad \forall Y \in \mathcal{L}^{\bar{g}}\right\}.
\end{aligned}
\tag{6}
$$

*Proof.* The proof is in Appendix A.1.7. $\qquad\square$

We typically write $R_g$ with $\varepsilon = 1$ in the following. The infimal convolution in (4) ensures translation equivariance of the resulting functional. The combination of monotonicity and translation equivariance then guarantees that the dual variables are legitimate probability measures, that is, $Z \geq 0$ and $\mathbb{E}[Z] = 1$ (see for instance (Fröhlich & Williamson, 2022)). An interesting perspective is also provided by (6): the envelope is the set of linear risk measures, i.e. expectations, which are $L^{\bar{g}}$-pointwise dominated by the regret measure. In the language of the *imprecise probability* literature, this process of projecting from $V_g$ to $R_g$ by forming a supremum over the set of dominated expectations is known as *natural extension* (Walley, 1991; Troffaes & De Cooman, 2014). This provides the rationale for interpreting the envelope as an ambiguity set and embeds the divergence risk measure in the field of imprecise probability.

The infimum-based representation in (5), under various technical assumptions, has been around in the literature (Ben-Tal & Nemirovski, 1999; Bayraksan & Love, 2015; Shapiro, 2017; Dommel & Pichler, 2021). However, our derivation from the Orlicz regret measure provides a new perspective and intuition for the result; we remark that Dommel & Pichler (2021), themselves inspired by Ahmadi-Javid (2012) and Ahmadi-Javid & Pichler (2017), have investigated the divergence risk measure in the context of Orlicz spaces, but without reference to an underlying Orlicz regret measure.

**Example 3.20** (Bayraksan & Love (2015)). With $f_{\text{CVar}_\alpha} := i_{[0,1/(1-\alpha)]}$ we obtain the well-known infimal convolution expression of the conditional value at risk:

$$
\text{CVar}_\alpha(X) = \inf_{\mu \in \mathbb{R}} \mu + \frac{1}{1-\alpha}\mathbb{E}[(X-\mu)^+],
$$

where the special case $\alpha = 0$ recovers the expectation.

Similar to the Orlicz regret measure, the divergence risk induces a norm by setting $||X||_{R_g} := R_g(|X|)$, with the induced space $\mathcal{R}_g = \{X \in \mathcal{M} : R_g(|X|) < \infty\}$. Then the following holds:

**Proposition 3.21.** *The norms $||\cdot||_{R_g}$ and $||\cdot||_{V_g}$ are equivalent. This implies that the spaces $\mathcal{V}_g$ and $\mathcal{R}_g$ are identical.*

*Proof.* Since $||\cdot||_{V_g} = ||X||_{\bar{g}}^O$, Theorem 4.5 in (Dommel & Pichler, 2021) is applicable. The authors did assume finiteness of $f$, but this is not required in the proof of the statement. $\qquad\square$

As a consequence, the divergence risk measure is finite on $\mathcal{V}_g = \mathcal{R}_g$ and this is the largest vector space, on which it is finite (cf. Remark 3.15).

**Corollary 3.22** (Dommel & Pichler (2021)). *Let $f \in \mathfrak{F}$. Then the divergence risk measure is equivalent to its corresponding Young divergence risk measure, that is, $||\cdot||_{R_g}$ is equivalent to $||\cdot||_{R_{\bar{g}}}$.*

*Proof.* The statement holds since $||\cdot||_{R_g}$ and $||\cdot||_{V_g}$ are equivalent and $||\cdot||_{V_g} = ||\cdot||_{V_{\bar{g}}}$ (Corollary 3.13). $\qquad\square$

**Corollary 3.23** (Dommel & Pichler (2021)). *Consider $f_{\varepsilon_1}, f_{\varepsilon_2} \in \mathfrak{F}$, that is, variants of the same underlying divergence function at different risk aversion thresholds. Then $||\cdot||_{R_{g_{\varepsilon_1}}}$ and $||\cdot||_{R_{g_{\varepsilon_2}}}$ are equivalent.*

*Proof.* This readily follows from the equivalence of the underlying Orlicz regret measures. $\qquad\square$

## 4   Rearrangement Invariant Banach Norms and Fundamental Functions

We now embed the Orlicz regret measure and the divergence risk measure in the framework of *rearrangement invariant Banach function spaces*, which provides more insight into the envelope representation and allows us to investigate equivalence relationships of the induced spaces.

A desirable property of a coherent regret or risk measure is *rearrangement invariance*, which is known as *law invariance* in the risk measure literature. The characterizing property is that a rearrangement invariant regret (risk) measure depends only on the distribution of the random variable, not on the order in which outcomes are associated with elementary events $\omega$. To capture this notion, we define the *distribution function*[6] $\mu_X :$ $\mathbb{R}^+ \to [0, 1]$ of a random variable $X \in \mathcal{M}$ as

$$\mu_X(\lambda) := P(\{\omega \in \Omega : |X(\omega)| > \lambda\}).$$

For nonnegative random variables, this decreasing (non-increasing) and right-continuous function is just the survival function $1 - F_X$. We say that random variables $X$ and $Y$ are *equimeasurable* if their distribution functions are the same, that is, $\mu_X(\lambda) = \mu_Y(\lambda) \ \forall \lambda \geq 0$. For each $X \in \mathcal{M}$, the right-continuous generalized inverse of its distribution function $X^* : [0, 1] \to \mathbb{R}^+$,

$$X^*(\omega) := \inf\{\lambda \geq 0 : \mu_X(\lambda) \leq \omega\},$$

is equimeasurable with $X$ itself. We call $X^*$ the *decreasing rearrangement* of $X$. In the language of probability theory, $X^*$ is just the quantile function backwards: $X^*(t) = F^{-1}_{|X|}(1-t)$. As an intuition, $X^*$ is the continuous analogue of sorting a list of values in decreasing order.

### 4.1   A Primer on Rearrangement Invariant Spaces

**Definition 4.1.** *A Banach space $(\mathcal{X}, ||\cdot||)$ is called rearrangement invariant if (Rubshtein et al., 2016)*

> ***R1.*** *If $|X| \leq |Y|$ and $Y \in \mathcal{X}$, then $X \in \mathcal{X}$ and $||X|| \leq ||Y||$.*

> ***R2.*** *If $X$ and $Y$ are equimeasurable and $Y \in \mathcal{X}$, then $X \in \mathcal{X}$ and $||X|| = ||Y||$.*

*We call such a norm $||\cdot||$ rearrangement invariant (r.i.).*

From their definitions, it is clear that $||\cdot||_{V_g}$ and $||\cdot||_{R_g}$ are r.i. norms and therefore $\mathcal{V}_g$ and $\mathcal{R}_g$ are r.i. spaces. In general, any coherent regret measure or risk measure, which satisfies R2, induces a legitimate r.i. norm by taking absolute values.

**Example 4.2.** The Lebesgue spaces $\mathcal{L}^p$, $1 \leq p \leq \infty$, are r.i. spaces, which are in particular Orlicz spaces.

Interestingly, any r.i. space is in between $\mathcal{L}^1$ and $\mathcal{L}^\infty$ due to the following embedding theorem.

**Proposition 4.3** (Bennett & Sharpley (1988, p. 77)). *Let $\mathcal{X}$ be any ri space. Then:*

$$\mathcal{L}^\infty \subseteq \mathcal{X} \subseteq \mathcal{L}^1,$$

*and the norms are in the relationship*[7]*:*

$$\|X\|_{\mathcal{L}^1} \leq \|X\|_{\mathcal{X}} \leq \|X\|_{\mathcal{L}^\infty} \quad \forall X \in \mathcal{M}.$$

Finally, we need the concept of the *associate norm*, which is linked to envelope representations.

---

[6]This is the standard terminology in the rearrangement invariant Banach space literature; it is not to be confused with the *cumulative* distribution function $F_X$.

[7]Under the assumption that $||1||_{\mathcal{X}} = 1$, otherwise renorm first with the multiplicative factor $1/||1||_{\mathcal{X}}$.

**Definition 4.4.** *Let $(\mathcal{X}, ||\cdot||_\mathcal{X})$ a Banach space, where $||\cdot||_\mathcal{X}$ satisfies the monotonicity condition R1. Its associate norm $||\cdot||_{\mathcal{X}'}$ is defined as:*

$$||X||_{\mathcal{X}'} := \sup\left\{\int_0^1 |X(\omega)Z(\omega)|\,d\omega : ||Z||_\mathcal{X} \le 1, Z \in \mathcal{X}\right\}.$$

*The associate space is $\mathcal{X}' := \{X \in \mathcal{M} : ||X||_{\mathcal{X}'} < \infty\}$.*

If $||\cdot||_\mathcal{X}$ is furthermore an r.i. norm (satisfies R2), then the associate norm can be expressed in terms of decreasing rearrangements.

**Proposition 4.5** (Bennett & Sharpley (1988, p. 60))**.** *Let $||\cdot||_\mathcal{X}$ be an r.i. norm. Then its associate norm is:*

$$||X||_{\mathcal{X}'} = \sup\left\{\int_0^1 X^*(\omega)Z^*(\omega)\,d\omega : ||Z||_\mathcal{X} \le 1, Z \in \mathcal{X}\right\}.$$

This remarkable result asserts that for an r.i. norm, the associate relationship can be understood in terms of certain admissible quantile reweightings; recall $Z^*(\omega) = 1 - F_{|Z|}^{-1}(\omega)$.

**Example 4.6** (Bennett & Sharpley (1988, p. 275))**.** For a pair of conjugate Young functions $\Phi$ and $\Psi$, consider $\mathcal{L}^\Phi$ with the Luxemburg norm $||\cdot||_\Phi^L$. Its associate space is $\mathcal{L}^\Psi$ with the Orlicz norm $||\cdot||_\Psi$ as the associate norm.

**Example 4.7** (Bennett & Sharpley (1988, p. 10))**.** The Lebesgue spaces $\mathcal{L}^p$: let $1 \le p \le \infty$ and the norm $||\cdot||_{\mathcal{L}^p}$. Its associate space is $\mathcal{L}^q$, where $1/p + 1/q = 1$, and the associate norm is $||\cdot||_{\mathcal{L}^q}$. Note that $p = 1$ is paired with $q = \infty$.

## 4.2 Fundamental Functions of Orlicz Regrets and Risk Measures

Rearrangement invariant norms have widely varying tail sensitivity. For instance, the $\mathcal{L}^1$ norm (expectation) is tail-*in*sensitive, whereas the $\mathcal{L}^2$ norm (variance) is rather tail-sensitive. To "stratify" the space of all rearrangement invariant norms, Bennett & Sharpley (1988) and Fröhlich & Williamson (2022) emphasize the importance of the *fundamental function*, a one-dimensional function, which characterizes indeed *fundamental* properties of the norm. For instance, its derivative at the origin can be used to characterize norm equivalences.

**Definition 4.8.** *Let $\mathcal{X}$ be an r.i. space with r.i. norm $||\cdot||_\mathcal{X}$. Let $A_t \subseteq \Omega$ be an arbitrary Lebesgue measurable set with measure $P(A_t) = t$. The fundamental function $\phi_\mathcal{X} : [0, 1] \to \mathbb{R}^+$ is defined as:*

$$\phi_\mathcal{X}(t) := ||\chi_{A_t}||_\mathcal{X}.$$

Due to rearrangement invariance, the choice of $A_t$ is arbitrary. We will also speak of the fundamental function of a coherent regret (risk) measure, which due to nonnegativity of indicator functions just coincides with the fundamental function of the induced norm.

**Proposition 4.9** (Bennett & Sharpley (1988, p. 67))**.** *For any r.i. space $\mathcal{X}$, $\phi_\mathcal{X}$ satisfies:*

> *$\phi_\mathcal{X}$ is increasing.*
> *$t \mapsto \phi_\mathcal{X}(t)/t$ is decreasing.*
> *$\phi_\mathcal{X}$ is continuous except perhaps at $0$ and $\phi_\mathcal{X}(0) = 0$.*

A function $\phi_\mathcal{X}$ satisfying these properties is called *quasiconcave*. Every increasing concave function, which vanishes only at the origin, is quasiconcave, but a quasiconcave function need not be concave in general.

**Example 4.10.** The $||\cdot||_{\mathcal{L}^1}$ norm has fundamental function $\phi_{\mathcal{L}^1}(t) = t$. In contrast, the $||\cdot||_{\mathcal{L}^\infty}$ norm (supremum) has $\phi_{\mathcal{L}^\infty}(t) = \chi_{(0,1]}$, which immediately jumps to 1, hence it is not continuous at 0. It holds that any fundamental function lies pointwise in between those of $\mathcal{L}^1$ and $\mathcal{L}^\infty$ (cf. also Proposition 4.3).

We denote the fundamental function of the Orlicz regret as $\phi_{V_g}(t) = V_g(\chi_{A_t}) = ||\chi_{A_t}||_{V_g}$ and similarly, the fundamental function of the divergence risk measure as $\phi_{R_g}(t) = R_g(\chi_{A_t}) = ||\chi_{A_t}||_{R_g}$. We have that

$\phi_{R_g}(1) = 1$ due to translation equivariance of $R_g$, which implies $R_g(c) = c \; \forall c \in \mathbb{R}$ (Rockafellar & Uryasev, 2013). We begin by characterizing the fundamental function of the Orlicz regret measure. We use the fact that a left-continuous function $f$ possesses a generalized left-continuous inverse $f^{-1}(y) := \sup\{x : f(x) < y\}$.

**Proposition 4.11** (Kosmol & Müller-Wichards (2011)). *Let $f \in \mathfrak{F}$. Then:*

$$\phi_{V_g}(t) = tf^{-1}\left(\frac{1}{t}\right)$$

*is the fundamental function of the Orlicz regret measure and it is concave.*

*Proof.* The proof is in Appendix A.2.1. □

**Proposition 4.12.** *Let $f \in \mathfrak{F}$. Then $\phi_{V_g}$ and $\phi_{R_g}$ both have the following properties (denote by $\phi$ either of them):*

**FF1.** $\phi(0+) := \lim_{t \downarrow 0} \phi(t) = 0$.

**FF2.** *If $f$ is finite on $\mathbb{R}^+$, then $\phi'(0) = \lim_{t \downarrow 0} \frac{\phi(t)}{t} = \infty$. Otherwise, $\phi'(0) < \infty$.*

*Proof.* The proof is in Appendix A.2.2. □

These properties have direct consequences for norm equivalence relationships.

**Corollary 4.13.** *Let $f \in \mathfrak{F}$. Then $\mathcal{L}^\infty \subsetneq \mathcal{V}_g = \mathcal{R}_g$, where $g$ is the conjugate of $f$.*

*Proof.* As the smallest rearrangement invariant space, $\mathcal{L}^\infty \subseteq \mathcal{V}_g = \mathcal{R}_g$ holds. But it is indeed a strict subset: from (Haroske, 2006, Prop. 3.4 iii)) we know that $\mathcal{R}_g \subseteq \mathcal{L}^\infty$ holds if and only if $\lim_{t \downarrow 0} \frac{1}{\phi(t)} < \infty$. But $\lim_{t \downarrow 0} \phi(t) = 0$ and therefore $\lim_{t \downarrow 0} \frac{1}{\phi(t)} = \infty$. □

**Corollary 4.14.** *Let $f \in \mathfrak{F}$. If $f$ is finite, then $\mathcal{V}_g = \mathcal{R}_g \subsetneq \mathcal{L}^1$. In contrast, if $f$ is not finite, then $\mathcal{V}_g = \mathcal{R}_g = \mathcal{L}^1$.*

*Proof.* As the largest rearrangement invariant space, $\mathcal{V}_g = \mathcal{R}_g \subset \mathcal{L}^1$ holds. However, we have from Theorem 31 in (Fröhlich & Williamson, 2022), that all rearrangement invariant norms with $\phi'(0) < \infty$ are equivalent to $\mathcal{L}^1$. In contrast, if $\phi'(0) = \infty$, the norm cannot be equivalent to $\mathcal{L}^1$ (Fröhlich & Williamson, 2022, Corollary 32) and thus $\mathcal{R}_g \subsetneq \mathcal{L}^1$. □

These corollaries are well-known in Orlicz space theory, but it is instructive to see how they easily follow from properties of the fundamental function.

A general formula for the fundamental function $\phi_{R_g}$ of the divergence risk measure seems infeasible[8]. However, we observe that in the case of a Young divergence function, it has a simple expression.

**Proposition 4.15.** *Let $f \in \mathfrak{F}_Y$. Then the fundamental function of the divergence risk measure is*

$$\phi_{R_g}(t) = \min\left\{1, tf^{-1}\left(\frac{1}{t}\right)\right\},$$

*that is, a capped version of the fundamental function of the corresponding regret measure.*

*Proof.* The proof is in Appendix A.2.3. □

**Example 4.16.** For the Young KL divergence, where $\bar{f}_{\mathrm{KL}}(x) = (x\log(x) - x + 1) \cdot \chi_{[1,\infty)}$, the fundamental function is

$$\phi_{\mathrm{KL}}(t) = \frac{t(1/t - 1)}{W\left(\frac{(1/t-1)}{e}\right)},$$

---

[8]It is easy to write down an equation, but one which is typically difficult to solve.

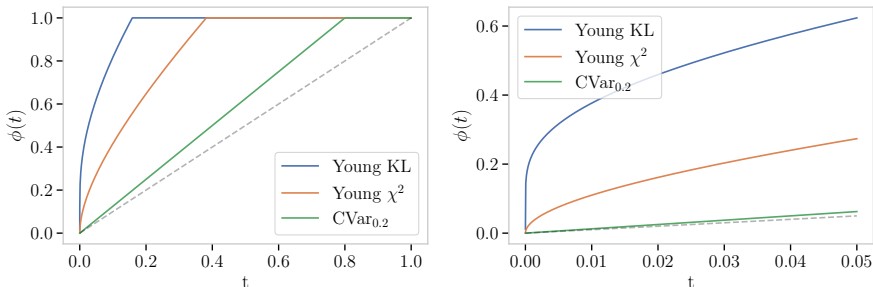

Figure 1: The fundamental functions of the Young KL, the Young $\chi^2$ and the $\text{CVar}_{\alpha=0.2}$ divergence risk measures, all with $\varepsilon = 1$. In practice, this will be an extreme choice of $\varepsilon$. The grey dotted line is the identity. Left: the full fundamental function. Right: the fundamental function for $0 \le x \le 0.05$. Here, the difference in tail behaviour is better visible: the fundamental function of the Young KL divergence risk measure is extremely steep near the origin, whereas that of $\text{CVar}_{\alpha=0.2}$ is close to the identity, which corresponds to the expectation. All of these fundamental functions are continuous at the origin.

where $W$ is the Lambert W function.

**Example 4.17.** For the Young $\chi^2$ divergence, where $\bar{f}_{\chi^2}(x) = (x-1)^2 \cdot \chi_{[1,\infty)}$, the fundamental function is

$$\phi_{\chi^2}(t) = \min\left\{1, t + \sqrt{t}\right\}.$$

**Example 4.18.** For $\text{CVar}_\alpha$, the fundamental function is $\phi_{\text{CVar}_\alpha}(t) = \min\left\{1, \frac{t}{1-\alpha}\right\}$. For $\alpha = 0$, $\phi_{\mathbb{E}}(t) = t$.

We illustrate these fundamental functions in Figure 1.

The fundamental function characterizes important aspects of an r.i. norm. Of particular interest is $\phi'(0)$, as is exemplified in Corollary 4.14. Furthermore, we have the following result which will be useful in Section 6.

**Proposition 4.19.** *Let $\phi : [0,1] \to \mathbb{R}^+$ be an increasing concave function with $\phi(0) = 0$ and $\phi(0+) = 0$. Then there exists an Orlicz norm $||\cdot||_\Psi^O$ which has $\phi$ as its fundamental function and a supercoercive Young function $\Phi$ as the conjugate of $\Psi$.*

*Proof.* The proof is in Appendix A.2.4. $\qquad\square$

### 4.3 Equivalences of Orlicz Norms

Through the framework of r.i. Banach spaces, we can associate each divergence risk measure with its natural space, on which it is finite. However, many non-identical divergence risk measures may induce the same space. Then, however, they are equivalent. Recall that two r.i. norms $||\cdot||_{\mathcal{X}_1}$ and $||\cdot||_{\mathcal{X}_2}$ are said to be equivalent if

$$\exists c, c' > 0: \quad c \cdot ||X||_{\mathcal{X}_1} \le ||X||_{\mathcal{X}_2} \le c' \cdot ||X||_{\mathcal{X}_1}, \quad \forall X \in \mathcal{M}.$$

Two r.i. norms are equivalent if and only if their respective spaces coincide (Bennett & Sharpley, 1988, p. 7). As an example, we have seen that the parameter $\varepsilon$ does not matter up to equivalence (Corollary 3.23).

Why are we interested in norm (non-)equivalences? Whether two norms are equivalent or not prima facie seems like a very coarse criterion for considering them similar. After all, the binary equivalence notion does not tell us how large the multiplicative constants are. Equivalence means that the norms are finite on the same set of functions: yet in practice, when computing the norm of an empirical loss distribution, we will never encounter infinities, since all real-world samples are bounded (from $\mathcal{L}^\infty$). In this way, unbounded tails are a useful, asymptotic idealization used to model the frequency of extreme events.

In a sense, those divergence risk measures with the same space share the same tail sensitivity; although we will later argue that sharing the same fundamental function provides a coarser and simpler way to capture

tail sensitivity. Consider what the risk assessment $R(X) = \infty$ means: the decision maker considers the tail risk in $X$ to be *unbearable*, that is, there is no finite certainty equivalent (gain), which in exchange would make the risk bearable. For example, irrespective of the choice of $\varepsilon > 0$, a decision maker who uses the KL divergence risk measure would assess $R(X) = \infty$ for a random variable $X$ which is not subexponential, (see Example 5.10) and thus prefer any subexponential random variable over it. Thus, investigating equivalence relationships between spaces reveals different attitudes towards tail risk. In general, the more tail-sensitive a risk measure, the smaller the induced space.

We briefly consider equivalence of Orlicz norms, as we require these results in what follows.

**Definition 4.20** (Rubshtein et al. (2016, p. 208)). *Let $g_1, g_2$ be Young functions. We write $g_1 \succ g_2$ when*

$$g_2(x) \le bg_1(ax), \quad \forall x \ge 0,$$

*for some $b > 0$, $a > 0$.*

We write $\sim$ when both $g_1 \succ g_2$ and $g_2 \succ g_1$ holds. We see immediately that $f \sim f_\varepsilon$ for any $\varepsilon > 0$ if $f \in \mathfrak{F}_Y$.

**Proposition 4.21** (Rubshtein et al. (2016, p. 209)). *Let $g_1, g_2$ be Young functions. The following statements are equivalent:*

  **E1.** $g_1 \succ g_2$.

  **E2.** $\mathcal{L}^{g_1} \subseteq \mathcal{L}^{g_2}$.

  **E3.** $\exists c > 0$: $\phi_{V_{g_2}}(t) \le c\phi_{V_{g_1}}(t), \quad \forall t \in [0,1]$.

  **E4.** $\exists c > 0$: $||X||_{V_{g_2}} \le c||X||_{V_{g_1}}$.

**Example 4.22.** The norm induced by the (Young) $\chi^2$ divergence risk measure is equivalent to the $\mathcal{L}^2$ norm, which has fundamental function $\phi_{\mathcal{L}^2}(t) = \sqrt{t}$. Observe that

$$\sqrt{t} \le t(1 + \sqrt{1/t}) \le 2\sqrt{t}, \quad \forall t \in (0,1]$$

and $\phi(t) = t(1 + \sqrt{1/t})$ is the fundamental function of the $\chi^2$ Orlicz regret measure. Therefore, the natural r.i. space for the $\chi^2$ divergence risk measure is that of random variables $X$ for which $|X|$ has a variance.

**Example 4.23.** $\mathrm{CVar}_\alpha$ is equivalent to the expectation for all $\alpha \in [0,1)$. As $\alpha \to 1$, it yields $\mathcal{L}^\infty$, however, and the risk measure is the essential supremum.

**Example 4.24.** Let $h$ be a Young function, which need not be divergence function, e.g. $h(x) = x^2$. Then $f(x) := (h(x) - h(1)) \cdot \chi_{[1,\infty)}$ is valid Young divergence function which is furthermore equivalent to $h$. As an example, $h(x) = x^2$ is equivalent to $f(x) = ((x-1)^2) \cdot \chi_{[1,\infty)}$, the $\chi^2$ Young divergence function.

*Proof.* Obviously $f$ is a valid Young divergence function. Consider the definition of the Luxemburg norm

$$||X||_f^L = \inf\left\{\lambda > 0 : \mathbb{E}\left(h\left[\frac{|X|}{\lambda}\right] - h(1)\right) \cdot \chi_{[1,\infty)} \le 1\right\}.$$

to see that $\mathcal{L}^h = \mathcal{L}^f$. $\qquad\qquad\qquad\qquad\qquad\qquad\qquad\qquad\qquad\qquad\qquad\qquad\qquad\qquad\qquad\square$

## 5  Tail-Specific Marcinkiewicz and Lorentz Norms

Our overarching goal is to show how r.i. norms (and hence, coherent risk measures) can be constructed which exhibit a desired tail sensitivity; thereby yielding natural spaces of "tail classes", on which the norm is finite, for instance the space of subexponential random variables. In this section we advance on this idea by showing how tail sensitivity is connected to the fundamental function and the *envelope function*. Intuitively, a desired tail sensitivity is in a very close correspondence with the fundamental function. For each thus constructed fundamental function, we consider the *smallest* and *largest* compatible coherent risk measure.

In the next section, we derive the corresponding divergence risk measure via the same approach, which is then situated in between these extremes or coincides with either of them.

In the language of the imprecise probability literature, the fundamental function specifies a *coherent upper probability* (Walley, 1991), which is the restriction of the risk measure to indicator functions. In classical precise probability theory, one can take either the expectation operator or a probability measure as the primitive notion, since applying an expectation to indicator functions yields a probability and vice versa, an expectation is uniquely defined via integration. In the more general theory of imprecise probability, this one-to-one correspondence does not hold anymore: a fundamental function can in general be extended in multiple ways to a coherent risk measure[9]. However, the fundamental function still characterizes *fundamental* properties of a norm. For the relation of imprecise probability and coherent risk measures we refer to (Vicig, 2008) and (Fröhlich & Williamson, 2022). Our approach is to specify a tail sensitivity via the *envelope function*, which is in a one-to-one relationship with the fundamental function. The specification of an envelope function is a least upper bound on the tail behaviour of the random variables, which are contained in the space. In some cases, the envelope function is itself in the space; in others it is a natural limit object, which is "just outside" the space.

Recall that for an r.i. norm, we have the associate relationship:

$$||X||_{\mathcal{X}} = \sup\left\{\int_0^1 X^*(\omega)Z^*(\omega)\, \mathrm{d}\omega : ||Z||_{\mathcal{X}'} \leq 1, Z \in \mathcal{X}'\right\}. \tag{7}$$

By specifying a set $\mathcal{Z} := \{Z^* : ||Z||_{\mathcal{X}'} \leq 1\}$, we have fully specified the norm in terms of which quantile reweightings are admissible. Conversely, any set of nonnegative and decreasing $Z^*$ leads to a legitimate r.i. norm (see e.g. Fröhlich & Williamson, 2022, Section 4.7). Thus, an r.i. norm is fully identified by its set of nonnegative and decreasing functions $Z^*$ on $[0, 1]$. However, the fundamental function, which is a far simpler object than a whole set of $Z^*$, already stratifies the space of r.i. norms in a useful way regarding tail sensitivity. To see this more clearly, we need to introduce the concept of *envelope functions* (Haroske, 2006).

**Definition 5.1** (Haroske (2006)). *Let $\mathcal{X}$ be any r.i. space. Define $E_{\mathcal{X}} : (0, 1] \to \mathbb{R}^+ \cup \{\infty\}$ by:*

$$E_{\mathcal{X}}(t) := \sup\{X^*(t) : ||X||_{\mathcal{X}} \leq 1\}.$$

*$E_{\mathcal{X}}$ is right-continuous and decreasing, hence $E_{\mathcal{X}}^* = E_{\mathcal{X}}$.*

Recall that since $X^*(t) = F_{|X|}^{-1}(1 - t)$, the tail behaviour is encoded in the behaviour as $t \downarrow 0$.

**Proposition 5.2** (Haroske (2006, p. 53)). *The envelope function is in a one-to-one relation with the fundamental function via*

$$E_{\mathcal{X}}(t) = \frac{1}{\phi_{\mathcal{X}}(t)}, \quad \forall t \in (0, 1].$$

Since multiplicative factors do not matter up to norm equivalence, we shall be concerned with equivalence classes of envelope functions and fundamental functions. That is, similar to E3, we call two envelope functions equivalent if

$$\exists c, c' > 0 : \quad c \cdot E_{\mathcal{X}_1}(t) \leq E_{\mathcal{X}_2}(t) \leq c' \cdot E_{\mathcal{X}_1}(t), \quad \forall t \in (0, 1].$$

and likewise for fundamental functions. Intuitively, the envelope function constrains the tail behaviour of random variables in the space. It is however more instructive to change perspective and consider the associate space. We need the following technical result.

**Proposition 5.3** (Rubshtein et al. (2016, p. 135)). *Let $\mathcal{X}$ be an r.i. space with fundamental function $\phi_{\mathcal{X}}$. Then the fundamental function of the associate space $\mathcal{X}'$ is $\phi_{\mathcal{X}'}(t) = \frac{t}{\phi_{\mathcal{X}}(t)}$.*

We may refer to $\phi_{\mathcal{X}'}$ as the *associate* fundamental function. Consequently, the envelope function in the associate space is:

$$E_{\mathcal{X}'}(t) = \frac{1}{\phi_{\mathcal{X}'}(t)} = \frac{\phi_{\mathcal{X}}(t)}{t} = \sup\{Z^*(t) : ||Z||_{\mathcal{X}'} \leq 1\}.$$

---

[9]In expectational cases, there is only a single extension, see Fröhlich & Williamson (2022, Section 4).

Thus, the envelope function constrains the tail behaviour of the dual variables $Z^*$ in (7) and it is the least such upper bound, the supremum. All in all, we have one-to-one relations between $\phi_{\mathcal{X}}, \phi_{\mathcal{X}'}, E_{\mathcal{X}}$ and $E_{\mathcal{X}'}$; which object is easiest to specify depends on motivation and perspective. In what follows, we often write simply $\phi$ instead of $\phi_{\mathcal{X}}$ for the primal fundamental function to prevent excessive subscripts. Essentially, the choice of either of these objects is a choice of a coherent upper probability underlying an r.i. norm. Also observe that

$$\phi'(0) = \lim_{t \downarrow 0} \frac{\phi_{\mathcal{X}}(t)}{t} = \lim_{t \downarrow 0} E_{\mathcal{X}'}(t).$$

The derivative of $\phi$ at the origin has been used to characterize norm equivalences (see for instance Corollaries 4.13, 4.14 and (Fröhlich & Williamson, 2022, Section 4.9)). Via the envelope function perspective we see that it corresponds to a constraint on the tail behaviour of the dual variables.

The envelope function naturally suggests the following approach to the construction of tail-sensitive r.i. norms: specify a constraint on the tail behaviour of dual or primal variables by specifying an envelope function ($E_{\mathcal{X}}$ or $E_{\mathcal{X}'}$) as the decreasing rearrangement of a reference distribution (e.g. as the decreasing rearrangement of an arbitrary random variable with exponential distribution); then construct an r.i. norm with the corresponding fundamental function. For example, when constraining the quantiles of primal variables $X$ to be bounded from above by the quantiles of an exponential distribution, we will arrive at the class of subexponential variables and the KL divergence risk measure.

For any concave fundamental function, we may consider the smallest and the largest r.i. norm, with the Orlicz norm in between. The largest norm is known as the *Lorentz norm* and the smallest is the *Marcinkiewicz norm.*

### 5.1 The Marcinkiewicz Norm

Assume we have specified a legitimate $E_{\mathcal{X}}$ and thus a quasiconcave $\phi(t) = 1/E_{\mathcal{X}}(t)$, since a fundamental functions must always be quasiconcave (Proposition 4.9). A first (unsuccessful) attempt to define a norm, which captures this constrained tail behaviour, is to define the following.

**Definition 5.4** (Kreĭn et al. (1982)). *Let $\phi(t) = 1/E_{\mathcal{X}}(t)$ be a quasiconcave fundamental function. We define the Marcinkiewicz quasi-norm as:*

$$|X|_{M_\phi} := \sup_{0 < t \leq 1} \phi(t) X^*(t).$$

This naturally expresses the constraint on the quantile behaviour of $X$, compared to the reference $E_{\mathcal{X}}$. Intuitively:

$$|X|_{M_\phi} < \infty \iff \exists c : X^*(t) \leq c \cdot E_{\mathcal{X}}(t), \quad t \in (0, 1].$$

Here, $E_{\mathcal{X}}$ is the decreasing rearrangement of our reference distribution, e.g. exponential. Unfortunately, $|\cdot|_{M_\phi}$ is not a norm, as it does not satisfy the triangle inequality. However, we can define the closely related Marcinkiewicz norm. By integrating the decreasing rearrangement, we obtain the *maximal function* (Bennett & Sharpley, 1988, pp. 52-53):

$$X^{**}(t) := \frac{1}{t} \int_0^t X^*(\omega) \, d\omega, \quad t > 0.$$

Since $X^*(t) = F_{|X|}^{-1}(1 - t)$, we find that $X^{**}(t) = \mathrm{CVar}_{1-t}(|X|)$; thus $X^{**}(0)$ is the essential supremum of $|X|$ and $X^{**}(1) = \mathbb{E}[|X|]$.

**Definition 5.5.** *Given any quasiconcave fundamental function $\phi$, the Marcinkiewicz norm $||\cdot||_{M_\phi}$ is defined as:*

$$\begin{aligned} ||X||_{M_\phi} &:= \sup_{0 < t \leq 1} \{\phi(t) X^{**}(t)\} \\ &= \sup_{0 < t \leq 1} \{\phi(t) \mathrm{CVar}_{1-t}(|X|)\}. \end{aligned}$$

*We denote the Marcinkiewicz space as $M_\phi := \{X \in \mathcal{M} : ||X||_{M_\phi} < \infty\}$.*

This norm also has fundamental function $\phi$. Intuitively, the Marcinkiewicz norm measures the growth of the CVar's, but weighted with the fundamental function. In the risk measure community, the Marcinkiewicz norm has been unknowingly re-discovered by Pichler (2013, Section 5). The difference between $|\cdot|_{M_\phi}$ and $||\cdot||_{M_\phi}$ is the use of the decreasing rearrangement (backwards quantiles) vs. the maximal function (backwards CVar's). Subadditivity of the maximal function yields a legitimate norm in the latter case. However, the quasi-norm and the norm are in fact equivalent in many cases of interest. Recall that $E_\mathcal{X} = E_\mathcal{X}^*$, since $E_\mathcal{X}$ is decreasing, and the maximal function is $E_\mathcal{X}^{**}$.

**Proposition 5.6** (Kreĭn et al. (1982)). *Let $\phi$ be quasiconcave. The following are equivalent:*

> *M1.* $\forall s > 1$: $\sup_{0 < t \le 1} \frac{\phi_{\mathcal{X}'}(t/s)}{\phi_{\mathcal{X}'}(t)} < 1$.
>
> *M2.* $|\cdot|_{M_\phi}$ and $||\cdot||_{M_\phi}$ are equivalent: $\exists K > 0 : |X|_{M_\phi} \le ||X||_{M_\phi} \le K \cdot |X|_{M_\phi}$.
>
> *M3.* $||\phi_{\mathcal{X}'}(t)/t||_{M_\phi} = ||E_\mathcal{X}||_{M_\phi} < \infty$.
>
> *M4.* $E_\mathcal{X}$ and $E_\mathcal{X}^{**}$ are equivalent: $\exists K > 0 : E_\mathcal{X}(t) \le E_\mathcal{X}^{**}(t) \le K \cdot E_\mathcal{X}(t), \quad \forall t \in (0, 1]$.
>
> *M5.* Let $\phi_{\mathrm{CVar}}(t) = 1/E_\mathcal{X}^{**}(t)$. $||\cdot||_{M_\phi}$ and $||\cdot||_{M_{\phi_{\mathrm{CVar}}}}$ are equivalent.

*Proof.* The proof is in Appendix A.3.1. $\qquad\square$

A necessary, but not sufficient criterion for these conditions to hold is that $\phi_{\mathcal{X}'}$ be strictly increasing. Intuitively, the conditions amount to requiring that the CVar's of the envelope function do not grow too rapidly.

The Marcinkiewicz norm naturally expresses a tail sensitivity based on CVar's. If any of the conditions in Proposition 5.6 are fulfilled, it does not matter up to equivalence whether the approach to specify a tail sensitivity is based on CVar's or quantiles of a reference distribution: the same space results.

To summarize, our approach is as follows: specify an envelope function $E_\mathcal{X}$ as the decreasing rearrangement of a reference distribution with a desired tail behaviour. This amounts to a choice of a coherent upper probability in form of the fundamental function $\phi_\mathcal{X}$. Then consider the corresponding Marcinkiewicz quasi-norm, which is typically equivalent to the Marcinkiewicz norm (we later demonstrate that the equivalence "often" holds). In any case, the Marcinkiewicz norm occupies a special status among all r.i. norms, which are compatible with a specified fundamental function. To state the result, we need to define its associate norm, the *Lorentz norm*.

## 5.2 The Lorentz Norm

**Definition 5.7.** *Given any concave fundamental function $\phi$, the Lorentz norm $||\cdot||_{\Lambda_\phi}$ is defined as:*

$$||X||_{\Lambda_\phi} := \int_0^1 X^*(\omega)\, \mathrm{d}\phi(\omega)$$

$$= X^*(0)\phi(0+) + \int_0^1 X^*(\omega)\phi'(\omega)\, \mathrm{d}\omega.$$

*We denote the Lorentz space as $\Lambda_\phi := \{X \in \mathcal{M} : ||X||_{\Lambda_\phi} < \infty\}$.*

The Lorentz norm is fully specified by its concave fundamental function, which indeed is $\phi$. It has an intuitive interpretation: the derivative of the fundamental function is used to reweight the decreasing rearrangement. Since $\phi$ is concave, $\phi'$ is decreasing and thus more weight is put on worse outcomes. Furthermore, if $\phi$ is not continuous at the origin, i.e. $\phi(0+) > 0$, then the essential supremum $X^*(0)$ receives the fixed weight $\phi(0+)$. We remark that in finance and insurance, the Lorentz norm appears as the norm induced by a *spectral risk measure* (Acerbi, 2002), also called a *distortion risk measure* (Wang, 2000), which plays a major role in financial risk assessment.

In fact, the Marcinkiewicz norm $||\cdot||_{M_{\phi_{\mathcal{X}'}}}$, where $\phi_{\mathcal{X}'}(t) = t/\phi(t)$, is the associate norm to the Lorentz norm $\Lambda_\phi$. The converse associate relation also holds, but with a technical subtlety regarding (quasi)concavity (see Section 5.3).

**Example 5.8.** $\mathcal{L}^1$ is a Lorentz space with $\phi_{\mathcal{L}^1}(t) = t$. Diametrically opposed, $\mathcal{L}^\infty$ is also a Lorentz space with $\phi_{\mathcal{L}^\infty}(t) = \chi_{(0,1]}$. In the case of $\mathcal{L}^1$ and $\mathcal{L}^\infty$, the Marcinkiewicz, Orlicz and Lorentz norm are in fact identical (Fröhlich & Williamson, 2022, Section 4). In contrast, for instance, $\mathcal{L}^2$ is *not* a Lorentz space, but an Orlicz space.

Given a concave fundamental function, the Marcinkiewicz norm yields the smallest (most optimistic) and the Lorentz norm the largest (most pessimistic) risk assessment.

**Theorem 5.9** (Bennett & Sharpley (1988, p. 72)). *Let $\mathcal{X}$ be an r.i. space with concave fundamental function $\phi$. Then the following embedding holds:*

$$\Lambda_\phi \subseteq \mathcal{X} \subseteq M_\phi$$

*and the norms are in the relationship:*

$$||X||_{M_\phi} \leq ||X||_{\mathcal{X}} \leq ||X||_{\Lambda_\phi} \quad \forall X \in \mathcal{M}.$$

We offer the following intuition for the result: both the Marcinkiewicz and the Lorentz norm are in a special, parsimonious one-to-one correspondence to the fundamental function $\phi$. Let $\phi$ be concave and continuous at the origin. Then the Lorentz norm is defined using only the single dual variable $\phi'$, which controls the reweighting (and it is the *only* r.i. norm for which a single dual variable suffices). As to the Marcinkiewicz norm, we have the extremal assessment $||\phi'||_{M_{\phi_{\mathcal{X}'}}} = 1$. In this sense, the Marcinkiewicz norm is exactly tuned to the associate fundamental function. Thus, the Marcinkiewicz and the Lorentz norm demarcate the space of r.i. norms, which are compatible with a specified fundamental function.

We build intuition for the use of the Marcinkiewicz and Lorentz norms with the following example of the spaces $L_{exp}$ and LlogL. In this example we consider the class of *subexponential* random variables and its "dual" tail class, the random variables with finite differential entropy.

**Example 5.10** (Bennett & Sharpley (1988, pp. 243-254)). $L_{exp}$, LlogL: Consider the envelope function that expresses subexponential tails. The decreasing rearrangement of an exponential distribution, which is shifted by 1, is $Y^*(t) = 1 - \log(t)$; the reason for the shift will become clear later. We take this to be the envelope function $E_{\mathcal{X}}(t) = 1 - \log(t)$ and define the Marcinkiewicz quasi-norm:

$$|X|_{L_{exp}} := \sup_{0 < t \leq 1} \frac{1}{1 - \log(t)} X^*(t).$$

Note that $\phi(t) = 1/(1 - \log(t))$ is quasiconcave. In this case, it is easy to verify that condition M1 in Proposition 5.6 holds, and therefore we may equivalently consider the Marcinkiewicz norm:

$$||X||_{L_{exp}} := \sup_{0 < t \leq 1} \frac{1}{1 - \log(t)} X^{**}(t).$$

Observe that the CVar's of an exponential distribution (not shifted) are in fact

$$CVar_\alpha(|X|) = 1 - \log(1 - \alpha) \quad \forall \alpha \in [0, 1).$$

We call a random variable *subexponential* if $||X||_{L_{exp}} < \infty$ and denote the space $L_{exp}$. It turns out, as we observe in the next section, that an equivalent approach to capture this tail class is via an Orlicz norm, which then yields the following definition: a random variable $X$ is subexponential if $\exists \lambda > 0$ such that:

$$\int_\Omega \exp\left(\lambda |X(\omega)|\right) \, d\omega = \mathbb{E}[\exp(\lambda |X|)] < \infty, \tag{8}$$

that is, if the moment-generating function of $|X|$ is finite for some $\lambda > 0$. This is a typical textbook definition of subexponential tails (Vershynin, 2018). In this case, we have equivalence of the two definitions of $L_{exp}$.

We will later observe that this is due to the fact that the Marcinkiewicz space here coincides with the Orlicz space with the same fundamental function.

The space $L_{exp}$ is in fact the associate space of the Lorentz space LlogL, defined by:

$$||X||_{LlogL} := \int_0^1 X^*(\omega) \, d\phi_{LlogL}(\omega), \quad \phi_{LlogL}(t) := t + t\log(1/t).$$

This is a distinguished space: the space LlogL consists of those random variables whose CVar's are integrable:

$$||X||_{LlogL} < \infty \iff \int_0^1 CVar_\alpha(|X|) \, d\alpha < \infty,$$

which is a very weak condition, hence the space is very "large". In fact, we have the remarkable embedding situation:

$$\mathcal{L}^\infty \subsetneq L_{exp} \subsetneq \mathcal{L}^p \subsetneq LlogL \subsetneq \mathcal{L}^1, \quad \forall p \in (1, \infty).$$

Thus, $L_{exp}$ is smaller than any $\mathcal{L}^p$ space, whereas its associate LlogL is larger than any $\mathcal{L}^p$ space, $p \in (1, \infty)$. This behaviour can be intuited from the fundamental functions as $x \downarrow 0$: the fundamental function of LlogL is very moderate, while that of $L_{exp}$ is extremely steep near the origin (see Appendix A.3.2).

Furthermore, LlogL consists of those random variables with finite differential entropy[10]:

$$||X||_{LlogL} < \infty \iff \int_\Omega |X(\omega)| \log(|X(\omega)|) \, dP(\omega) < \infty,$$

which, if $X = \frac{dQ}{dP}$ is a legitimate density, is just the KL divergence of $Q$ to the base measure $P$. Hence we observe that the KL divergence risk measure is finite on $L_{exp}$ (a very "small" space) and the feasible dual variables are from LlogL (a very "large" space). This means that almost arbitrary tail reweightings are allowed, which entails an extremely tail-sensitive behaviour of the KL divergence risk measure.

We remark that, confusingly, according to some authors a subexponential random variable is one which has *heavier* tails than an exponential random variable. Such terminology appears counter-intuitive to us and conflicts with the definition of a sub-Gaussian random variable.

### 5.3   Quasiconcavity and Concavity

The question remains whether any random variable $Y$ is a valid reference distribution as an envelope function, with the identification $\phi(t) = 1/Y^*(t)$ for $0 < t \le 1$. We always set $\phi(0) = 0$. Then $\phi$ is continuous at 0 if and only if $\lim_{t\downarrow 0} Y^*(t)$ is unbounded. Throughout, we shall only consider such $Y^*$ (with the instructive exception of Example 6.5). The critical issue is that $\phi$ must be quasiconcave. For this, we need that $\phi$ is nonnegative and increasing, which is satisfied for any $Y$, as $Y^*$ is nonnegative and decreasing. Also, we require that $t \mapsto \phi(t)/t$ be decreasing:

$$\frac{\phi(t)}{t} = \frac{1}{Y^*(t) \cdot t} = \frac{1}{t/\phi(t)} = \frac{1}{\phi_{\mathcal{X}'}(t)},$$

which is decreasing if and only if $t \mapsto Y^*(t) \cdot t$ is increasing. Hence, any nonzero $Y$ which ensures that $t \mapsto Y^*(t) \cdot t$ is increasing yields a legitimate envelope function. Observe that since $\phi_{\mathcal{X}'}(t) = Y^*(t) \cdot t$, this is equivalent to requiring that $\phi_{\mathcal{X}'}$ be quasiconcave (note that $t \mapsto \phi_{\mathcal{X}'}(t)/t = Y^*(t)$ is decreasing). Therefore it is also equivalent to requiring that the envelope function of the associate space $\mathcal{X}'$ is decreasing, which of course is necessary for any legitimate r.i. norm. We make an observation, which relates to the embedding theorem.

**Proposition 5.11.** *Assume $Y^*$ is differentiable on $(0, 1)$. Then for $t \mapsto \phi_{\mathcal{X}'}(t) = Y^*(t) \cdot t$ to be increasing it is necessary that $Y^*(t) \le Y^*(1) \cdot \frac{1}{t}$.*

---

[10]Observe the content in (Bennett & Sharpley, 1988, p. 243-245) and combine it with the equivalence of the Orlicz functions $(x\log(x))\chi_{[1,\infty)}$ and $x\log(x) - (x-1)$.

*Proof.* The proof is in Appendix A.3.3. $\qquad\square$

Recall that $\mathcal{L}^1$ is the largest of all r.i. spaces and has envelope function $E_{\mathcal{L}^1}(t) = \frac{1}{t}$. Since the constraint $Y^*(1) = 1$ (equivalently, $\phi(1) = 1$) can be imposed without loss of generality, we see that unsurprisingly, the envelope function of $\mathcal{L}^1$ is the largest of all legitimate envelope functions.

Working with the maximal function is even simpler: by setting $\phi_{\mathrm{CVar}}(t) = 1/Y^{**}(t)$ for any nonnegative and nonzero $Y$, the resulting $\phi$ is guaranteed to be quasiconcave, since $t \mapsto Y^{**}(t) \cdot t = \int_0^t Y^*(\omega)\, d\omega$ is increasing.

**Example 5.12.** *Taking the exponential distribution, $Y^*(t) = -\log(t)$, fails since $t \mapsto -\log(t) \cdot t$ is not increasing on $(0, 1]$; it violates the condition in Proposition 5.11. However, when shifting the distribution to have minimum value 1, i.e. $Y^*(t) = 1 - \log(t)$, we get $t \mapsto (1 - \log(t)) \cdot t$, which is increasing on $(0, 1]$. Of course, such a shift is insignificant for tail behaviour.*

With regard to the quasiconcavity vs. concavity distinction, we remark that it is not of essential importance. For instance, the Lorentz norm requires a concave $\phi$ in its definition. In this case, one can use the least concave majorant $\tilde{\phi}$ of $\phi$, since a quasiconcave function $\phi$ is always equivalent to its least concave majorant $\tilde{\phi}$: $\frac{1}{2}\tilde{\phi} \le \phi \le \tilde{\phi}$ (Rubshtein et al., 2016, p. 131). In fact, any r.i. space with a quasiconcave fundamental function can be equivalently renormed to have its least concave majorant as the fundamental function (Bennett & Sharpley, 1988, p. 71), (Pick et al., 2013, p. 268), so the quasiconcave–concave distinction is not essential.

Finally, we note that even if $\phi$ is concave, it may be that $\phi_{\mathcal{X}'}$ is only quasi-concave. The associate relationship of the Marcinkiewicz and Lorentz space is preserved using the least concave majorant then (Rubshtein et al., 2016, p. 147).

## 6 Tail-Specific Divergence Risk Measures

In the previous section, we have constructed Marcinkiewicz and Lorentz norms, which capture a desired tail behaviour. In this section, we proceed with the construction of tail-specific Orlicz norms, which yield corresponding divergence risk measures. We focus on the construction of Orlicz norms for a specific tail sensitivity; from this, equivalent Orlicz regret and divergence risk measures can easily be derived (see Example 4.24 and recall Corollary 3.21).

**Remark 6.1** ( Rubshtein et al. (2016, p. 162, p. 209)). *Let $\phi$ and $\tilde{\phi}$ be equivalent concave fundamental functions. Then $\Lambda_\phi$ and $\Lambda_{\tilde{\phi}}$ are equivalent; $M_\phi$ and $M_{\tilde{\phi}}$ are equivalent; and the Orlicz norms with fundamental functions $\phi$ and $\tilde{\phi}$ are equivalent.*

We follow the same logic as in the previous section, beginning with the specification of a fundamental function or, via its one-to-one correspondence, of an envelope function. In this section, we denote by $\Phi$ and $\Psi$ a conjugate pair of Young functions, which need not be divergence functions. Intuitively, they play the role of $f$ and $g$, respectively. The corresponding fundamental functions of the Orlicz spaces are $\phi_\Phi^L(t)$ and $\phi_\Psi(t) = t/\phi_\Phi(t)$, where we use the $L$ to emphasize that it corresponds to the Luxemburg norm in $\mathcal{L}^\Phi$. Then:

$$||X||_\Psi^Q = \sup\left\{ \int_0^1 X^*(\omega) Z^*(\omega)\, d\omega : Z \in \mathcal{L}^\Phi, Z \ge 0, \mathbb{E}[\Phi(Z)] \le 1 \right\} \quad \forall X \in \mathcal{L}^\Psi.$$

In fact, the Orlicz norm is the associate norm to the Luxemburg norm, that is:

$$||X||_\Psi^Q = \sup\left\{ \int_0^1 X^*(\omega) Z^*(\omega)\, d\omega : Z \in \mathcal{L}^\Phi, Z \ge 0, ||Z||_\Phi^L \le 1 \right\} \quad \forall X \in \mathcal{L}^\Psi. \tag{9}$$

To achieve a desired tail sensitivity, we aim to constrain the tail behaviour of dual variables $Z^*$ in an Orlicz norm (a divergence risk measure, respectively). For example, for the LlogL divergence risk measure, we allow dual variables to be from $\mathrm{L}_{\exp}$. Conversely, for the $\mathrm{L}_{\exp}$ (KL) divergence risk measure, we allow dual variables from LlogL, which merely need to have finite differential entropy. We denote the envelope function in $\mathcal{L}^\Phi$ of the Luxemburg norm, indexed by the Young function $\Phi$, as:

$$E_\Phi^L(t) := \sup\{ Z^*(t) : ||Z||_\Phi^L \le 1 \}.$$

By specifying this envelope function based on the decreasing rearrangement of some reference distribution $Y^*$, setting $E_\Phi^L = Y^*$, we have the least upper bound on the tail behaviour of the dual variables in (9) in terms of $Y^*$ and thereby delimit possible quantile reweightings. The more extreme our envelope function, the higher the reweighting can be. We assume that the envelope function is finite for all $t \in (0,1]$ and unbounded for $t = 0$ (otherwise there would be no tails).

**Proposition 6.2.** *Choose $E_\Phi^L : (0,1] \to \mathbb{R}^+$ as the decreasing rearrangement of some reference distribution, for which $E_\Phi^L(0+) = \infty$, $\phi_\Psi(t) := E_\Phi^L(t) \cdot t$ is increasing and concave on $(0,1]$ and which satisfies $\phi_\Psi(0+) = 0$. We set $\phi_\Psi(0) = 0$. Then there exists an Orlicz norm $||\cdot||_\Psi^Q$ which has fundamental function $\phi_\Psi$. The conjugate Young function of $\Psi$, which is $\Phi$, induces $\mathcal{L}^\Phi$ with envelope function $E_\Phi^L$. Furthermore, $\Phi$ is finite and supercoercive and therefore $\Psi$ is finite.*

*Proof.* Let $\phi_\Psi$ be the fundamental function of $||\cdot||_\Psi^Q$, that is, $\phi_\Psi(t) = t\Phi^{-1}(1/t)$. Denote by $\phi_\Phi^L(t) = t/\phi_\Psi(t)$ the fundamental function of the Luxemburg norm with Young function $\Phi$. We know that

$$E_\Phi^L(t) = \frac{1}{\phi_\Phi^L(t)} = \frac{1}{t/\phi_\Psi(t)} = \frac{\phi_\Psi(t)}{t},$$

by Proposition 5.1 and Proposition 5.3; recall that the Orlicz norm $||\cdot||_\Psi^Q$ is in an associate relationship to $||\cdot||_\Phi^L$. Therefore:

$$\phi_\Psi(t) = tE_\Phi^L(t) = t\Phi^{-1}(1/t), \quad \forall t \in (0,1]$$
$$\Leftrightarrow E_\Phi^L(1/x) = \Phi^{-1}(x), \quad \forall x \in [1, \infty).$$

By specifying a legitimate $E_\Phi^L(t)$, we thus obtain $\Phi^{-1}(x)$ for $x \geq \Phi^{-1}(1)$. Of course, the behaviour of $\Phi$ on $0 \leq x < \Phi^{-1}(1)$ is immaterial for tail sensitivity. As we have seen before, Orlicz norms have *concave* fundamental functions (Proposition 4.11), hence we have to ensure that $\phi_\Psi(t) = E_\Phi^L(t) \cdot t$ is a valid concave fundamental function. Given that we specified some $E_\Phi^L$ which yields a concave $\phi(t) = E_\Phi^L(t) \cdot t$ and thereby specifies the behaviour of $f$ on $x \geq \Phi^{-1}(1)$, can we always find a compatible $\Phi$, so that $\Phi$ is indeed a legitimate Young function? The answer is affirmative and a constructive approach is in the proof of Proposition 4.19. For the supercoercivity of $\Phi$, see the proof of FF1 in Proposition 4.12. For the finiteness of $\Phi$, observe that $E_\Phi^L(0+) = \infty \implies \phi_\Psi'(0) = \infty$ and then see FF2 in Proposition 4.12. $\qquad\square$

Of course, the primal approach is also feasible, where the envelope of the Orlicz norm in $\mathcal{L}^\Psi$ is specified as $E_\Psi^Q(t) := \sup\{Z^*(t) : ||Z||_\Psi^Q \leq 1\} = 1/\phi_\Psi(t)$, hence we then demand that $\phi_\Psi(t)$ is concave and $\phi_\Psi(0+) = 0$. If we at least have a quasiconcave $\phi$, that is, if $t \mapsto \phi_\Psi(t) := E_\Phi^L(t) \cdot t$ is increasing, we may take its equivalent least concave majorant (see Section 5.3). Thus, to achieve a tail sensitivity we merely need that $t \mapsto \phi_\Psi(t) := E_\Phi^L(t) \cdot t$ is increasing, since $t \mapsto \phi_\Psi(t)/t = E_\Phi^L(t) = Y^*(t)$ is decreasing for any $Y$.

We have seen that specifying a least upper bound on the tail behaviour of primal or dual variables yields a corresponding Orlicz space. Assume that we have therefore obtained a Young function $\Phi$ and an equivalent Young divergence function $f \in \mathfrak{F}$ via Example 4.24. To further motivate our approach, observe that if $Z \notin \mathcal{L}^\Phi$, no finite $\varepsilon$ will suffice for $Z$ to satisfy the $f$-divergence constraint in the envelope representation of the divergence risk measure. The converse direction is more subtle due to the $\Delta_2$ condition (see Definition A.6). Thus, our approach is about the *structure* of the divergence ball, essentially given by the tail constraint, not its particular size, which of course is also significant in practical implementations.

**Example 6.3.** $\mathcal{L}^\Psi = $ LlogL: we choose the dual envelope $E_\Phi^L(t)$ to be the decreasing rearrangement of a random variable with exponential distribution, shifted by 1, that is, $E_\Phi^L(t) = 1 - \log(t)$. As desired, $t \mapsto \phi_\Psi(t) = E_\Phi^L(t) \cdot t$ is increasing and concave and $\phi_\Psi(0+) = 0$. Then $E_\Phi^L(1/x) = 1 - \log(1/x) = \Phi^{-1}(x)$, $x \geq 1$. This is satisfied by the Young function

$$\Phi(x) = \begin{cases} x & 0 \leq x < 1 \\ \exp(x-1) & x \geq 1, \end{cases}$$

which is in fact equivalent (in the sense of Section 4.3) to the function $\Phi(x) = \exp(x) - 1$. Therefore we obtain the LlogL Orlicz norm as $||\cdot||_\Psi^Q$.

**Example 6.4.** $\mathcal{L}^\Psi = L_{\exp}$: assume that we want to construct an Orlicz norm where the primal variables $X$ are constrained to be subexponential. This can simply be obtained from the previous example by duality. To provide more intuition, we derive it in another way. When choosing the primal envelope function as $\phi_\Psi(t) = 1/E_\Psi^O(t) = 1/(1 - \log(t))$ based on a shifted exponential distribution, we face the problem that this is only quasiconcave. However, we have the equivalence to the concave function $t \mapsto \frac{1}{\log(1/t+1)}$:

$$\frac{1}{1 - \log(t)} \leq \frac{1}{\log(1/t + 1)} \leq 2 \cdot \frac{1}{1 - \log(t)}, \quad \forall t \in (0, 1].$$

Setting $1/E_\Psi^O(t) = \phi_\Psi(t) = t\Phi^{-1}(1/t)$ yields an unwieldy form for $f$ involving the Lambert W function instead of $\Phi(x) = x\log(x) - (x - 1)$, albeit equivalent. Instead, we consider the Luxemburg norm $\|\cdot\|_\Psi^L$ with envelope function $E_\Psi^L(t) = 1/\phi_\Psi^L(t)$ and choose $E_\Psi^L(t) = \log(1/t + 1)$, where the fundamental function is (Rubshtein et al., 2016, p. 177):

$$\phi_\Psi^L(t) = \frac{1}{\Psi^{-1}(1/t)} \implies \Psi^{-1}(1/t) = \log(1/t + 1) = E_\Psi^L(t), \quad \forall t \in (0, 1].$$

Thus we find that $\Psi(y) = \exp(y) - 1$ satisfies the equation. Recall that the Luxemburg and Orlicz norms with the same Young function are equivalent. It follows:

$$\|X\|_\Psi^O < \infty \iff \exists \lambda > 0 : \mathbb{E}[\exp(\lambda|X|)] - 1 < \infty,$$

which is a standard definition of a subexponential random variable (Vershynin, 2018), cf. (8).

**Example 6.5.** $\mathcal{L}^\Psi = \mathcal{L}^1$; Expectation and $\mathrm{CVar}_\alpha$: in contrast to the assumption of Proposition 6.2 that $E_\Phi^L(0+) = \infty$, assume we want to allow only bounded quantile reweightings, i.e. dual variables from $\mathcal{L}^\infty$. For instance, let $E_\Phi^L(t) = 1$, $t \in (0, 1]$, so that all quantiles of the dual variables are bounded from above by 1. Therefore $1 = \Phi^{-1}(1/t)$, $\forall t \in (0, 1]$, whence it follows up to equivalence that

$$\Phi(x) := \begin{cases} 0 & 0 \leq x \leq 1 \\ \infty & x > 1, \end{cases}$$

which is also a divergence function. Let $g(x) := \Psi(x)$ for $x \geq 0$ and $g(x) := 0$ for $x < 0$, which is the convex conjugate of $\Phi$. In fact, $g(x) = \max(0, x)$. Then $\|X\|_{V_g} = \mathbb{E}[|X|]$ and $V_g(X) = \mathbb{E}[X^+]$. This is nothing more but the $\mathcal{L}^1 - \mathcal{L}^\infty$ associate relation: the function $\Phi$ is the Young function whose Orlicz space is $\mathcal{L}^\infty$, thus its associate space is $\mathcal{L}^1$.

Similarly, when allowing reweightings up to $1/(1 - \alpha)$, i.e. $E_\Phi^L(t) = 1/(1 - \alpha)$, we have $V_g(X) = \frac{1}{1-\alpha}\mathbb{E}[X^+]$ and $R_g(X) = \mathrm{CVar}_\alpha(X)$ via the infimal convolution in Section 3.5. This does not change the primal space, which is still $\mathcal{L}^1$, whose associate space is $\mathcal{L}^\infty$. The situation is clarified by the envelope representation

$$\mathrm{CVar}_\alpha(X) = \sup\left\{\int_0^1 X^*(\omega)Z^*(\omega)\,\mathrm{d}\omega : \mathbb{E}[Z] = 1, 0 \leq Z \leq 1/(1 - \alpha)\right\}.$$

**Example 6.6.** Pareto, $\chi^2$: The decreasing rearrangement of a Pareto distribution with shape parameter $p$ (typically denoted $\alpha$) is $Y^*(t) = t^{-\frac{1}{p}}$. Set the dual envelope function $E_\Phi^L(t) = t^{-\frac{1}{p}}$. Then $\Phi(x) = x^p$. For instance, $p = 2$ yields the $\chi^2$ divergence risk measure (up to equivalence, see Example 4.22). Of course, $L^\Phi = \mathcal{L}^p$ and $L^\Psi$ is then just $\mathcal{L}^q$, where $q$ is the dual exponent. Since 2 is self dual, that is, $1/2 + 1/2 = 1$, we have that the $\chi^2$ risk measure is finite on $\mathcal{L}^2$. The same procedure naturally works for any $1 < p \leq \infty$, as $t \mapsto t^{-\frac{1}{p}} \cdot t$ is increasing, giving the space $\mathcal{L}^p$.

An interesting case is $p = 1$: Then $E_\Phi^L(t) = 1/t$, but $t \mapsto E_\Phi^L(t) \cdot t$ is *not* continuous at 0. Then $\Phi^{-1}(x) = x$ and $\Phi(x) = x$, which is not supercoercive. In fact, $\Phi(x) = x$ is the Young function whose Orlicz norm is the expectation $\mathbb{E}[|\cdot|]$. By duality, we would obtain the essential supremum as $\|\cdot\|_\Psi^O$, where

$$\Psi(x) = \begin{cases} 0 & 0 \leq x \leq 1 \\ \infty & x > 1. \end{cases}$$

Due to involved technical complications, however, we have excluded such norms (risk measures, respectively) from our framework and assumed supercoercivity. In contrast, as $p \to \infty$, we obtain $E_\Phi^L(t) = 1$ and we recover the expectation ($\mathcal{L}^\Psi = \mathcal{L}^1$) as in the previous example 6.5.

A curious phenomenon can be observed: when constraining the quantiles of dual variables by a Pareto distribution with $p = 2$, we obtain the space $\mathcal{L}^2$. However, the Pareto distribution itself only has a finite variance when $p > 2$. Similarly, $p = 3$ yields the $\mathcal{L}^3$ space, but the Pareto distribution only has a Kurtosis for $p > 3$ (and so on). This means that the random variable $Y$ whose decreasing rearrangement we used to construct the Orlicz space via the envelope function $E_\Phi^L = Y^*$ is itself *not* in the space $\mathcal{L}^\Phi$, although it is the least upper bound on the decreasing rearrangements $\{t \mapsto Z^*(t) : ||Z||_\Phi^L \leq 1\}$ (recall Definition 5.1). Thus, the envelope function is a natural limit object. For example, we might say informally that the "limit" of those distributions which have a finite variance is the Pareto $p = 2$ distribution. An interesting special case is the pathological $p = 1$: the Pareto distribution only has an $\mathcal{L}^1$ norm (an expectation) for $p > 1$. In this way, the extremely heavy-tailed Pareto $p = 1$ distribution with envelope function $t \mapsto 1/t$ is as extreme as it gets, as the corresponding Orlicz space is the largest of all r.i. spaces (recall Proposition 4.3).

The question of when the reference distribution is itself in the space, i.e. $E_\Phi^L = Y^* \in \mathcal{L}^\Phi$ has a surprisingly simple and insightful answer, which is related to M3 in Proposition 5.6.

**Proposition 6.7.** *Let $\phi_\Psi$ (resp. $\phi_\Phi$) be the fundamental function of $\mathcal{L}^\Psi$ (resp. $\mathcal{L}^\Phi$) and $\phi_\Psi(t)/t = E_\Phi^L$. Then*

$$||t \mapsto \phi_\Psi(t)/t||_\Phi^L = ||E_\Phi^L||_\Phi^L < \infty \text{ if and only if } M_{\phi_\Phi} = \mathcal{L}^\Phi.$$

*Proof.*

$$||t \mapsto \phi_\Psi(t)/t||_\Phi^L = ||t \mapsto \Phi^{-1}(1/t)||_\Phi^L = \inf\left\{\lambda > 0 : \mathbb{E}\Phi\left[t \mapsto \frac{\Phi^{-1}(1/t)}{\lambda}\right] \leq 1\right\}. \tag{10}$$

From Pick et al. (2013, p. 412) we have that if the last expression is finite, then $M_{\phi_\Phi} = \mathcal{L}^\Phi$. Conversely, assume $M_{\phi_\Phi} = \mathcal{L}^\Phi$. Then we obtain from M3 in Proposition 5.6 together with the embedding theorem 5.9 that $E_\Phi^L = Y^* \in \mathcal{L}^\Phi$. $\square$

To put this into words, the random variable whose decreasing rearrangement (backwards quantiles) we used to specify the envelope function $E_\Phi^L$ is itself in the space $\mathcal{L}^\Phi$ if and only if $\mathcal{L}^\Phi$ coincides with its corresponding Marcinkiewicz space, that is, with the shared fundamental function $\phi_\Phi$. On the other hand, we know from Proposition 5.6 that $t \mapsto E_\Phi^L = \phi_\Psi(t)/t \in M_{\phi_\Phi}$ if and only if the envelope approach based on quantiles is equivalent to that based on CVar's; formally, if the Marcinkiewicz quasi-norm $|\cdot|_{M_{\phi_\Phi}}$ and the Marcinkiewicz norm $||\cdot||_{M_{\phi_\Phi}}$ coincide (we use the "dual" of Proposition 5.6). Due to the embedding theorem, $t \mapsto \phi_\Psi(t)/t \in M_{\phi_\Phi}$ is a necessary, but not sufficient condition for $t \mapsto \phi_\Psi(t)/t \in \mathcal{L}^\Phi$. We clarify this with some examples.

**Example 6.8.** $L^\Phi = L_{\exp} : \Phi(y) = \exp(y) - 1$, $\Phi^{-1}(x) = \log(x + 1)$. Then:

$$\exists \lambda > 0 : ||t \mapsto \phi_\Psi(t)/t||_\Phi^L \leq \int_0^1 \exp(\lambda \log(1/x + 1)) - 1 < \infty$$

is finite, for instance with $\lambda = 1/e$. Therefore $E_\Phi^L \in L_{\exp}$, that is, an exponential distribution is itself in the space of subexponential random variables.

**Example 6.9.** $L^\Phi = \text{LlogL}$: $\Phi(x) = (x\log(x) - x + 1) \cdot \chi_{[1,\infty)}$. We have that $\phi_\Psi(t) = 1/(1 - \log(t))$ up to equivalence, since $L^\Psi = L_{\exp}$. Then condition M1 is (here $\phi = \phi_\Psi$):

$$\forall s > 1 : \sup_{0 < t \leq 1} \frac{1 - \log(t)}{1 - \log(t/s)} < 1.$$

But for any $s > 1$, we obtain that $\lim_{t \downarrow 0} \frac{1 - \log(t)}{1 - \log(t/s)} = 1$. Hence, $t \mapsto \phi_\Psi(t)/t \notin \text{LlogL}$. Consequently, LlogL does not coincide with the Marcinkiewicz space $M_{\phi_\Phi}$ and moreover the envelope function $E_\Phi^L$ grows so rapidly as $t \downarrow 0$ that the Marcinkiewicz quasi-norm does not coincide with the Marcinkiewicz norm.

**Example 6.10.** $L^\Phi = \mathcal{L}^p$: let $1 < p < \infty$ and $\Phi(x) = x^p$, as in the Pareto example 6.6. Then $||t \mapsto \phi_\Psi(t)/t||_\Phi^L = \infty$ is easily checked by using (10). Thus, for instance the $\chi^2$ divergence risk measure is *not* finite on the Marcinkiewicz space with fundamental function $\phi(t) = \sqrt{t}$ (here $L^\Phi = L^\Psi = \mathcal{L}^2$). However, we have that $||t \mapsto \phi(t)/t||_{M_{\phi_\Phi}} < \infty$, that is, the conditions of Proposition 5.6 apply. Therefore the Marcinkiewicz quasi-norm and the Marcinkiewicz norm are equivalent, but their induced space is strictly larger than the Orlicz space (the $\mathcal{L}^p$ space). In fact, this larger space is known as the "*weak $\mathcal{L}^p$ space*", which is used in the theory of interpolation of operators. Thus, we find that the Pareto distribution of parameter $p$, even if it does not have a finite $\mathcal{L}^p$ norm, is the natural representative and member of the weak $\mathcal{L}^p$ space, as it is the least upper bound on the quantiles of random variables in $\mathcal{L}^p$. For $p = 1$ and $p \to \infty$, the Marcinkiewicz and the Orlicz space coincide[11].

In summary, in this section we have proposed a way to construct an Orlicz norm with a desired tail sensitivity based on the specification of the least upper bound on the quantile behaviour of random variables in the space. Due to the equivalence relation in Example 4.24, this approach can then be used to construct a corresponding divergence risk measure, where the tail sensitivity is finely tuned to the reference distribution, which is the envelope function.

## 6.1 Utility-Based Shortfall Risk

Until now, we have always considered *coherent* risk measures to encode a specific tail sensitivity. However, there is another related class, the *convex* risk measures which have gained much attention recently in financial risk measurement. A potential problem is that due to positive homogeneity, $R(\lambda X) = \lambda R(X)$, the risk assessment scales just like the loss variable itself. This might be undesirable due to hazards arising from very large risks. Thus it may be desirable to construct risk measures which are not positively homogeneous, so that the risk assessment can grow faster in the presence of large risks. A convex risk measure, which has received much attention in finance particularly as a tool to measure heavy-tailed risk (Giesecke et al., 2008), is the *utility-based shortfall risk* (UBSR) (Föllmer & Schied, 2002). In the context of machine learning it has not seem widespread use yet; but, for example, Shen et al. (2014) apply it as a tool for risk-sensitive reinforcement learning. Furthermore, Li et al. (2021) apply a special case of the UBSR[12] to practical machine learning problems and show that it can flexibly tune the impact of individual losses. In Appendix A.4, we discuss the UBSR, which has been linked to Orlicz norms. We make this connection yet more explicit; consequently, the insights from the previous section can then be directly applied to construct a utility-based shortfall risk measure with a specific tail sensitivity. For instance, the choice in Li et al. (2021) corresponds to a subexponential tail sensitivity.

## 7 Tail-Specific Orlicz Deviation Inequalities

Few papers in the machine learning literature have made explicit use of Orlicz space theory; see for instance (Andoni et al., 2018), (Song et al., 2019) and (Chamakh et al., 2020). In contrast, *concentration inequalities* are widely used in machine learning in the context of statistical learning. Many concentration inequalities are in fact merely special cases of a general *Orlicz deviation inequality*; the standard textbook Boucheron et al. (2013, p. 45) on concentration inequalities only mentions Orlicz spaces in passing. We refer to (van der Vaart & Wellner, 1996, Part 2.2) for the general approach; concrete cases of tail sensitivity which have been studied in the literature are the sub-Gaussian and the sub-Exponential norm (see e.g. (Vershynin, 2018)), the sub-Weibull norm (Vladimirova et al., 2020), the Bernstein-Orlicz norm (van de Geer & Lederer, 2013), the Bennett-Orlicz norm (Wellner, 2017) and more (Chamakh et al., 2021). Consequently, all of these give rise to equivalent divergence risk measures.

We briefly present the Orlicz deviation inequality, since we believe that this general viewpoint may be helpful to the machine learning community. Furthermore, we show how it naturally fits within our approach to tail sensitivity based on the envelope function (hence, the fundamental function).

---

[11]This is due to the special situation that for $\mathcal{L}^1$ and $\mathcal{L}^\infty$ all r.i. norms are equivalent.

[12]For positive values of their parameter $t$, the objective in (Li et al., 2021) is the convex entropic risk measure (Föllmer & Knispel, 2011) on the empirical distribution. Note that the convex entropic risk measure is a special case of the UBSR (Giesecke et al., 2008).

Assume a finite Young function $\Psi$ and $X \geq 0$. Then, applying Markov's inequality to the nonnegative and increasing function $x \mapsto \Psi(x/||X||_\Psi^L)$ yields (van der Vaart & Wellner, 1996, Part 2.2):

$$P(X \geq x) \leq P\left(\Psi(X/||X||_\Psi^L) \geq \Psi(x/||X||_\Psi^L)\right), \quad \forall x > 0$$
$$\leq \frac{\mathbb{E}\left[\Psi(X/||X||_\Psi^L)\right]}{\Psi(x/||X||_\Psi^L)}, \quad \forall x > 0$$
$$\leq \frac{1}{\Psi(x/||X||_\Psi^L)}, \quad \forall x > 0,$$

which follows from the definition of the Luxemburg norm. In summary, for a finite Young function $\Psi$ we obtain:

$$P(X \geq x) \leq \frac{1}{\Psi(x/||X||_\Psi^L)}, \quad \forall x > 0. \tag{11}$$

For each finite Young function $\Psi$, we therefore obtain a deviation inequality. In fact, we observe that it bounds the tail behaviour by the distribution function of the envelope function. Recall that the distribution function is $\mu_Y(\lambda) := P(\{\omega \in \Omega : |Y(\omega)| > \lambda\})$ and that its generalized inverse is $Y^*$. For simplicity, we shall assume that $\mu_Y$ is continuous and invertible. Of course, $\mu_Y = \mu_{Y^*}$ holds. Since confusion could arise, we distinguish the fundamental function of the Orlicz norm $\phi_\Phi^O$ from the fundamental function of the Luxemburg norm $\phi_\Phi^L$, where the subscript indicates the Young function.

**Proposition 7.1** (Rubshtein et al. (2016, p. 177)). *The fundamental function of the Luxemburg norm $||\cdot||_\Psi^L$ is given by:*

$$\phi_\Psi^L(t) = \frac{1}{\Psi^{-1}(1/t)} = \frac{t}{\phi_\Phi^O(t)}, \quad \forall t \in (0, 1].$$

For a given Luxemburg norm, we can directly relate the deviation inequality to the envelope function.

**Proposition 7.2.** *With the envelope function $Y^*(t) := E_\Psi^L(t) = 1/\phi_\Psi^L(t)$, where we assume $Y^*(1) = 1$ (without loss of generality) and finite $\Psi$, we obtain the deviation inequality:*

$$P(X \geq x) \leq \mu_{Y^*}(x/||X||_\Psi^L), \quad x \geq ||X||_\Psi^L, X \geq 0.$$

*Proof.* Just observe that

$$\phi_\Psi^L(t) = \frac{1}{\Psi^{-1}(1/t)} = \frac{1}{E_\Psi^L(t)} = \frac{1}{Y^*(t)} \implies \Psi(x) = \frac{1}{\mu_{Y^*}(x)}, \quad \forall x \geq 1,$$

and use (11). $\square$

To intuit this result, recall that the envelope function constrains the quantile behaviour of random variables in an Orlicz space. Thus, it is unsurprising that the corresponding distribution function plays an analogous role. Conversely, we offer the following constructive approach to obtain a deviation inequality with a desired tail sensivity based on a reference distribution.

**Proposition 7.3.** *Choose $E_\Psi^L : (0, 1] \to \mathbb{R}^+$ as the decreasing rearrangement of some reference distribution, where $E_\Psi^L(0+) = \infty$, $E_\Psi^L(1) = 1$, and for which $t \mapsto \phi_\Phi^O(t) = E_\Psi^L(t) \cdot t$ is increasing and concave on $(0, 1]$ and which satisfies $\phi_\Phi^O(0+) = 0$. We set $\phi_\Phi^O(0) = 0$. With the envelope function $Y^*(t) := E_\Psi^L(t) = 1/\phi_\Psi^L(t)$, the following Orlicz deviation inequality holds:*

$$P(X \geq x) \leq \mu_{Y^*}(x/||X||_\Psi^L), \quad x \geq ||X||_\Psi^L, X \geq 0.$$

*Proof.* We apply the "dual" of Proposition 6.2 and obtain an Orlicz norm $||\cdot||_\Phi^O$ with fundamental function $\phi_\Phi^O$. Observe that by the associate relationship

$$\phi_\Phi^O(t) = \frac{t}{\phi_\Psi^L(t)} = E_\Psi^L(t) \cdot t.$$

The associate of the Orlicz norm $||\cdot||_\Phi^O$ is the Luxemburg norm $||\cdot||_\Psi^L$ with envelope $E_\Psi^L$. Also, we get from Proposition 6.2 that $\Psi$ is a finite Young function and therefore the deviation inequality holds due to Proposition 7.2. $\square$

Again, if $\phi_\Phi^O(t) := E_\Psi^L(t) \cdot t$ is at least quasiconcave, for which it suffices that $\phi_\Phi^O(t) := E_\Psi^L(t) \cdot t$ is increasing (cf. Section 5.3), we may use the equivalent least concave majorant in the construction to obtain a deviation inequality, which is equivalent up to a constant multiplicative factor.

**Example 7.4.** The sub-Exponential deviation inequality: let $E_\Psi^L(t) = 1 - \log(t)$ (Example 5.10). Then:

$$P(X \geq x) \leq \exp(-x/||X||_\Psi^L + 1), \quad x \geq ||X||_\Psi^L, X \geq 0.$$

where the Young function $\Psi(x) = x$ for $x \in (0, 1]$ and $\Psi(x) = \exp(x - 1)$ for $x \geq 1$ satisfies the equation $\Psi^{-1}(1/t) = E_\Psi^L(t)$ for $t \in (0, 1]$.

**Example 7.5.** The Pareto deviation inequality: let $E_\Psi^L(t) = t^{-1/p}$, $1 < p < \infty$. Then $\Psi(x) = x^p$ and:

$$P(X \geq x) \leq \frac{1}{\left(x/||X||_\Psi^L\right)^p}, \quad x \geq ||X||_\Psi^L, X \geq 0.$$

For $p = 2$, this gives Chebyshev's inequality.

# 8 Application

We have demonstrated how, given a specific tail-sensitivity in the form of a fundamental function, one can canonically construct three r.i. norms: the extremal Marcinkiewicz and Lorentz norms, and the Orlicz norm in between. In some cases, the Orlicz norm is equivalent to one or even both of them. This raises the question of whether one of these norms is superior. To give some practical advice, we consider corresponding risk measures and how to compute with them. These are then candidates to replace the expectation in expected risk minimization, in order to obtain a specified tail sensitivity.

The downside of the Marcinkiewicz norm for such practical use is that it lacks the property of translation equivariance (C4), even when restricted to nonnegative random variables and constants only. However, Fröhlich & Williamson (2022) recently defined an equivalent norm, which is induced by a translation equivariant risk measure[13]:

**Proposition 8.1.** *(Fröhlich & Williamson, 2022, Theorem 4.16) Given any concave fundamental function $\phi$, which is continuous at 0, the positive translation equivariant Marcinkiewicz norm is defined as:*

$$||X||_{TM_\phi} := \sup_{0<t<1} \frac{1 - \phi(1-t)}{t} \cdot \mathbb{E}[|X|] + \left(1 - \frac{1 - \phi(1-t)}{t}\right) \cdot \text{CVar}_t(|X|).$$

*It holds that $||\cdot||_{M_\phi}$ and $||\cdot||_{TM_\phi}$ are equivalent. A corresponding coherent risk measure, which induces this norm, is:*

$$TM_\phi(X) := \sup_{0<t<1} \frac{1 - \phi(1-t)}{t} \cdot \mathbb{E}[X] + \left(1 - \frac{1 - \phi(1-t)}{t}\right) \cdot \text{CVar}_t(X).$$

*The $||\cdot||_{TM_\phi}$ norm is the smallest r.i. norm with fundamental function $\phi$, which is induced by a coherent risk measure.*

*Proof.* We observe that this norm (risk measure, respectively) is a supremum over convex combinations of the expectation and $\text{CVar}_\alpha$, where the weightings depend on the fundamental function to achieve the desired tail-sensitivity. The class of risk measures is closed under convex combinations and taking suprema, hence $TM_\phi$ is a legitimate coherent risk measure, which is obviously r.i. by definition. By taking absolute values, the norm $||\cdot||_{TM_\phi}$ is induced. For the remaining statements, see Fröhlich & Williamson (2022, Theorem 4.16). □

---

[13]An r.i. norm can *never* be translation equivariant due to absolute homogeneity. Therefore Fröhlich & Williamson (2022) have defined the related property of positive translation equivariance (PTE) for a norm: a norm is called PTE if $\forall X \geq 0$, $c \in \mathbb{R}$ such that $X + c \geq 0$: $||X + c|| = ||X|| + c$. Indeed, the norm in Proposition 8.1 turns out to be the smallest PTE r.i. norm with a given fundamental function (Fröhlich & Williamson, 2022, Theorem 4.16).

Thus, the $TM_\phi$ risk measure is distinguished in the sense that the induced norm is the smallest of all r.i. norms which are induced by risk measures of fundamental function $\phi$, thereby providing the smallest (most optimistic) risk assessment.

A spectral risk measure, which induces a Lorentz norm, extends the fundamental function in the most cautious (pessimistic) possible way. The Lorentz norm $|| \cdot ||_{\Lambda_\phi}$ corresponds to the spectral risk measure $R_\phi$:

$$R_\phi(X) := \int_0^1 F_X^{-1}(1 - \omega) \, \mathrm{d}\phi(\omega).$$

Thus, having specified a desired tail sensitivity, using the coherent risk measures $TM_\phi$ and $R_\phi$ are the extremal choices and they hence provide a maximal "model uncertainty" regarding the choice of risk measure.

**Example 8.2.** (Fröhlich & Williamson, 2022) Let $\phi(t) = 1 - (1 - t)^2$. Then consider

$$R_\phi(X) = \mathrm{MaxVar}(X) = \mathbb{E}_{X_1, X_2 \sim X} \left[ \max(X_1, X_2) \right]$$
$$TM_\phi(X) = \mathrm{Dutch}(X) = \mathbb{E} \left[ \max(X, \mathbb{E}[X]) \right],$$

where $X_1, X_2 \sim X$ means that $X_1$ and $X_1$ are independent and identically distributed. These risk measures are known as $\mathrm{MaxVar}$[14] (Cherny & Madan, 2009) and the Dutch risk measure (Van Heerwaarden & Kaas, 1992).

The Orlicz norm (the Orlicz regret and risk measure, respectively) is situated in between these two ends of optimism and pessimism. However, we refrain from making a general recommendation as to which of the three candidates (or even another risk measure with the given fundamental function) to use in practice. We make some remarks regarding the choice.

- Spectral risk measures are embedded in a rich literature, from economics (Quiggin, 2012), insurance (Wang, 2000) to philosophy (Buchak, 2013), and can be motivated from various perspectives as a tool for rational decision making. However, the computation of spectral risk measures is non-trivial. We refer to (Pandey et al., 2021) for the estimation of spectral risk measures from samples and to (Holland & Haress, 2022), (Leqi et al., 2022) and (Mehta et al., 2022) for learning schemes, which may be used in practice to replace ERM with spectral risk measures. Bäuerle & Glauner (2021) study the minimization of spectral risk measures in the context of Markov decision processes. Some spectral risk measures, like $\mathbb{E}$, $\mathrm{CVar}_\alpha$ or convex combinations thereof are easily computed and optimized; a useful form is that of Example 3.20. To improve the stability of optimizing $\mathrm{CVar}_\alpha$ in a batch setting, Curi et al. (2020) have proposed an adaptive sampling method; see also (Soma & Yoshida, 2020) and (Holland & Haress, 2021).

- The $TM_\phi$ risk measure, related to the Marcinkiewicz norm, is perhaps not as intuitive. However, it is easy to compute in a risk minimization loop. As of now, we see little motivation for employing it in practice.

- The $f$-divergence risk measure is situated in between, albeit in many cases equivalent to one of the above, as we observed in Section 6. Specifically, a highly tail-sensitive divergence risk measure is typically equivalent to the corresponding Marcinkiewicz norm. An advantage is that a divergence risk measure naturally comes with a corresponding deviation inequality, as we established in Section 7. Also, some prominent spectral risk measures like $\mathbb{E}$ or $\mathrm{CVar}_\alpha$ are divergence risk measures. The infimum-based representation (5) allows practical computation. For instance, when training a neural network, a single line suffices to implement (5) in a framework such as `PyTorch`. We also refer to Chouzenoux et al. (2019) who propose a suitable accelerated projected gradient algorithm.

On the other hand, the utility-based shortfall risk has different use cases such as reinforcement learning, as it lacks positive homogeneity and is thus not suitable in a standard supervised learning risk minimization

---

[14]The name MaxVar is due to our loss-based orientation here; the authors originally called it MinVar due to a sign flip.

loop. Finally, use cases for the Orlicz regret measure may appear in risk-sensitive regression; we refer to Rockafellar & Uryasev (2013) and Rockafellar et al. (2008) for regression based on *error measures* E, which are based on regret measures $V$ via the unique correspondence $\mathrm{E}(X) = V(X) - \mathbb{E}[X]$. For instance, quantile regression is associated with the $\mathrm{CVar}_\alpha$ Orlicz regret measure.

## 9 Conclusion

In this work, we have brought together concepts such as the fundamental function and envelope function from the theory of r.i. Banach function spaces with the framework of coherent risk measures, in particular divergence risk measures. As a consequence, we have obtained a general approach to fine-tune the tail sensitivity of a risk measure, which then functions as a more cautious replacement for the expectation in ERM, or more generally, can be used to assess the risk inherent in a loss distribution.

We have demonstrated how tail sensitivity is related to the fundamental function. To a first approximation, those risk measures with the same fundamental function share the same tail sensitivity; on a finer level, however, it might be the case that the spaces on which different such risk measures are finite do not coincide. Nonetheless, the fundamental function offers a useful and simple stratification of the space of coherent risk measures. In this way, the choice of fundamental function indeed appears more *fundamental* than the exact choice of risk measure; Marcinkiewicz, Orlicz, Lorentz or a still different one.

Abundant opportunities for future work arise. For instance, a quantitative comparison of the (positive translation equivariant) Marcinkiewicz, Orlicz and Lorentz norm and their corresponding risk measures on practical problems has not been conducted yet. Also, estimatability from finite samples is an open problem: we conjecture that the more tail-sensitive a risk measure, the harder it is to estimate it from finite samples. For instance, can one obtain convergence rates based on properties of the fundamental function?

### Broader Impact Statement

We study risk measures, which add tail sensitivity to empirical risk minimization. As a consequence, the use of such risk measures emphasizes highly negative outcomes and thus may reduce inequality in the loss distribution. However, this will certainly not fully remedy the problem of disparate impact that machine learning systems can have, for instance on individuals. Moreover, the choice of tail sensitivity and risk measure is an additional ethical choice for the machine learning engineer to make, which must be explicitly stated and about which disagreement is to be expected.

### Acknowledgments

This work was was funded by the Deutsche Forschungsgemeinschaft (DFG, German Research Foundation) under Germany's Excellence Strategy — EXC number 2064/1 — Project number 390727645. The authors thank the International Max Planck Research School for Intelligent Systems (IMPRS-IS) for supporting Christian Fröhlich. Thanks to Rabanus Derr for many helpful discussions and comments. We thank one anonymous reviewer in particular for providing helpful comments.

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

# A  Appendix

## A.1  Orlicz Regret Measures

### A.1.1  Proof of Proposition 3.2

*Proof.* The convex conjugate of a (not necessarily convex) function is always convex. To see that $g$ is finite-valued on $\mathbb{R}$ due to the supercoercivity of $f$ (a well-known statement, see (Edgar & Sucheston, 1992; Dommel & Pichler, 2021; Föllmer & Schied, 2002)), assume by contradiction that there exists a $y \in \mathbb{R}$ such that

$$\infty = g(y) = \sup_{x \geq 0} xy - f(x).$$

Since $f$ is proper, i.e. never takes on $-\infty$, the supremum cannot be attained for some $x \in \mathbb{R}$. Therefore we need to investigate only the limit as $x \to \infty$:

$$g(y) = \lim_{x \to \infty} xy - f(x) = \lim_{x \to \infty} x \left( y - \frac{f(x)}{x} \right).$$

But since $y$ is fixed, this could only yield $\infty$ if $\lim_{x \to \infty} \left( y - \frac{f(x)}{x} \right) \geq 0$, which contradicts the supercoercivity assumption $\lim_{x \to \infty} \frac{f(x)}{x} = \infty$.

To see that $g$ is increasing (non-decreasing), observe that if $y_1 \leq y_2$, then

$$g(y_1) = \sup_{z \geq 0} y_1 z - f(z) \leq \sup_{z \geq 0} y_2 z - f(z) = g(y_2).$$

$\square$

### A.1.2 Proof of Proposition 3.9

*Proof.* V1: Positive homogeneity: let $\lambda > 0$.

$$\begin{aligned}
V_g(\lambda X) &= \inf_{t > 0} t \left( 1 + \mathbb{E}g \left( \frac{\lambda X}{t} \right) \right) \\
&= \inf_{\tilde{t} > 0} \tilde{t}\lambda \left( 1 + \mathbb{E}g \left( \frac{X}{\tilde{t}} \right) \right), \quad \tilde{t} = \frac{t}{\lambda} \\
&= \lambda V_g(X).
\end{aligned}$$

V2: For subadditivity, we present the proof from (Cui et al., 2008) for subadditivity of an Orlicz norm, which also works in our setting of an Orlicz regret measure. First observe that $X \mapsto 1 + \mathbb{E}g(X)$ is convex due to the convexity of $g$. We also know that for an arbitrary $\epsilon > 0$ there exist some $t_1, t_2 > 0$ so that

$$\frac{1}{t_1}(1 + \mathbb{E}g(t_1 X)) \leq V_g(X) + \frac{\epsilon}{2},$$
$$\frac{1}{t_2}(1 + \mathbb{E}g(t_2 Y)) \leq V_g(Y) + \frac{\epsilon}{2}.$$

Let $t = \frac{t_1 t_2}{t_1 + t_2}$. Then it holds

$$\begin{aligned}
V_g(X + Y) &\leq \frac{1}{t}(1 + \mathbb{E}g(t(X + Y))) \\
&= \frac{t_1 + t_2}{t_1 t_2} \left( 1 + \mathbb{E}g \left( \frac{t_2}{t_1 + t_2} t_1 X + \frac{t_1}{t_1 + t_2} t_2 Y \right) \right) \\
&\leq \frac{1}{t_1}(1 + \mathbb{E}g(t_1 X)) + \frac{1}{t_2}(1 + \mathbb{E}g(t_2 Y)) \leq V_g(X) + V_g(Y) + \epsilon.
\end{aligned}$$

V3: Monotonicity directly follows from monotonicity of $g$ (Lemma 3.2). $\square$

V4: We want to show $V_g(X) \geq \mathbb{E}[X] \ \forall X \in \mathcal{M}$. Let $f \in \mathfrak{F}$ be a divergence function, i.e. we have $f(1) = 0$. Consequently, we have that

$$g(y) = \sup_{x \geq 0} xy - f(x) \geq y - f(1) = y \quad \forall y \in \mathbb{R}.$$

But then:

$$\begin{aligned}
\mathbb{E}[X] = t\mathbb{E}\left[ \frac{X}{t} \right] &\leq \inf_{t > 0} t + t\mathbb{E}\left[ \frac{X}{t} \right] \\
&\leq \inf_{t > 0} t \left( 1 + \mathbb{E}g \left( \frac{X}{t} \right) \right) = V_g(X).
\end{aligned}$$

V5: To see that $X = 0 \Rightarrow V_g(0) = 0$, observe that

$$V_g(0) = \inf_{t>0} t\left(1 + \mathbb{E}g\left(0\right)\right) = \lim_{t\to 0} t\left(1 + \mathbb{E}g(0)\right) = 0,$$

due to

$$g(0) = \sup_{x\geq 0} 0 \cdot z - f(z) < \infty.$$

Also note that since $f(1) = 0$, we have $g(0) \geq 0$.

For the converse direction, assume $V_g(X) = 0$ for $X \geq 0$. We want to show that it follows that $X = 0$. By contradiction, assume $X \neq 0$ (P-a.e.). But then since $X \geq 0$, we have $\mathbb{E}[X] > 0$. Due to aversity (V4) we know $V_g(X) \geq \mathbb{E}[X]$, therefore $V_g(X) > 0$.

### A.1.3   Proof of Proposition 3.10

*Proof.*

$$\begin{aligned}
V_{g_\varepsilon}(X) &= \inf_{t>0} t\left(1 + \mathbb{E}g_\varepsilon\left(\frac{X}{t}\right)\right) \\
&= \inf_{t>0} t\left(1 + \frac{1}{\varepsilon}\mathbb{E}g\left(\frac{\varepsilon X}{t}\right)\right) \\
&\overset{\tilde{t}=\frac{t}{\varepsilon}}{=} \inf_{\tilde{t}>0} \tilde{t}\varepsilon\left(1 + \frac{1}{\varepsilon}\mathbb{E}g\left(\frac{X}{\tilde{t}}\right)\right) \\
&= \inf_{\tilde{t}>0} \tilde{t}\left(\varepsilon + \mathbb{E}g\left(\frac{X}{\tilde{t}}\right)\right).
\end{aligned}$$

$\square$

### A.1.4   Proof of Proposition 3.11

*Proof.* That $\bar{g}$ is a valid Young function is obvious since $\bar{g}(0) = 0$ and it inherits the properties of $g$ (increasing, convex etc.) on the nonnegative domain. Recall that $\bar{f} = f \cdot \chi_{[1,\infty)}$. Its conjugate is therefore given by

$$h(y) \coloneqq \sup_{x\geq 0} xy - \bar{f}(x) = \max\left\{\sup_{0\leq x<1} xy, \sup_{x\geq 1} xy - \bar{f}(x)\right\}.$$

For negative $y$, we have $h(y) = 0$, since the maximum is attained in the left term with value $0$ (here the supremum is attained at $x = 0$), since the right term will be negative. Thus $h(y) = \bar{g}(y) = 0$ for $y < 0$. It remains to show that for positive $y$ we have $h(y) = g(y)$. For this, we show that for positive $y$, only the second term of $g$ suffices:

$$g(y) = \max\left\{\sup_{0\leq x<1} xy - f(x), \sup_{x\geq 1} xy - f(x)\right\} = \sup_{x\geq 1} xy - f(x), \quad y > 0.$$

Assume by contradiction that $y$ is positive and the maximum is attained in the first term. Note that $\sup_{x\geq 1} xy - f(x) \geq y$ since $f(1) = 0$. But:

$$\sup_{0\leq x<1} xy - f(x) \leq y,$$

since $xy \leq y$ and $f(x) \geq 0$. Hence the maximum is attained in the second term. Thus we have shown that $\bar{g} = h$ and therefore $\bar{g}$ is the conjugate of $\bar{f}$. $\square$

### A.1.5 Proof of Proposition 3.14

*Proof.* The first inequality is obvious. Second,

$$
\begin{aligned}
||X||_{V_{g_{\varepsilon_2}}} &= \inf_{t>0} t\left(\varepsilon_2 + \mathbb{E}g\left(\frac{X}{t}\right)\right) \leq \inf_{t>0} t\left(\varepsilon_2 + \frac{\varepsilon_2}{\varepsilon_1}\mathbb{E}g\left(\frac{X}{t}\right)\right) \\
&= \frac{\varepsilon_2}{\varepsilon_1}\inf_{t>0} t\left(\varepsilon_1 + \mathbb{E}g\left(\frac{X}{t}\right)\right) = \frac{\varepsilon_2}{\varepsilon_1}||X||_{V_{g_{\varepsilon_1}}},
\end{aligned}
$$

which is true since $\frac{\varepsilon_2}{\varepsilon_1} > 1$. $\qquad\square$

### A.1.6 Proof of Proposition 3.17

To establish the dual representation

$$
V_g(X) = \sup\{\mathbb{E}[XZ] : Z \in \mathcal{L}^{\bar{f}}, Z \geq 0, \mathbb{E}[f(Z)] \leq 1\} \quad \forall X \in \mathcal{L}^{\bar{g}},
$$

we follow roughly (Kosmol & Müller-Wichards, 2011, pp. 238-239). First note that $V_g$ is finite on $\mathcal{L}^{\bar{g}}$ (Proposition 3.12). We begin with dual variables $Z \in \mathcal{L}^1$, but later show that the restriction to $\mathcal{L}^{\bar{f}}$ is possible. The crucial point is that the functionals $X \mapsto \mathbb{E}[g(X)]$ and $Z \mapsto \mathbb{E}[f(Z)]$ stand in a conjugate relationship due to the conjugacy of $g$ and $f$:

$$
\mathbb{E}[g(X)] = \sup_{Z \in \mathcal{L}^1, Z \geq 0} \mathbb{E}[XZ] - \mathbb{E}[f(Z)] \tag{12}
$$

This result is from (Rockafellar, 1968). It follows from the fact that $g$ is a lower semicontinuous proper convex function with interior points in its domain and therefore, $G(\omega, x) = g(x)$ is a normal convex integrand (Rockafellar, 1968, Lemma 2). The conjugacy then follows from (Rockafellar, 1968, Corollary in Theorem 2). Also note that the restriction to dual variables with $Z \geq 0$ is possible due to $f(z) = \infty$ for $z < 0$ (this is essentially due to the monotonicity of $V_g$[15]). Then the Orlicz regret measure can be formulated as

$$
\begin{aligned}
V_g(X) &= \inf_{t>0} t\left(1 + \mathbb{E}g\left(\frac{X}{t}\right)\right) = \inf_{t>0} t\left(1 + \sup_{Z \in \mathcal{L}^1, Z \geq 0}\left(\mathbb{E}\left[\frac{X}{t}Z\right] - \mathbb{E}[f(Z)]\right)\right) \\
&= \inf_{t>0}\sup_{Z \in \mathcal{L}^1, Z \geq 0}\{\mathbb{E}[XZ] - t(\mathbb{E}[f(Z)] - 1).
\end{aligned}
$$

Now consider the dual representation:

$$
DR(X) \coloneqq \sup\{\mathbb{E}[XZ] : Z \in \mathcal{L}^1, Z \geq 0, \mathbb{E}[f(Z)] \leq 1\} \quad \forall X \in \mathcal{L}^1.
$$

Observe that $Z = 1$ is an interior point of the constraint set due to the fact that $f(1) = 0$. We therefore get strong Lagrangian duality since Slater's condition holds, as observed for instance by Shapiro (2017):

$$
DR(X) = \inf_{t>0}\sup_{Z \in \mathcal{L}^1, Z \geq 0}\{\mathbb{E}[XZ] - t(\mathbb{E}[f(Z)] - 1)\} = V_g(X),
$$

which establishes the desired equality. Furthermore, it is sufficient to restrict ourselves to dual variables $Z \in \mathcal{L}^{\bar{f}}$. Assume by contraposition $Z \notin \mathcal{L}^{\bar{f}}$. Hence $||Z||_{\bar{f}}^L = \infty$. But $||Z||_{\bar{f}}^L \leq \mathbb{E}[\bar{f}(Z)]$ holds, see Rubshtein et al. (2016, Proposition 14.2.1). But then $\mathbb{E}[\bar{f}(Z)] = \infty$, and since $\mathbb{E}[\bar{f}(Z)] \leq \mathbb{E}[f(Z)]$ such $Z$ is not feasible.

---

[15] The following statement is well-known under different technical setups: a regret measure is monotone if and only if all dual variables are nonnegative. See for instance (Rockafellar & Uryasev, 2013) for the $\mathcal{L}^2$ case.

### A.1.7   Proof of Proposition 3.19

*Proof.* We begin to show that the dual representation holds:

$$R_{g,\varepsilon}(X) = \sup\{E[XZ] : Z \in \mathcal{L}^{\bar{f}}, Z \geq 0, \mathbb{E}[Z] = 1, \mathbb{E}f(Z) \leq \varepsilon\} \quad \forall X \in \mathcal{L}^{\bar{g}}. \tag{13}$$

Consider how we defined the divergence risk measure in the first place:

$$R_{g,\varepsilon}(X) := \sup_{Q:d_f(Q,P)\leq\varepsilon} \mathbb{E}_Q[X].$$

Here, the supremum is taken over the set of all valid probability measures $Q$ which are absolutely continuous with respect to $P$, and further satisfy the $f$-divergence constraint. The necessary shift in perspective is to view $Q$ not as a probability measure, but instead take the following route:

$$Q_Z(A) = \int_A Z(\omega) \, \mathrm{d}P(\omega), \tag{14}$$

for the Radon-Nikodym derivative $Z$ (which can be more suggestively written as $\frac{\mathrm{d}Q}{\mathrm{d}P}$). $Q$ is a measure, i.e. a set function, whereas $Z$ is a random variable. Then observe that

$$d_f(Q, P) \leq \varepsilon \iff \mathbb{E}f(Z) \leq \epsilon,$$

where the expectation is with respect to the base measure $P$. Furthermore, imposing the constraint that $Z$ yields a valid probability measure via (14) amounts exactly to requiring that $Z \geq 0$ and $\mathbb{E}[Z] = 1$.

Observe that the envelope representation (13) is the envelope representation of the corresponding Orlicz regret measure, but intersected with the constraint that $\mathbb{E}[Z] = 1$. If we worked on a reflexive Banach space (where the bidual coincides with the space itself), together with an appropriate closedness condition, we could use the following result. For instance, consider for a moment the dual pair of $\mathcal{L}^2$ and $\mathcal{L}^2$ (the exponent 2 has dual exponent 2 since $1/2 + 1/2 = 1$).

Denote the support function of the set $\mathcal{Q}$ as $\sigma_{\mathcal{Q}}$:

$$\sigma_{\mathcal{Q}}(X) = \sup_{Q\in\mathcal{Q}} \langle X, Q \rangle, \quad \mathcal{Q} = \left\{ Q \in \mathcal{L}^2 : \langle X, Q \rangle \leq \sigma_{\mathcal{Q}}(X) \, \forall X \in \mathcal{L}^2 \right\},$$

and define $\mathcal{E}_1 := \{Z \in \mathcal{L}^2 : \mathbb{E}[Z] = 1\}$. The following is well-known (see for instance Fröhlich & Williamson (2022) for a statement on $\mathcal{L}^2$):

**Proposition A.1.** *On $\mathcal{L}^2$: let $V = \sigma_{\mathcal{Q}}$ be a positively homogeneous, subadditive and monotonic functional, i.e. a coherent regret measure, and $\mathcal{Q}$ be its supported set. Let $\mathcal{Q}' := \mathcal{Q} \cap \mathcal{E}_1 \neq \emptyset$. Then $R := \sigma_{\mathcal{Q}'}$ is a coherent risk measure and $R(X) = \inf_{c\in\mathbb{R}} V(X - c) + c$.*

The proof uses reflexivity of $\mathcal{L}^2$. Unfortunately, Orlicz spaces are reflexive if and only if they satisfy the $\Delta_2$ condition, which limits the speed of growth of the Young function; see for instance (Kosmol & Müller-Wichards, 2011, p. 197, p. 234). Since we do not want to assume $\Delta_2$ throughout (this would for instance exclude the KL divergence risk measure), we cannot rely on the statement, but we are guided by it for intuition.

Our aim is to show that $R_{g,\varepsilon}$ is given by the infimal convolution

$$R_{g,\varepsilon}(X) = \inf_{\mu\in\mathbb{R}} \mu + V_{g_\varepsilon}(X - \mu).$$

The proof works analogously to A.1.6 and hence (Kosmol & Müller-Wichards, 2011); confer also (Dommel & Pichler, 2021, Theorem 5.1). Start with the envelope representation

$$R_g(X) := \sup\left\{ \mathbb{E}[XZ] : Z \in \mathcal{L}^{\bar{f}}, Z \geq 0, \mathbb{E}[Z] = 1, \mathbb{E}[f_\varepsilon(Z)] \leq 1 \right\} \quad \forall X \in \mathcal{L}^1.$$

Regarding the restriction to $Z \in \mathcal{L}^{\bar{f}}$, the same remark as in A.1.6 applies. Again, since $Z = 1$ is an interior point of the constraint set, observe that by Lagrange duality:

$$R_g(X) = \inf_{\tilde{\mu} \in \mathbb{R}, t > 0} \sup_{Z \in \mathcal{L}^1, Z \geq 0} \{\mathbb{E}[XZ] - t(\mathbb{E}[f_\varepsilon(Z)] - 1) - \tilde{\mu}(\mathbb{E}[Z] - 1)\}$$

$$= \inf_{\tilde{\mu} \in \mathbb{R}, t > 0} t \left(1 + \frac{\tilde{\mu}}{t} + \sup_{Z \in \mathcal{L}^1, Z \geq 0} \mathbb{E}\left[\left(\frac{X - \tilde{\mu}}{t}\right) Z\right] - \mathbb{E}[f_\varepsilon(Z)]\right)$$

$$\overset{\mu = \frac{\tilde{\mu}}{t}}{=} \inf_{\mu \in \mathbb{R}, t > 0} t \left(1 + \mu + \sup_{Z \in \mathcal{L}^1, Z \geq 0} \mathbb{E}\left[\left(\frac{X}{t} - \mu\right) Z\right] - \mathbb{E}[f_\varepsilon(Z)]\right)$$

$$= \inf_{\mu \in \mathbb{R}, t > 0} t \left(1 + \mu + \mathbb{E} g_\varepsilon \left(\frac{X}{t} - \mu\right)\right), \tag{15}$$

where we applied (12) to the conjugacy relation of $f_\varepsilon$ and $g_\varepsilon$. We started with $X \in \mathcal{L}^1$, but the subsequent Proposition 3.21 shows that $R_g$ is finite on $\mathcal{L}^{\bar{g}}$, which is its natural domain (Pichler, 2013). Henceforth we restrict $X$ to $\mathcal{L}^{\bar{g}}$.

Now we explicitly compute the infimal convolution of $V_{g_\epsilon}$:

$$R_{g,\varepsilon}(X) = \inf_{\mu \in \mathbb{R}} \mu + V_{g_\varepsilon}(X - \mu)$$

$$= \inf_{\mu \in \mathbb{R}, \tilde{t} > 0} \mu + \tilde{t} \left(\varepsilon + \mathbb{E} g \left(\frac{X - \mu}{\tilde{t}}\right)\right)$$

$$\overset{\mu = \tilde{t}\tilde{\mu}}{=} \inf_{\tilde{\mu} \in \mathbb{R}, \tilde{t} > 0} \tilde{t}\tilde{\mu} + \tilde{t} \left(\varepsilon + \mathbb{E} g \left(\frac{X - \tilde{t}\tilde{\mu}}{\tilde{t}}\right)\right)$$

$$= \inf_{\tilde{\mu} \in \mathbb{R}, \tilde{t} > 0} \tilde{t} \left(\varepsilon + \tilde{\mu} + \mathbb{E} g \left(\frac{X}{\tilde{t}} - \tilde{\mu}\right)\right),$$

where we used the form of $V_{g_e}$ with explicit $\varepsilon$ (Proposition 3.10). This is the representation obtained by Dommel & Pichler (2021). But we can also hide $\varepsilon$ in the function $g_\varepsilon$:

$$R_{g,\varepsilon}(X) = \inf_{\tilde{\mu} \in \mathbb{R}, \tilde{t} > 0} \tilde{t} \left(\varepsilon + \tilde{\mu} + \mathbb{E} g \left(\frac{X}{\tilde{t}} - \tilde{\mu}\right)\right)$$

$$\overset{\tilde{\mu} = \mu\varepsilon}{=} \inf_{\mu \in \mathbb{R}, \tilde{t} > 0} \tilde{t}\varepsilon \left(1 + \mu + \frac{1}{\varepsilon} \mathbb{E} g \left(\frac{X}{\tilde{t}} - \mu\varepsilon\right)\right)$$

$$\overset{\tilde{t} = \frac{t}{\varepsilon}}{=} \inf_{\mu \in \mathbb{R}, t > 0} t \left(1 + \mu + \frac{1}{\varepsilon} \mathbb{E} g \left(\varepsilon \left(\frac{X}{t} - \mu\right)\right)\right)$$

$$= \inf_{\mu \in \mathbb{R}, t > 0} t \left(1 + \mu + \mathbb{E} g_\varepsilon \left(\frac{X}{t} - \mu\right)\right),$$

which coincides with (15) and therefore we have established that $R_{g,\varepsilon}(X) = \inf_{\mu \in \mathbb{R}} \mu + V_{g_\varepsilon}(X - \mu)$.

*The "natural extension" representation:* Finally, we establish that the following envelope representation also holds:

$$R_{g,\varepsilon}(X) = \sup \left\{E_{Q_Z}[X] : Q_Z \in \mathcal{Q}, \mathbb{E}_{Q_Z}[Y] \leq V_{g_\varepsilon}(Y) \quad \forall Y \in \mathcal{L}^{\bar{g}}\right\} \quad \forall X \in \mathcal{L}^{\bar{g}}.$$

When one can work with support functions, this representation is easy to show[16]. However, in our Orlicz setting we have to prove it. For this, consider the representation (the validity of which we show later):

$$R_{g,\varepsilon}(X) = \sup \left\{E[XZ] - \alpha(Z) : Z \in \mathcal{L}^{\bar{f}}, Z \geq 0, \mathbb{E}[Z] = 1\right\} \quad \forall X \in \mathcal{L}^{\bar{g}}, \tag{16}$$

---

[16] Just consider the definition of the supported set of a support function and then use standard results to obtain the constraints $Z \geq 0$ and $\mathbb{E}[Z] = 1$ from monotonicity and translation equivariance, which imply $Q_Z \in \mathcal{Q}$.

where

$$\alpha(Z) := \sup\left\{\mathbb{E}[XZ] : X \in \mathcal{L}^{\bar{g}}, R_{g,\varepsilon}(X) \leq 0\right\},$$

which is the envelope representation of a convex risk measure. Since in our case, we have a *coherent* risk measure, it is standard that in fact the "penalty function" $\alpha$ can be taken to be the convex indicator $i_{\mathcal{Z}}$ of the envelope $\mathcal{Z}$, where $\mathcal{Z} := \{Z \in \mathcal{L}^{\bar{f}} : Z \geq 0, \mathbb{E}[Z] = 1, \mathbb{E}f_{\varepsilon}(Z) \leq 1\}$. Then:

$$R_{g,\varepsilon}(X) = \sup\left\{E[XZ] - i_{\mathcal{Z}}(Z) : Z \in \mathcal{L}^{\bar{f}}, Z \geq 0, \mathbb{E}[Z] = 1\right\}.$$

The *acceptance set* is $\mathcal{A} = \{X \in \mathcal{L}^{\bar{g}} : R_{g,\varepsilon}(X) \leq 0\}$. We have:

$$\alpha(Z) = i_{\mathcal{Z}}(Z) = \sup\left\{\mathbb{E}[XZ] : X \in \mathcal{L}^{\bar{g}}, R_{g,\varepsilon}(X) \leq 0\right\}.$$

Recall that the convex indicator function is $i_{\mathcal{Z}}(Z) = 0$ if $Z \in \mathcal{Z}$ and $+\infty$ otherwise. Thus we find that

$$\mathcal{Z} = \left\{Z \in \mathcal{L}^{\bar{f}} : Z \geq 0, \mathbb{E}[Z] = 1, 0 = \sup\{\mathbb{E}[XZ] : X \in \mathcal{L}^{\bar{g}}, R_{g,\varepsilon}(X) \leq 0\}\right\} \tag{17}$$

$$= \left\{Z \in \mathcal{L}^{\bar{f}} : Z \geq 0, \mathbb{E}[Z] = 1, \mathbb{E}[XZ] \leq 0 \quad \forall X \in \mathcal{L}^{\bar{g}} \text{ with } R_{g_{\varepsilon}}(X) \leq 0\right\},$$

This follows since $R_{g,\varepsilon}(0) = 0$, i.e. 0 is always in the acceptance set, hence the inner supremum in (17) can never be $< 0$. Further, with similar reasoning as in Bellini & Gianin (2008, Proposition 17):

$$\mathcal{Z} = \left\{Z \in \mathcal{L}^{\bar{f}} : Z \geq 0, \mathbb{E}[Z] = 1, \mathbb{E}[XZ] \leq 0 \quad \forall X \in \mathcal{L}^{\bar{g}} \text{ with } R_{g_{\varepsilon}}(X) \leq 0\right\}$$

$$= \left\{Z \in \mathcal{L}^{\bar{f}} : Z \geq 0, \mathbb{E}[Z] = 1, \mathbb{E}[XZ] \leq R_{g_{\varepsilon}}(X) \quad \forall X \in \mathcal{L}^{\bar{g}}\right\} =: \mathcal{Z}^{\dagger}.$$

That $\mathcal{Z}^{\dagger} \subseteq \mathcal{Z}$ is obvious from the definition of $R_{g,\varepsilon}$. Conversely, we show $\mathcal{Z} \subseteq \mathcal{Z}^{\dagger}$. Let $Z \in \mathcal{Z}$, i.e. $Z \geq 0$, $\mathbb{E}[Z] = 1$ and $\mathbb{E}[XZ] \leq 0 \quad \forall X \in \mathcal{L}^{\bar{g}}$ with $R_{g_{\varepsilon}} \leq 0$. Then consider $Y := X - R_{g_{\varepsilon}}(X)$ for arbitrary $X \in \mathcal{L}^{\bar{g}}$. Then $Y \in \mathcal{L}^{\bar{g}}$ since an Orlicz space is closed under subtraction of constants (consider the definition of the Luxemburg norm). Due to translation equivariance, $R_{g_{\varepsilon}}(Y) = R_{g_{\varepsilon}}(X - R_{g_{\varepsilon}}(X)) = R_{g_{\varepsilon}}(X) - R_{g_{\varepsilon}}(X) = 0 \leq 0$, hence $Y$ is acceptable and therefore $\mathbb{E}[YZ] \leq 0$ by assumption that $Z \in \mathcal{Z}$. Then we know that

$$\mathbb{E}[YZ] \leq 0 \Leftrightarrow \mathbb{E}[(X - R_{g_{\varepsilon}}(X))Z] \leq 0 \Leftrightarrow \mathbb{E}[XZ] - R_{g_{\varepsilon}}(X) \leq 0 \Leftrightarrow \mathbb{E}[XZ] \leq R_{g_{\varepsilon}}(X),$$

noting that for a constant $c$, it holds $\mathbb{E}[(X - c)Z] = \mathbb{E}[XZ] - c$ since $\mathbb{E}[Z] = 1$. Since $X$ was arbitrary in $\mathcal{L}^{\bar{g}}$, we conclude that $Z \in \mathcal{Z}^{\dagger}$, and since $Z$ was arbitrary, $\mathcal{Z} \subseteq \mathcal{Z}^{\dagger}$. Having established $\mathcal{Z} = \mathcal{Z}^{\dagger}$, we continue:

$$\mathcal{Z} = \mathcal{Z}^{\dagger} = \left\{Z \in \mathcal{L}^{\bar{f}} : Z \geq 0, \mathbb{E}[Z] = 1, \mathbb{E}[XZ] \leq \inf_{\mu \in \mathbb{R}} \mu + V_{g_{\varepsilon}}(X - \mu) \quad \forall X \in \mathcal{L}^{\bar{g}}\right\}$$

$$= \left\{Z \in \mathcal{L}^{\bar{f}} : Z \geq 0, \mathbb{E}[Z] = 1, \mathbb{E}[XZ] \leq \mu + V_{g_{\varepsilon}}(X - \mu) \quad \forall X \in \mathcal{L}^{\bar{g}}, \forall \mu \in \mathbb{R}\right\}$$

$$= \left\{Z \in \mathcal{L}^{\bar{f}} : Z \geq 0, \mathbb{E}[Z] = 1, \mathbb{E}[XZ] - \mu \leq V_{g_{\varepsilon}}(X - \mu) \quad \forall X \in \mathcal{L}^{\bar{g}}, \forall \mu \in \mathbb{R}\right\}$$

$$= \left\{Z \in \mathcal{L}^{\bar{f}} : Z \geq 0, \mathbb{E}[Z] = 1, \mathbb{E}[(X - \mu)Z] \leq V_{g_{\varepsilon}}(X - \mu) \quad \forall X \in \mathcal{L}^{\bar{g}}, \forall \mu \in \mathbb{R}\right\}$$

$$= \left\{Z \in \mathcal{L}^{\bar{f}} : Z \geq 0, \mathbb{E}[Z] = 1, \mathbb{E}[YZ] \leq V_{g_{\varepsilon}}(Y) \quad \forall Y \in \mathcal{L}^{\bar{g}}\right\},$$

again noting that $\mathbb{E}[(X - \mu)Z] = \mathbb{E}[XZ] - \mu$ since $\mathbb{E}[Z] = 1$. Also, we may substitute $Y = X - \mu$, where $\mu = 0$.

We still need to establish that the representation in (16) holds. It follows from Arai (2010, Theorem 1); note that their strictness assumption on the Young function is not needed for this. To apply the result, it remains to show that $R_{g,\varepsilon}$ has the "order lower-semicontinuous" property (Biagini & Frittelli, 2009). But this follows from the fact that we have before established the envelope representation (13), which expresses $R_{g,\varepsilon}$ as a pointwise supremum over a family of order lower-semicontinuous functionals. Thus it is itself order lower-semicontinuous (Biagini & Frittelli, 2009, Section 3.1); note that all Orlicz Banach lattices are order separable (Biagini & Frittelli, 2009, Remark 16). $\qquad\square$

### A.2 Rearrangement Invariant Banach Norms and Fundamental Functions

#### A.2.1 Proof of Proposition 4.11

*Proof.* Due to nonnegativity of indicator functions, the fundamental function coincides with the fundamental function of the Orlicz norm $||\cdot||_{\bar{g}}^O$ (due to Proposition 3.12). It is known (Kosmol & Müller-Wichards, 2011, Lemma 7.4.4) that this fundamental function is $\phi(t) = t\bar{f}^{-1}(1/t)$; since $1/t \geq 1$, the choice of $\bar{f}$ or $f$ does not matter; note that we defined $f^{-1}(y) := \sup\{x : f(x) < y\}$. To intuit the result, we give a non-rigorous "computation". Let $E$ be an arbitrary set of measure $t$:

$$\phi_{V_g}(t) = V_g(\chi_E) = \sup\left\{\int_0^1 \chi_E(\omega)Z^*(\omega)\,\mathrm{d}\omega : \mathbb{E}f(Z) \leq 1\right\}$$
$$= \sup\{t \cdot z_1 : t \cdot f(z_1) + (1-t) \cdot f(z_0) \leq 1\}.$$

That it is possible to decompose $Z^*$ into $z_1$ and $z_0$ is intuitive, but requires formal justification. Due to the nonnegativity of $f$, clearly $z_0 = 0$ is the best choice. For $t \cdot z_1$ to be as large as possible, we then need $t \cdot f(z_1) = 1$ (recall the left-continuity of $f$), hence $z_1 = f^{-1}(1/t)$. Thus $\phi_{V_g}(t) = tf^{-1}\left(\frac{1}{t}\right)$. Noting that $\left(\frac{1}{\varepsilon}f(x)\right)^{-1} = f^{-1}(\varepsilon x)$, the fundamental function at different risk aversion levels can be obtained easily.

To see that $\phi_{V_g}$ is concave, note that since $\bar{f}$ is convex and increasing (non-decreasing), $\bar{f}^{-1}$ is concave (Rubshtein et al., 2016, p. 172). Let $a_{\bar{f}} := \sup\{x : \bar{f}(x) = 0\}$. Note that $\bar{f}$ is strictly increasing on $[a_{\bar{f}}, \infty)$ and $\bar{f}^{-1}$ can only have a jump at 0. Then $p \mapsto p(-\bar{f}^{-1}(1/p))$ is convex, due to the perspective transform of a convex function being convex in both arguments, whence it follows that $p \mapsto p\bar{f}^{-1}(1/p)$ is concave on $(0,1]$. The next proposition, A.2.2 shows that it is furthermore continuous at 0 and therefore concave on $[0,1]$. $\square$

#### A.2.2 Proof of Proposition 4.12

*Proof.* FF1: Consider $\phi_{V_g}(t) = t\bar{f}^{-1}\left(\frac{1}{t}\right)$. Hence

$$\lim_{t\downarrow 0}\phi_{V_g}(t) = \lim_{t\downarrow 0}t\bar{f}^{-1}\left(\frac{1}{t}\right) \overset{x=1/t}{=} \lim_{x\to\infty}\frac{\bar{f}^{-1}(x)}{x}.$$

By the supercoercivity assumption on $f$ (equivalently, on $\bar{f}$), we have $\lim_{x\to\infty}\frac{\bar{f}(x)}{x} = \infty$. But then, its inverse $f^{-1}$ (resp. $\bar{f}^{-1}$)) satisfies the mirror property of "anti-supercoercivity": $\lim_{x\to\infty}\frac{\bar{f}^{-1}(x)}{x} = 0$. For the case when $f$ is finite, see (Kosmol & Müller-Wichards, 2011, Lemma 6.1.26). If $f$ is not finite everywhere, then its inverse is bounded; therefore $\lim_{x\to\infty}\frac{\bar{f}^{-1}(x)}{x} = 0$.

It is easy to see that this holds also in the other direction. Assume that $\lim_{t\downarrow 0}\phi_{V_g}(t) = 0$. Then $\bar{f}$ is supercoercive.

It remains to establish that the same property holds for $\phi_{R_g}$. This readily follows from the equivalence of risk and regret: we can apply Theorem 4.22 in Fröhlich & Williamson (2022) and consider the fraction:

$$K'' = \lim_{t\downarrow 0}\frac{\phi_{R_g}(t)}{\phi_{V_g}(t)}.$$

If $\phi_{V_g}$ was continuous at 0, but $\phi_{R_g}$ was not, then $K'' = \infty$ and then it follows that the norms would not be equivalent.

FF2: We need to show that if $f$ is finite then $\phi'(0) = \lim_{t\downarrow 0}\frac{\phi(t)}{t} = \infty$ and otherwise $\phi'(0) < \infty$. We begin again by considering $\phi_{V_g}$. From the definition of the difference quotient we get:

$$\phi'_{V_g}(0) = \lim_{t\downarrow 0}\frac{\phi_{V_g}(t)}{t},$$

and further

$$\lim_{t \downarrow 0} \frac{\phi_{V_g}(t)}{t} = \lim_{t \downarrow 0} \bar{f}^{-1}\left(\frac{1}{t}\right) = \lim_{x \to \infty} \bar{f}^{-1}(x).$$

If $f$ is finite, then so is $\bar{f}$ and its inverse is unbounded and therefore we obtain $\phi'_{V_g}(0) = \infty$. If $f(x) = \infty$ for some $x \in \mathbb{R}^+$, then its inverse is bounded; consequently, $\phi'_{V_g}(0) < \infty$.

We also have $\phi'_{V_g}(0) = \infty \Leftrightarrow \phi'_{R_g}(0) = \infty$, that is, the fundamental function of the risk measure shares the same property. This readily follows by applying Fröhlich & Williamson (2022, Corollary 4.27).

$\square$

### A.2.3 Proof of Proposition 4.15

*Proof.* First, we observe that there exists $t_0 < 1$ such that $\phi_{V_g}(t_0) = t_0 f^{-1}(1/t_0) = 1$, unless $f$ is the pathological divergence function corresponding to the expectation. This is due to $f^{-1}(1) > 1$ (unless $\lim_{x \downarrow 1} f(x) = \infty$). Since the fundamental function is increasing and continuous, it follows that there exists $t_0 < 1$ such that $\phi_{V_g}(t_0) = t_0 f^{-1}(1/t_0) = 1$.

Let $E$ be an arbitrary measurable set of measure $t$. The fundamental function is then:

$$\phi_{R_g}(t) = R_g(\chi_E) = \sup\left\{\int_0^1 \chi_E(\omega) Z^*(\omega) \, d\omega : Z \geq 0, \mathbb{E}[Z] = 1, \mathbb{E}f(Z) \leq 1\right\}.$$

Next, observe that for $t \leq t_0$ the following dual variable $f^{-1}(1/t)$ is feasible:

$$Z(\omega) = \begin{cases} 1/t & \omega \in E \\ \frac{1 - tf^{-1}(1/t)}{1-t} & \omega \notin E. \end{cases}$$

Since $Z \geq 0$, $\mathbb{E}[Z] = 1$ (by construction) and furthermore, noting the left-continuity of $f^{-1}$:

$$\mathbb{E}f(Z) = t \cdot f(f^{-1}(1/t)) + (1-t) \cdot f\left(\frac{1 - tf^{-1}(1/t)}{1-t}\right) \leq 1.$$

This holds since if $0 < t < 1$ and $f^{-1}(1/t) > 1$ and $tf^{-1}(1/t) \leq 1$ (since $t \leq t_0$), we have

$$0 \leq \underbrace{\frac{1 - tf^{-1}(1/t)}{1-t}}_{=:\star} \leq 1,$$

and because $f$ is by assumption a Young divergence, $f(\star) = 0$. Thus we have shown that this $Z$ is a feasible dual variable. With this $Z$, we obtain that $\phi_{R_g}(t) \geq tf^{-1}(1/t)$ for $t \leq t_0$. But this is just the fundamental function of the corresponding regret measure. Since it holds that $R_g(X) \leq V_g(X)$ due to the infimal convolution (this is also clear from the dual representations), we find that $\phi_{R_g}(t) = tf^{-1}(1/t)$ for $t \leq t_0$.

Due to translation equivariance it holds that $\phi_{R_g}(t) \leq 1 \ \forall t$ and $\phi_{R_g}(1) = 1$. Thus we obtain the statement $\phi_{R_g}(t) = \min\{1, tf^{-1}(1/t)\}$.

Finally, in the special case of the expectation, $R_g(X) = \mathbb{E}[X]$, where

$$f(x) = \begin{cases} 0 & 0 \leq x \leq 1 \\ \infty & x > 1, \end{cases}$$

we obtain $f^{-1}(1/t) = 1$ for $0 < t \leq 1$ and hence the formula holds and reduces to $\phi_{\mathbb{E}}(t) = t$. $\square$

### A.2.4 Proof of Proposition 4.19

*Proof.* The fundamental function of an Orlicz norm $|| \cdot ||_\Psi^O$ with conjugate Young function $\Phi$ is (Kosmol & Müller-Wichards, 2011, Lemma 7.4.4):

$$\phi(t) = t\Phi^{-1}(1/t), \quad t \in (0, 1].$$

Due to properties of the perspective transform, it is guaranteed that the resulting $\Phi^{-1}$ is concave and increasing on $[\Phi^{-1}(1), \infty)$: observe that

$$\Phi^{-1}(x) = x\phi(1/x), \quad x \geq 1.$$

If $\phi$ is concave, $-\phi$ is convex and then its perspective transform $(s, x) \mapsto x(-\phi)(s/x)$ is convex in both arguments (this is well known). By setting $s = 1$ we see that $x \mapsto -\Phi^{-1}(x)$ is convex, hence $\Phi^{-1}$ is concave.

To see that $\Phi^{-1}$ is increasing on $[\Phi^{-1}(1), \infty)$, note that $t \mapsto \phi(t)/t$ is decreasing due to quasiconcavity, hence $t \mapsto \Phi^{-1}(1/t)$ is decreasing and therefore $x \mapsto \Phi(x)$ is increasing.

The condition that $\phi$ is continuous at 0 is indeed necessary for $\Phi$ to be supercoercive, cf. A.2.2.

Also, if $\phi'(0)$ is unbounded, $\Phi$ is finite and the behaviour of $\Phi$ is fully specified for all $x \geq \Phi^{-1}(1)$. If in contrast $\phi'(0) < \infty$, we have a bounded $\Phi^{-1}$ and can therefore identify the behaviour of $\Phi$ for $\Phi^{-1}(1) \leq x \leq b_\Phi$, where $b_\Phi = \sup\{\Phi < \infty\} = \phi'(0)$. Thus such $\Phi$ is infinite for all $x > b_\Phi$.

However, for $0 \leq x < \Phi^{-1}(1)$, we cannot identify the behaviour of $f$. Consequently, any left-continuous increasing convex "completion" of $f$ in this interval will be compatible with the fundamental function. To see that at least one such choice of $f$ must exist, take $\phi$ and extend it it to the whole nonnegative domain by:

$$\phi^\dagger(t) = \begin{cases} \phi(t) & 0 \leq t \leq 1 \\ h(t) & t > 1, \end{cases}$$

in such a manner that $\phi^\dagger : \mathbb{R}^+ \to \mathbb{R}^+$ is concave; such a $\phi^\dagger$ obviously exists: for instance choose a linear continuation with slope $\Phi'(1)$, shifted by $\Phi(1)$, i.e. $h(t) = \Phi(1) + \Phi'(1)(t - 1)$. Then using the equality

$$t\Phi^{-1}(1/t) = \phi^\dagger(t),$$

we can construct an Orlicz norm with the desired properties. Repeating the above arguments then yields that $\Phi$ is a legitimate Young function, where we explicitly set $\Phi(0) = 0$.

$\square$

## A.3 Tail-Specific Marcinkiewicz and Lorentz Norms

### A.3.1 Proof of Proposition 5.6

*Proof.* From (Kreĭn et al., 1982, Lemma 5.3, Theorem 5.3)[17]. we know that M1 to M3 are equivalent for the case of an infinite measure space, i.e. $P(\Omega) = \infty$. First assume that we knew that M1 to M3 are also equivalent for $P(\Omega) = 1$. Then, M3 $\implies$ M4: let $||t \mapsto \phi_{\mathcal{X}'}(t)/t||_{M_\phi} < \infty$. Consequently:

$$||t \mapsto \phi_{\mathcal{X}'}(t)/t||_{M_\phi} = ||E_\mathcal{X}(t)||_{M_\phi} = \sup_{0<t\leq 1} \frac{1}{E_\mathcal{X}(t)} E_\mathcal{X}^{**}(t) < \infty \implies \exists K : E_\mathcal{X}^{**}(t) \leq K \cdot E_\mathcal{X}(t)$$

Also, as a general fact, the decreasing rearrangement is dominated by the maximal function, i.e. $E_\mathcal{X}(t) \leq E_\mathcal{X}^{**}(t)$, $\forall t \in (0, 1]$. Conversely, if $E_\mathcal{X}$ and $E_\mathcal{X}^{**}$ are equivalent, M5 holds (Rubshtein et al., 2016, p. 162). Finally, if M5 holds, then M3 obviously holds by definition of $|| \cdot ||_{M_{\phi_{\mathrm{CVar}}}}$:

$$||E_\mathcal{X}||_{M_\phi} \leq K \cdot ||E_\mathcal{X}||_{M_{\phi_{\mathrm{CVar}}}} = K \cdot \sup_{0<t\leq 1} \frac{1}{E_\mathcal{X}^{**}(t)} E_\mathcal{X}^{**}(t) = 1 < \infty.$$

---

[17]To prevent confusion, we remark that Kreĭn et al. (1982) use the symbol $M_\phi$ to denote what in our notation is $M_{\phi_{\mathcal{X}'}}$, that is, we index the Marcinkiewicz norm by its fundamental function, whereas Kreĭn et al. (1982) index it by the fundamental function of its associate space.

Observing that $\phi_{\mathcal{X}'}(t)/t = E_{\mathcal{X}}(t)$, the equivalence of M2 and M3, for the case of both $P(\Omega) = 1$ or $P(\Omega) = \infty$, can be obtained from Pick et al. (2013, p. 265). In summary, we have shown that the implications form a circle.

We shall now show that $M1$ to $M3$ are also equivalent in the case of $P(\Omega) = 1$. Assume without loss of generality that $\phi_{\mathcal{X}}(1) = \phi_{\mathcal{X}'}(1) = 1$ (this corresponds to simply rescaling one norm and then using the associate relation of the fundamental functions). Consider the following extension of $\phi_{\mathcal{X}'} : [0, \infty) \to [0, \infty)$ to the whole nonnegative domain:

$$\tilde{\phi}_{\mathcal{X}'}(t) = \begin{cases} \phi_{\mathcal{X}'}(t) & 0 \le t \le 1 \\ 1 + \phi'_{\mathcal{X}'}(1)(t - 1) & 1 < t < \infty, \end{cases}$$

which extends $\phi_{\mathcal{X}'}$ as a straight line and where the slope is chosen so as to guarantee concavity (assume without loss of generality that $\phi_{\mathcal{X}}$ is concave, otherwise first use the least concave majorant to concavify the quasi-concave $\phi_{\mathcal{X}'}$, thereby preserving equivalence of Marcinkiewicz norms). Let $\tilde{\phi}(t) = t/\tilde{\phi}_{\mathcal{X}}(t)$. We now show the following:

**Lemma A.2.** $|\cdot|_{M_\phi}$ and $||\cdot||_{M_\phi}$ are equivalent on $\Omega = [0, 1]$ if and only if $|\cdot|_{M_{\tilde{\phi}}}$ and $||\cdot||_{M_{\tilde{\phi}}}$ are equivalent on $\Omega = [0, \infty)$.

The Marcinkiewicz norm in the case of $P(\Omega) = \infty$ is defined in the obvious way, as

$$||X||_{M_{\tilde{\phi}}} := \sup_{0 < t < \infty} \left\{ \tilde{\phi}(t) X^{**}(t) \right\}$$

and $X^* : \mathbb{R}^+ \to \mathbb{R}^+$ and $X^{**} : (0, \infty) \to \mathbb{R}^+$ are defined in the obvious way.

*Proof of Lemma*: First assume that $|\cdot|_{M_{\tilde{\phi}}}$ and $||\cdot||_{M_{\tilde{\phi}}}$ are equivalent on $\Omega = [0, \infty)$. Then it is trivial that $|\cdot|_{M_\phi}$ and $||\cdot||_{M_\phi}$ are equivalent on $\Omega = [0, 1]$, by simply extending a random variable $X \in M_\phi$ from $\Omega = [0, 1]$ to $\Omega = [0, \infty)$ as $X^*(t) = 0$ for $t > 1$. For the converse direction, assume that $|\cdot|_{M_\phi}$ and $||\cdot||_{M_\phi}$ are equivalent on $\Omega = [0, 1]$. Then we want to show that $|\cdot|_{M_{\tilde{\phi}}}$ and $||\cdot||_{M_{\tilde{\phi}}}$ are equivalent by demonstrating that $t \mapsto \tilde{\phi}_{\mathcal{X}'}(t)/t = 1/\tilde{\phi}(t) \in M_{\tilde{\phi}}$, which is equivalent to the desired statement due to Pick et al. (2013, p. 265). Then:

$$||t \mapsto \tilde{\phi}_{\mathcal{X}'}(t)/t||_{M_{\tilde{\phi}}} = \sup_{0 < t < \infty} \tilde{\phi}(t) \frac{1}{t} \int_0^t 1/\tilde{\phi}_{\mathcal{X}'}(\omega) \, d\omega \overset{!}{<} \infty.$$

By assumption, we have that $t \mapsto \phi_{\mathcal{X}'}(t)/t \in M_\phi$ and therefore only have to investigate $t > 1$. But:

$$\star := \sup_{1 < t < \infty} \frac{t}{1 + \phi'_{\mathcal{X}'}(1)(t - 1)} \frac{1}{t} \left( \int_0^1 1/\phi(\omega) \, d\omega + \int_1^t 1/\tilde{\phi}(\omega) \, d\omega \right).$$

Since $t \mapsto \phi_{\mathcal{X}'}(t)/t \in M_\phi$, we have $c := \int_0^1 1/\phi(\omega) \, d\omega \le \sup_{0 < t \le 1} \phi(t) \frac{1}{t} \int_0^t 1/\phi(\omega) \, d\omega < \infty$; essentially this follows from $\mathcal{L}^1$ being the largest of all r.i. spaces. Furthermore:

$$\star = \sup_{1 < t < \infty} \frac{t}{1 + \phi'_{\mathcal{X}'}(1)(t - 1)} \frac{1}{t} \left( c + \int_1^t \frac{1 + \phi'_{\mathcal{X}'}(1)(\omega - 1)}{\omega} \, d\omega \right)$$

$$= \sup_{1 < t < \infty} \frac{t}{1 + \phi'_{\mathcal{X}'}(1)(t - 1)} \frac{1}{t} \left( c + \phi'_{\mathcal{X}'}(1)t - \phi'_{\mathcal{X}'}(1) \log(t) - \phi'_{\mathcal{X}'}(1) + \log(t) \right) \overset{!}{<} \infty.$$

To ensure that this is finite, we need to check only the limit as $t \to \infty$:

$$\lim_{t \to \infty} \frac{\phi'_{\mathcal{X}'}(1)t - \phi'_{\mathcal{X}'}(1) \log(t) - \phi'_{\mathcal{X}'}(1) + \log(t)}{1 + \phi'_{\mathcal{X}'}(1)(t - 1)} = 1.$$

Thus we have that $t \mapsto \tilde{\phi}_{\mathcal{X}'}(t)/t = 1/\tilde{\phi} \in M_{\tilde{\phi}}$, which completes the proof of the lemma. Finally, we need to show the following:

**Lemma A.3.** $\forall s > 1$: $\sup_{0<t\leq 1} \frac{\phi_{\mathcal{X}'}(t/s)}{\phi_{\mathcal{X}'}(t)} < 1$ *if and only if* $\forall s > 1$: $\sup_{0<t<\infty} \frac{\tilde{\phi}_{\mathcal{X}'}(t/s)}{\tilde{\phi}_{\mathcal{X}'}(t)} < 1$.

*Proof of Lemma*: That the condition on $\tilde{\phi}_{\mathcal{X}'}$ implies the condition of $\phi_{\mathcal{X}'}$ is obvious. Conversely, assume the condition holds for $\phi_{\mathcal{X}'}$ on $\Omega = [0,1]$. Then we need to show only that it also holds for $\tilde{\phi}_{\mathcal{X}'}$ when $1 < t < \infty$, as they coincide for $0 \leq t \leq 1$. Let $s > 1$ be arbitrary. For any fixed $1 < t$, we obviously have that the fraction is $< 1$ and thus need to investigate only the limit as $t \to \infty$. Then:

$$\lim_{t\to\infty} \frac{1 + \phi'_{\mathcal{X}'}(1)(t/s - 1)}{1 + \phi'_{\mathcal{X}'}(1)(t - 1)} = \frac{1}{s} < 1.$$

Chaining all equivalences together, we obtain that M1 to M5 are equivalent for both finite and infinite measure spaces. $\qquad\square$

### A.3.2 Fundamental Functions of $\mathrm{L_{exp}}$ and $\mathrm{LlogL}$

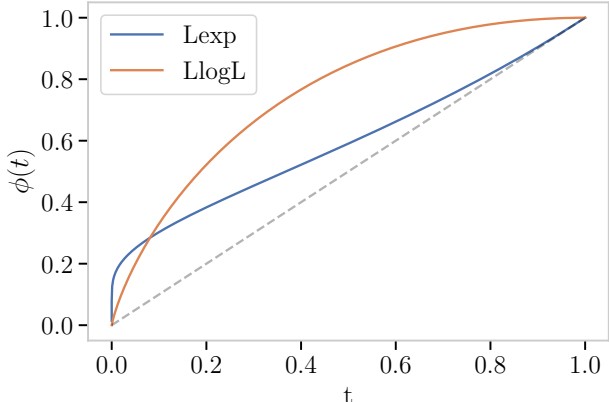

Figure 2: Fundamental functions of $\mathrm{L_{exp}}$ and $\mathrm{LlogL}$.

The fundamental function of $\mathrm{L_{exp}}$, given by $\phi_{\mathrm{L_{exp}}}(t) = 1/(1 - \log(t))$, is only quasiconcave, not concave. In contrast, the fundamental function $\phi_{\mathrm{LlogL}}(t) = t + t\log(1/t) = t/\phi_{\mathrm{L_{exp}}}(t)$ is concave; see Figure 2.

### A.3.3 Proof of Proposition 5.11

Assume $Y^*$ is differentiable on $(0,1)$. Then the condition that $\phi_{\mathcal{X}'}(t) = Y^*(t) \cdot t = F_X^{-1}(1-t) \cdot t$ is increasing becomes

$$\frac{\mathrm{d}}{\mathrm{d}t}(F_X^{-1}(1-t) \cdot t) > 0 \Leftrightarrow \frac{\mathrm{d}}{\mathrm{d}s}F_X^{-1}(s) \leq \frac{1}{1-s}F_X^{-1}(s) \quad \forall s \in (0,1).$$

We use a Gronwall-type inequality.

**Proposition A.4** (adapted from Bainov & Simeonov (2013, Lemma 1.1)). *Let $f : [0,\infty)$ be continuous and differentiable and $g : [0,\infty)$ be continuous; assume $f(0) \leq f_0$ and*

$$\frac{\mathrm{d}}{\mathrm{d}s}f(s) \leq g(s)f(s), \quad \forall s \geq 0.$$

*Then:*

$$f(s) \leq f_0 \exp\left(\int_0^s g(r)\,\mathrm{d}r\right) \quad \forall s \geq 0.$$

Applying this statement with $f(s) = F_X^{-1}(s)$, $f_0 := F_X^{-1}(0)$ and $g(s) = 1/(1-s)$ and constraining the domain yields

$$F_X^{-1}(s) \leq F_X^{-1}(0) \exp\left(\int_0^s \frac{1}{1-r}\,\mathrm{d}r\right) \quad \forall s \in (0,1)$$

$$\Leftrightarrow F_X^{-1}(s) \leq F_X^{-1}(0) \exp\left(-\log(1-s)\right) \quad \forall s \in (0,1)$$

$$\Leftrightarrow F_X^{-1}(s) \leq F_X^{-1}(0)\frac{1}{1-s} \quad \forall s \in (0,1)$$

$$\Leftrightarrow Y^*(t) \leq Y^*(1)\frac{1}{t} \quad \forall s \in (0,1).$$

### A.4 Utility-Based Shortfall Risk

**Definition A.5** (Föllmer & Schied (2002; 2016)). *A convex risk measure on some space $\mathcal{X}$ is a functional $R : \mathcal{X} \to \mathbb{R} \cup \{\infty\}$, which is monotone (C3), translation equivariant (C4) and convex:*

$$R(\alpha X + (1-\alpha)Y) \leq \alpha R(X) + (1-\alpha)R(Y).$$

Typically, also $R(0) = 0$ is demanded. Positive homogeneity and subadditivity together imply convexity; but convexity on its own is a weaker condition. Thus a coherent risk measure is also a convex risk measure.

In this section, we define a loss function $\ell : \mathbb{R} \to \mathbb{R}^+$ to be a function which is a Young function on $\mathbb{R}^+$ and further satisfies $\ell(x) = 0$ for $x \leq 0$. Consider the *acceptance set*:

$$\mathcal{A} = \{X : \mathbb{E}[\ell(X)] \leq x_0\}.$$

The intuition is that a decision maker who has a loss function and some decision threshold $x_0$ considers a loss variable $X$ "acceptable" if $\mathbb{E}[\ell(X)] \leq x_0$. Subsequently, let $x_0 = 1$. We assumed $\ell(x) = 0$ for $x \leq 0$, which is non-standard, but allows for a cleaner connection to Orlicz space theory. Intuitively, this implies that acceptable random variables are characterized by their losses only, whereas gains (negative values) are neglected in the acceptance set. This also fits with our overall focus on right tails only. However, the trade-off between losses and gains naturally arises by enforcing translation equivariance. A convex risk measure, the UBSR, can be constructed as follows:

$$\mathrm{UBSR}_\ell(X) = \inf\{m \in \mathbb{R} : X - m \in \mathcal{A}\}.$$

Hence, $\mathrm{UBSR}_\ell$ specifies the smallest amount that must be subtracted from the risky position $X$ so as to make it acceptable. This construction automatically guarantees translation equivariance and since $\ell$ is increasing and convex, $\mathrm{UBSR}_\ell$ is a convex risk measure (Föllmer & Schied, 2002).

To study the envelope representation of $\mathrm{UBSR}_\ell$, one must choose a pair of Banach spaces. Föllmer & Schied (2002) studied the UBSR on $\mathcal{L}^\infty$, which is an overly conservative choice. In fact, the natural setting is the framework of Orlicz spaces, as shown by Biagini & Frittelli (2009) and Arai (2011). We denote the conjugate of $\ell$ as $\ell^*$. Observe that $\ell^*$ is also a legitimate Young function[18] on $\mathbb{R}^+$, which is supercoercive[19] and $\ell^*(x) = \infty$ for $x < 0$. To state the envelope representation of $\mathrm{UBSR}_\ell$ on its natural space, we need to define the *Orlicz class*.

**Definition A.6.** *Given a Young function $\Phi$, the* Orlicz class $P^\Phi$ *is defined as*

$$P^\Phi := \{X \in \mathcal{M} : \mathbb{E}[\Phi(|X|)] < \infty\}.$$

*By definition of the Luxemburg norm, we obviously have $P^\Phi \subseteq \mathcal{L}^\Phi$. We remark that $P^\Phi$ need not be a linear subspace of the Orlicz space $\mathcal{L}^\Phi$, but it does contain a linear subspace itself, called the Orlicz heart $M^\Phi$. However, if $\Phi$ satisfies the $\Delta_2$ condition (Kosmol & Müller-Wichards, 2011, p. 197, p. 234), then in fact $M^\Phi = P^\Phi = \mathcal{L}^\Phi$. Intuitively, such $\Phi$ cannot grow too fast. For instance, the rapidly growing $\Phi(x) = (\exp(x) - 1)\chi_{[1,\infty)}$ does* not *satisfy $\Delta_2$.*

---

[18]For this, our assumption that $\ell(y) = 0$ for $y \leq 0$ is important, then $\ell^*$ is increasing on $[0,\infty)$. Confer (Föllmer & Schied, 2002, Lemma 11).

[19]Supercoercive means that $\lim_{y\to\infty} \ell^*(y)/y = \infty$, cf. Definition 3.1 and see (Föllmer & Schied, 2002).

Indeed, the UBSR is finite on the Orlicz class $P^\ell$ (consider $m = 0$); here, understand $\ell$ as its restriction to $\mathbb{R}^+$, where it is a Young function.

**Proposition A.7** (Föllmer & Schied (2002); Biagini & Frittelli (2009))**.** *The following envelope representation holds for the utility-based shortfall risk measure* $\mathrm{UBSR}_\ell : P^\ell \to \mathbb{R}$:

$$\mathrm{UBSR}_\ell(X) = \sup\{\mathbb{E}[XZ] - \alpha(Z) : Z \in \mathcal{L}_+^{\ell^*}, Z \geq 0, \mathbb{E}[Z] = 1\},$$

*where the penalty function* $\alpha : \mathcal{L}_+^{\ell^*} \to \mathbb{R}$ *is:*

$$\alpha(Z) = \sup\{\mathbb{E}[XZ] : X \in \mathcal{L}^\ell, X \in \mathcal{A}\} = \sup\{\mathbb{E}[XZ] : X \in \mathcal{L}^\ell, X \geq 0, \mathbb{E}[\ell(X)] \leq 1\}. \quad (18)$$

In fact, we observe that $\alpha(Z) = ||Z||_{\ell^*}^O$ as $Z \geq 0$, i.e. the penalty function is an Orlicz norm, and therefore:

$$\mathrm{UBSR}_\ell(X) = \sup\left\{ \mathbb{E}[XZ] - \inf_{t>0} t(1 + \mathbb{E}[\ell^*(Z/t)] : Z \in \mathcal{L}^{\ell^*}, Z \geq 0, \mathbb{E}[Z] = 1 \right\}. \quad (19)$$

*Proof.* We adapt Proposition 32 in (Biagini & Frittelli, 2009) to our loss-based orientation and to the additional assumption that $\ell(x) = 0$ for $x \leq 0$. By definition, $\alpha(Z) = ||Z||_{\ell^*}^O$ is finite on $\mathcal{L}^{\ell^*}$, but $\alpha(Z)$ is only evaluated for $Z \geq 0$, hence we take $\mathcal{L}_+^{\ell^*}$ as its domain to emphasize this. The restriction to $X \geq 0$ in (18) is possible since $Z \geq 0$ and $\ell(x) = 0$ for $x \leq 0$, hence a negative part of $X$ is neglected. $\quad\square$

Föllmer & Schied (2002), who established a variant (19) in the $\mathcal{L}^\infty$ setting, remarked in passing the similarity to the Amemiya formulation of an Orlicz norm, but did not further go into it.

Among the convex risk measures, the utility-based shortfall risk measure has particularly desirable properties, such as invariance under randomization (Giesecke et al., 2008) and elicitability (Bellini & Bignozzi, 2015). Most important in the present context, however, is that since the penalty function is an Orlicz norm, we can immediately apply our method of Section 6 to achieve a desired tail sensitivity. In contrast to a coherent risk measure, a convex risk measure considers *all* alternative probability measures $Q_Z$ from an entire space, but they are considered more or less relevant due to the penalty function; note that $Z \geq 0$ and $\mathbb{E}[Z] = 1$, hence the $Q_Z$ induced via (14) are valid probability measures. If the penalty function only takes on the values 0 and $+\infty$, the risk measure is coherent. In this sense, a convex risk measure acts in a smoother fashion than a coherent risk measure, where only a subset of probability measures is considered at all. By specifying the tail sensitivity of the penalty function $||Z||_{\ell^*}^O$, we specify how alternative probability measures, conceptualized as density ratios with respect to the base measure, are punished for their reweighting in a tail-sensitive way.

In the context of machine learning, the UBSR seems particularly useful to handle tail risk in reinforcement learning applications, as exemplified by Shen et al. (2014). More generally, any context which calls for a risk measure without positive homogeneity is a possible application for the UBSR.

**Example A.8.** *(Föllmer & Schied, 2002) Let* $1 < p \leq \infty$ *and* $\ell(x) = (\frac{1}{p}x^p) \cdot \chi_{[0,\infty)}$. *Then*

$$\ell^*(y) = \begin{cases} \infty & y < 0 \\ (\frac{1}{q}y^q) & y \geq 0, \end{cases}$$

*and*

$$\alpha(Z) = p^{1/p} \left(\mathbb{E}[Z^q]\right)^{1/q}.$$

*Consider* $p \to \infty$. *Then* $\ell^*(y) = i_{[0,1]}$ *and therefore* $\mathrm{UBSR}_\ell(X) = \mathbb{E}[X]$, *since any Z which has* $Z(\omega) > 1$ *for some $\omega$ is infinitely punished; this is just the* $\mathcal{L}^1 - \mathcal{L}^\infty$ *associate relationship.*

### A.5 Plain Language Summary and Diagram

When training machine learning systems with data, engineers typically try to minimize average prediction error. Yet the justification for why the average is the best summary of those errors is weak: some errors could be very high, some very low. When a prediction error corresponds to a human individual (for example when we predict who gets a loan), this is problematic, as some individuals then suffer a much higher error than others. In such a situation, we would like to have a summary which also penalizes inequality of prediction errors. To this end, we can minimize a "risk measure" of the prediction errors. A risk measure is like an average, but puts more weight on the higher prediction errors. Many researchers have studied risk measures and proposed many different families of risk measures. But one important aspect, which has not been studied systematically yet, is how sensitive risk measures react to very large errors. This is called the tail sensitivity and it is the focus of our work. We study risk measures in a unified framework, which was proposed by mathematicians. In this framework, we find that there exists a simple way of characterizing how tail sensitive a risk measure is. An outcome of our work is a method to generate risk measures which have exactly a desired tail sensitivity. This can be used by a machine learning engineer to flexibly tune the influence of very high prediction errors.

Figure 3: Summary of key elements in the paper.

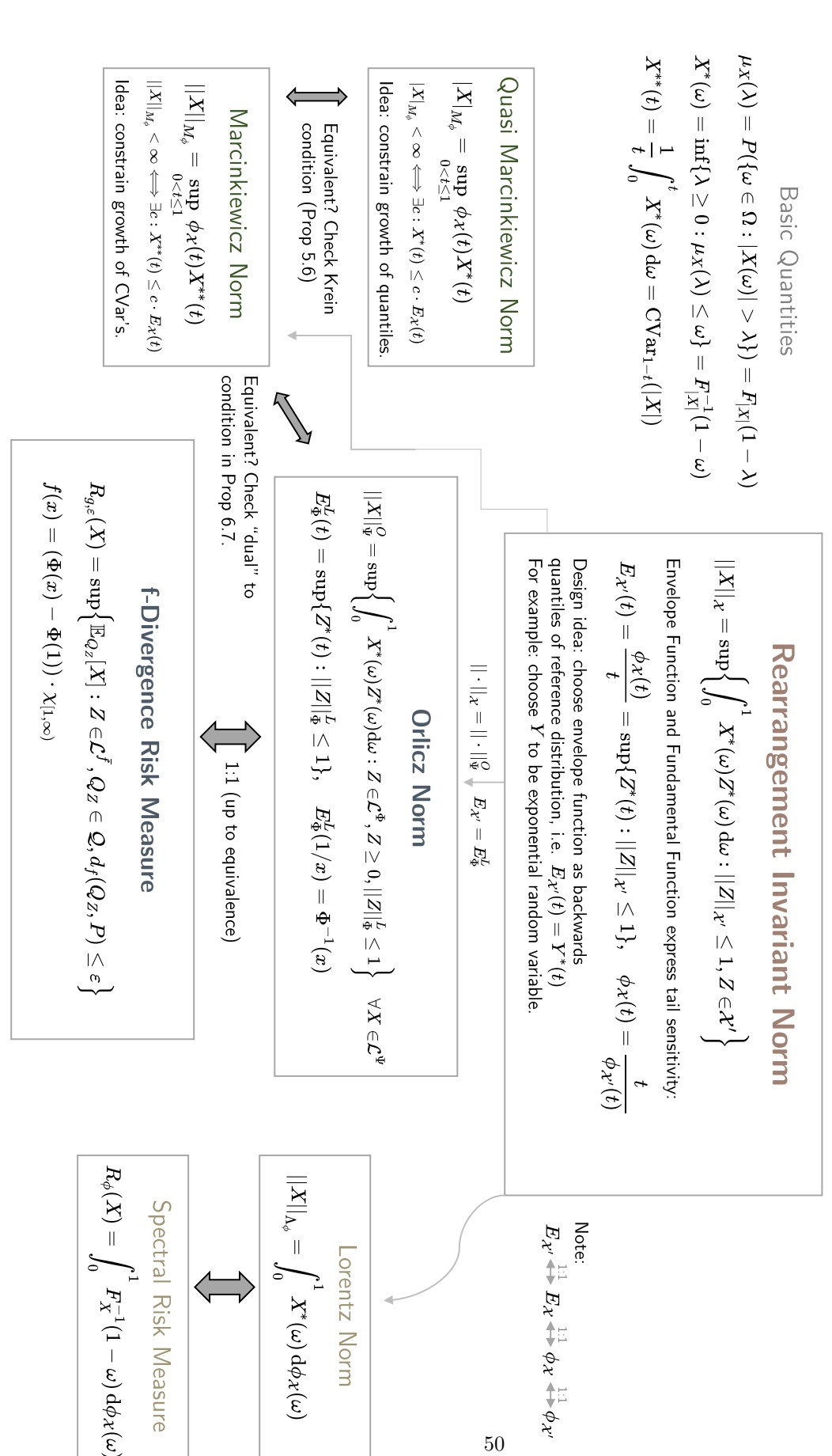

