# OpenReview forum: "Tailoring to the Tails: Risk Measures for Fine-Grained Tail Sensitivity"
_TMLR — Accepted by TMLR_

### Review · Reviewer_2ucL · 2022-09-17

**Summary Of Contributions:**

In this work the authors propose a general approach to tune tail sensitivity of a risk measure and add this concept to empirical risk minimisation.
They also provide a connection between tail sensitivity and fundamental functions. For example, the Orlicz regret measures are proposed and it is shown that its extension is f-divergence risk measures.

**Broader Impact Concerns:**

This submission does not require a Broader Impact Statement.

**Requested Changes:**

I propose the following changes:

- **(Critical)** The authors must compare the implementation of their risk measure to ERM in a straightforward example or experiment.
- **(Critical)** I know there are some applications of the proposed risk measures in finance, but I prefer the applications in machine learning or statistical learning. For instance, why should someone care about the tail sensitivity in machine learning? The authors should make clear the applications of their proposed risk measures in machine learning or statistical learning. When should the risk measure be designed depending on our scenario?
- **(non-critical)** How can we evaluate the robustness of your suggested risk measures in light of label noise or outlier?
- **(non-critical)** Is it possible to analysis the generalisation error or excess risk under the proposed risk measures for a supervised learning scenario?
- **(non-critical)** As the paper is too long, I propose to add a summary page or table to summarise your result in that page or table.


**Strengths And Weaknesses:**

**Strengths**
--
- This work provides new risk measure framework which extend the previous risk measures
- Some of the discussions of this paper is interesting for me, e.g., the section 7 discussion is really interesting.

**Weaknesses**
--
- There is *no experiment* in this work to show that their risk measures can improve the some metrics in machine learning problems in comparison with empirical risk minimisation.
- The *application of risk measures* in machine learning is not clarified.
- Some related works are not discussed, e.g., [1], [2].



**References**
--
- [1]: Föllmer, Hans, and Thomas Knispel. "Entropic risk measures: Coherence vs. convexity, model ambiguity and robust large deviations." Stochastics and Dynamics 11.02n03 (2011): 333-351.
- [2]: Li, Tian, et al. "Tilted empirical risk minimization." arXiv preprint arXiv:2007.01162 (2020).

---

> ### Author Response · Authors · 2022-10-24
> **Reviewer 2ucL**
>
> We thank reviewer 2ucL for their engagement, comments and criticisms. Specifically, we are glad that reviewer 2ucL found section 7 on "Tail-Specific Orlicz Deviation Inequalities" interesting. We think that this section has the potential to be very helpful to the machine learning community, as it presents a unified viewpoint on concentration inequalities.
>
> As to the weaknesses and requested changes:
> > There is no experiment in this work to show that their risk measures can improve the some metrics in machine learning problems in comparison with empirical risk minimisation.
> > [..] The authors must compare the implementation of their risk measure to ERM in a straightforward example or experiment.
>
> > The application of risk measures in machine learning is not clarified.
> > [..] why should someone care about the tail sensitivity in machine learning?
>
> There is already ample evidence in the machine learning literature about the usefulness of risk measures and their feasibility. We agree with reviewer 2ucL that we had not made this explicit enough, since we ourselves focus on theoretical understanding without conducting experiments ourselves. We have added a short section "1.3 - Use in Machine Learning" to the revised version, which we believe now makes the application of risk measures to machine learning much clearer. We there refer to empirical evidence of their usefulness. Our work starts with the assumption that risk measures *have already been proven useful*. Our goal in the paper is to provide more intuition and understanding about the implications of a particular choice of risk measure, specifically with respect to tail behaviour.
>
> > Some related works are not discussed, e.g., [1], [2].
>
> We agree that [1] is a relevant reference. Here,
> the coherent entropic risk measure is introduced, which is the Kullback-Leibler special case of the f-divergence risk measure. We added this to our related work section. As to tilted empirical risk minimization (TERM) in [2], we were not aware of this work and are thankful for learning about it. Indeed, what the authors of [2] study is an empirical variant of the convex entropic risk measure in [1]; with a subtlety regarding negative values of t, which corresponds to the concept of a lower prevision in imprecise probability. Furthermore, TERM is a special case of the utility-based shortfall risk measure (UBSR) with the choice of function f(x)=exp(t x). This provides a neat argument for our discussion of the UBSR in section 6.1 / appendix A.4. In our revised version, we discuss the work of [2] in Section 6.1 and mention it in the related work section.
>
> > How can we evaluate the robustness of your suggested risk measures in light of label noise or outlier?
> > Is it possible to analysis the generalisation error or excess risk under the proposed risk measures for a supervised learning scenario?
>
> We agree that both these questions are very interesting and relevant. However, the paper is already dense and focuses on the theoretical aspects of the risk measures. There many many questions which remain regarding practical considerations. We wish to study these questions in detail in a follow-up paper. For the special case of CVar (arguably the most interesting spectral risk measure, cf. Kusuoka's representation theorem), work in this direction already exists, cf. for instance [3] and [4].
>
> > As the paper is too long, I propose to add a summary page or table to summarise your result in that page or table.
>
> We agree that the paper is rather long and involved due to its technical nature. In our revised version, we included a plain language language in the appendix and referred to it in the introduction. We hope that this aids the communication of the paper to a wider scientific audience.
>
> [1]: Föllmer, Hans, and Thomas Knispel. "Entropic risk measures: Coherence vs. convexity, model ambiguity and robust large deviations." Stochastics
> and Dynamics 11.02n03 (2011): 333-351.
>
> [2]: Li, Tian, et al. "Tilted empirical risk minimization." arXiv preprint arXiv:2007.01162 (2020).
>
> [3]: Soma, T., & Yoshida, Y. (2020). Statistical learning with conditional value at risk. arXiv preprint arXiv:2002.05826.
>
> [4]: Mhammedi, Z., Guedj, B., & Williamson, R. C. (2020). PAC-bayesian bound for the conditional value at risk. Advances in Neural Information Processing Systems, 33, 17919-17930.

---

### Review · Reviewer_gcQw · 2022-09-28

**Summary Of Contributions:**

This contribution regards the construction of spaces of random variables which inherit specific tail properties. These spaces are Orlicz spaces which are closely related to risk measures derived from f-divergences. The paper relates their norms with the tail properties of the random variables contained in the space. It further provides a deviation inequality, bounding the excess probability with the Luxemburg norm as well as the envelope function.

**Broader Impact Concerns:**

No concerns

**Requested Changes:**

Adress the questions **a.)** - **e.)**

**Strengths And Weaknesses:**

**Strengths**:

a)	The paper is written in a clear and understandable way

b)	The authors provide an interesting connection between the tail behavior of random variables and their Luxemburg norms.


**Weakness**:

The authors start the paper emphasizing the relevance of their contribution with respect to the field of machine learning. They justify this claim by stating that the Average Loss is an insufficient measure of risk as it does not take into account tail properties of the loss. However, afterwards the authors almost solely emphasize on theoretical properties of risk measures, not considering this connection to ML anymore. This leaves important questions unanswered.
Some are:

**a.)** How exactly should the risk measures/norms be included into ML algorithms and how does it change the resulting prediction?

**b.)** What is the benefit of employing the risk measures/norms?

**c.)** Can the presented risk measures/ norms be computed efficiently if there are only samples available?

**d.)** How much computational cost does it add to the ML algorithm?

**e.)** What is the interpretation of small loss tails? Just fitting the observed data?

---

> ### Author Response · Authors · 2022-10-24
> **Response to reviewer gcQw**
>
> We thank reviewer gcQw for engaging with our paper and their assessment of it as "clear and understandable". Reviewer gcQw points out that practical considerations, i.e. how the proposed risk measures can be applied in actual ML systems, were not sufficiently discussed. We agree to some extent. In particular, the following points were raised.
>
> > How exactly should the risk measures/norms be included into ML algorithms and how does it change the resulting prediction?
>
> > What is the benefit of employing the risk measures/norms?
>
> We have added a short section "1.3 - Use in Machine Learning" to the paper, which now answers these questions much clearer than before. We thank reviewer gcQw for raising these questions, and believe that the revised version now gives the reader a better idea about the use of risk measures in machine learning. We refer to the already existing ample empirical evidence on the usefulness of risk measures in machine learning in different use cases: risk-averse decision making, distributional robustness and fairness.
>
> >Can the presented risk measures/ norms be computed efficiently if there
> are only samples available?
>
> >How much computational cost does it add to the ML algorithm?
>
> We have expanded on Section 8 ("Application") in the revised version and now provide more references regarding the practical computation of the studied risk measures. The risk measures on which we focus in the paper, f-divergence risk measures, are indeed easily added to e.g. a pytorch model, since Equation 5 can be implemented in a single line, effectively without adding overhead as only two scalar parameters are optimized. The computation of spectral risk measures is in general more involved, but we now provide the appropriate references. The arguably most important spectral risk measure, the conditional value at risk CVar, is easily implemented. We now also refer to work which provides more sophisticated methods of optimizing CVar in practice, which are more stable in a batch setting.
>
> >What is the interpretation of small loss tails? Just fitting the observed
> data?
>
> 1. Consider a fairness context as in [1], where individual losses correspond to individual humans. Here, small loss tails would mean that the distribution is "fair", i.e. there is low inequality.
> 2. In a reinforcement learning context, a distribution with small loss tails would mean that "catastrophes hardly occur", which would be desirable.
> 3. We remark that the heaviness of the loss tails seems to be intricately related to rates of convergence of learning methods, cf. [2]
>
> [1]: Williamson, R., & Menon, A. (2019, May). Fairness risk measures. In International Conference on Machine Learning (pp. 6786-6797). PMLR.
>
> [2]: Grünwald, P. D., & Mehta, N. A. (2020). Fast Rates for General Unbounded Loss Functions: From ERM to Generalized Bayes. J. Mach. Learn. Res., 21, 56-1.

---

### Review · Reviewer_chCF · 2022-10-16

**Summary Of Contributions:**

The authors propose alternatives to expected risk minimization by introducing risk measures that are tail sensitive. First, they introduce _Orlicz regret measures_, a generalization of Orlicz norms that treat gains and losses differently (Definition 3.8). However, regret measures are not translation invariant, so that authors introduce _divergence risk measures_ (Proposition 3.19). Then, Orlicz regret measures and divergence risk measures are analyzed through the lens of rearrangement Banach norms. In particular, their fundamental functions are derived (Propositions 4.11 and 4.15). The latter are in one-to-one relationship with the envelope functions, that define the tail sensitivity by constraining the tail behavior of the dual variable. In Section 6, the authors provide several examples of Orlicz norms obtained by different choices of envelope functions (Examples 6.3 to 6.6). In Section 7 some Orlicz deviation inequalities are presented, while Section 8 focuses on computing some risk measures.

**Requested Changes:**

**Major**: See _weaknesses_ paragraph above

**Minor:**
- General: I would not number the equation which are not referred to in the text, but it is more a matter of taste
- p.2, 2nd paragraph: "coherent" is not yet defined. At least a pointer to Sec. 3,4 could help
- Def. 2.3: I would emphasize more on the Amemiya norm (even including it in the definition), as it is the one which is used to generalize to Orlicz regret measures
- Prop. 3.2: "and and"
- Def. 3.8: I would add a remark, as $g$ is used here instead of $\Phi$ (an exact analogy would require using $f$). Things are clarified in Prop. 3.11, stating that $g$ is the young function, so the latter could be moved next to the definition for more clarity.
- Prop. 3.9: V5, should be $V_g$ instead of $V$
- Prop. 3.10 and 3.14: the link between these propositions and the rest of the section could be improved
- I find it a bit misleading to introduce risk measures for $g_\varepsilon$. Why not a for generic $g$ and then specifying that the envelope representation changes when considering $g_\varepsilon$
- Eq. (49): $\phi_\mathcal{X}(0) = 0$
- Eq. (104): remove one closing parenthesis in the l.h.s.
- Eq. (128): $V_g$
- Appendix A.2.3.: first line, $t_0$ instead of $t$

**Strengths And Weaknesses:**

**Strengths**
- The purpose of the paper is relevant, and the proposed solution is elegant
- The paper is globally well-written, although it is pretty dense
- It draws interesting connections with fields that could be of interest to the machine learning community
- As far as I could check the maths are neat and correct

**Weaknesses**
- The paper is very dense, and pretty involved to read, due to the introduction of many concepts that most people in the machine learning community will not be aware of. I must say that the latter seems inevitable given the nature of the results provided, and some credit should be given to the authors, who did their best to make it as digest as possible
- One side effect of the above point is that it is sometimes difficult to identify the proper contribution of this work, with respect to previously introduced notions. Again, this seems inevitable, as a lot of context is required, and the authors are extremely clear about what is new, it is more of a byproduct of the nature of the article
- I might missed something, but the authors motivate the introduction of regret measures by the need for considering gains and losses differently. However, in the Expected Risk Minimization framework considered, most loss functions have only positive values, such that the latter seems overly sophisticated. Do the authors have precise examples where regret measures are needed? In addition, the norm induced by $V_g$ (which is studied in the rest of the paper) uses the absolute value and looses the asymmetry, so it was not clear to me why introducing an asymmetric measure was needed
- Another important drawback I see about the proposed risk measures is their computability. In particular, how easy is it to estimate them with finitely many observations? Furthermore, Expected (or Empirical) Risk Minimization is often tackled through gradient descent, which means differentiating through the risk measure: how would this tranpose with the propose risk measures?

---

> ### Author Response · Authors · 2022-10-24
> **Response to reviewer chCF**
>
> We thank reviewer chCF for their thorough engagement of our text and appreciate the points that were raised.
> With respect to the weaknesses which were pointed out:
> > The paper is very dense, and pretty involved to read, due to the introduction of many concepts that most people in the machine learning community will not be aware of. I must say that the latter seems inevitable given the nature of the results provided, and some credit should be given to the authors, who did their best to make it as digest as possible
>
> We are thankful for noting our effort to clearly present dense theoretical material to the machine learning community. We agree that the paper is "pretty involved to read". In our revision, we included a plain language summary of the paper in the appendix and referred to it in the introduction, in the hope that it communicates our work better to a wider audience.
>
> > [..] it is sometimes difficult to identify the proper contribution [..]
>
> Again, we thank reviewer chCF for appreciating our effort to make our contribution as clear as possible. A major aspect of the paper is the synthesis of concepts from different literatures, yet it may not be obvious that this process itself as a contribution.
>
> > [..] it was not clear to me why introducing an asymmetric measure was needed
>
> It is correct that in ERM we typically deal only with nonnegative values. There are (at least) three reasons why one might consider asymmetric regret measures: first, in the risk-sensitive regression as suggested by [1], the asymmetry is needed. Second, when risk measures are applied in the context of reinforcement learning, asymmetry of gains and losses may be important. Third and most relevant to us, the infimal convolution (natural extension, Section 3.5) which gives rise to the f-divergence risk measure requires the asymmetric regret measure to start with, even if later the f-divergence risk measure is applied in an ERM context where only nonnegative values appear.
>
> > Another important drawback I see about the proposed risk measures is their computability. [..]
>
> We acknowledge that we have not made it clear enough that the risk measures we study are indeed useful in practice. Hence we have expanded on Section 8 ("Application"), supplementing it with more references. The f-divergence risk measure, which is the focus of our work, can be implemented in a single pytorch line due to Equation 5, effectively without any overhead. The computation of spectral risk measures is more involved, but we now provide the appropriate references. We agree that many questions regarding the application/estimation/computation of risk measures in machine learning are still open; we would like to tackle these in a separate paper.
>
>
> As to the minor comments/mistakes which were raised, we fully agree and are thankful for spotting those. In our revised submission, we take them into account. Also, we have added a short section "1.3 - Use in Machine Learning" to the paper, which clarifies how risk measures can be used and have been used in machine learning. We refer to empirical evidence regarding the usefulness and feasibility of risk measures in machine learning.
>
> Again, we thank reviewer chCF for their constructive comments which we believe have improved the revised version.
>
> [1]: R. Tyrrell Rockafellar and Stan Uryasev. The fundamental risk quadrangle in risk management, optimization and statistical estimation. Surveys in
> Operations Research and Management Science, 18(1-2):33–53, 2013.

---

> > ### Comment · Reviewer_chCF · 2022-10-30
> > **Answer**
> >
> > I thank the authors for their feedback. Section 1.3 and the extended discussion in Section 8 of the revised version indeed help.
> >
> > Regarding the asymmetry, I feel that a remark might still be in order when introducing $V_g$. Beyond recalling the examples where a gain/loss asymmetry is needed, it would anticipate that although the quantities studied in the remaining of the paper are symmetric (they are either based on $||\cdot||_{V_g}$ or $R_g$), the asymmetry is needed to properly define $R_g$ in the infimal convolution.
> >
> > I agree with Reviewer gcQw that a remark on the interpretation of small loss tails would be valuable.
> >
> > Provided these modification, I would recommend acceptance as I globally feel positive about the paper.

---

> > > ### Author Response · Authors · 2022-11-03
> > > **Answer**
> > >
> > > We are glad that you feel positive about the paper; in the final version, we will add the suggested remarks.

---

### Decision · Action_Editors · 2022-12-09

**Recommendation:** Accept with minor revision

**Comment:**

The authors have addressed the concerns raised by reviewers, as well as incorporated changes to the presentation, especially towards providing descriptions of the work in closer-to-lay terms. The authors have promised to do another round of proof checking, which I am fine to permit, and so I'm recommending acceptance, with minor revisions, to allow these small changes. I'll add that the opening sentence of the second paragraph (starting "Another concern...") is a garden path sentence for me, and should be revised to have simpler structure.

**Audience:**

The paper will be of interest to theoreticians interested in alternatives to empirical risk minimization that are more sensitive to the tails of losses. Such a shift in focus may be appropriate in settings where there are asymmetries to losses. Risk averse RL is held up as one example, as a robust ML and fair ML. The paper is perhaps best viewed as a resource on a variety of relevant subtopics (risk measures, tail sensitivity, Orlicz norms, rearrangement invariant Banach spaces, Orlics deviation inequalities, and interrelationships thereof).

**Claims And Evidence:**

This paper draws attention to the problem of replacing empirical risk minimization with risk measure minimization, employed to shift focus from average loss (pandering to the masses) to extreme losses or tail losses (tailoring to the tails). One sees this shift employed already in approaches to robust machine learning and also fairness. The paper surveys a range of topics on risk measures and their tail sensitivy, tying together disparate areas, several of which are unlikely to be known to most machine learning researchers. The authors extend some existing notions to build such connections. Besides definitions, the authors also point out tools for concentration inequalities in these novel settings that may not be widely known. Reviewers agreed that the paper was well-written, if dense, and would be a valuable resource for theoreticians.